# Optineurin provides a mitophagy contact site for TBK1 activation

Koji Yamano [1,2,✉], Momoha Sawada[1], Reika Kikuchi[1,2], Kafu Nagataki[3], Waka Kojima[1,2], Ryu Endo[1], Hiroki Kinefuchi[1], Atsushi Sugihara[3], Tomoshige Fujino [3], Aiko Watanabe[1], Keiji Tanaka[4], Gosuke Hayashi[3], Hiroshi Murakami [3] & Noriyuki Matsuda [1,2]

## Abstract

**Tank-binding kinase 1 (TBK1) is a Ser/Thr kinase that is involved in many intracellular processes, such as innate immunity, cell cycle, and apoptosis. TBK1 is also important for phosphorylating the autophagy adaptors that mediate the selective autophagic removal of damaged mitochondria. However, the mechanism by which PINK1-Parkin-mediated mitophagy activates TBK1 remains largely unknown. Here, we show that the autophagy adaptor optineurin (OPTN) provides a unique platform for TBK1 activation. Both the OPTN-ubiquitin and the OPTN-pre-autophagosomal structure (PAS) interaction axes facilitate assembly of the OPTN-TBK1 complex at a contact sites between damaged mitochondria and the autophagosome formation sites. At this assembly point, a positive feedback loop for TBK1 activation is initiated that accelerates hetero-autophosphorylation of the protein. Expression of monobodies engineered here to bind OPTN impaired OPTN accumulation at contact sites, as well as the subsequent activation of TBK1, thereby inhibiting mitochondrial degradation. Taken together, these data show that a positive and reciprocal relationship between OPTN and TBK1 initiates autophagosome biogenesis on damaged mitochondria.**

**Keywords** Autophagy; Mitochondria; Parkin; PINK1; Ubiquitin
**Subject Category** Autophagy & Cell Death

## Introduction

Macroautophagy (hereafter referred to as autophagy) is a conserved catabolic process that delivers cytoplasmic components, organelles and cellular pathogens to lysosomes for degradation. These cargo/substrates are encapsulated by double membrane structures called autophagosomes that subsequently fuse with lysosomes for cargo degradation. Various proteins have been identified as drivers of the de novo autophagosomal membrane synthesis. Although autophagosome biogenesis is induced by various signals including starvation, oxidative stress, and DNA damage, the unique process of autophagosome formation begins with ATG9A-containing vesicles, and the ULK complex that, in mammals, is comprised of FIP200, ATG13, ATG101 and ULK1/2 kinases. ATG9A functions as a lipid scramblase and the ATG9A-containing vesicles act as initial isolation membrane seeds for autophagosome formation (Maeda et al, 2020; Matoba et al, 2020; Sawa-Makarska et al, 2020). Incorporation of multiple ULK1 complexes into the developing scaffold further contributes to the recruitment of downstream autophagy proteins (Fujioka et al, 2020; Yamamoto et al, 2016). Autophosphorylation of ULK1/2 in turn triggers ULK1/2-dependent phosphorylation of diverse downstream substrates including ATG9A, BECN1, and ATG14 in mammals (Egan et al, 2015; Park et al, 2016; Russell et al, 2013; Wold et al, 2016; Zhou et al, 2017). Subsequent activation of a phosphatidylinositol 3-kinase complex composed of BECN1, ATG14, VPS15, and VPS34 is required for the production of the phosphatidylinositol 3-phosphate that facilitates recruitment of the ATG2-WIPI complex to the autophagosomal formation site (Obara et al, 2008). Two ubiquitin-like conjugation systems, the ATG5-ATG12/ATG16L1 complex and phosphatidylethanolamine-conjugated ATG8 (LC3 and GABARAP family proteins in mammals), are important for elongation of the isolation membrane as well as efficient degradation of the inner autophagosomal membrane in lysosomes (Kabeya et al, 2000; Nguyen et al, 2016; Tsuboyama et al, 2016).

While autophagy has been recognized as non-selective degradation process, multiple lines of evidence show that selective-type autophagy mediates the clearance of specific unwanted/damaged organelles (Vargas et al, 2022). PINK1/Parkin-mediated mitophagy is one of the best characterized selective pathways that functions to maintain cellular homeostasis (Onishi et al, 2021; Pickles et al, 2018; Yamano et al, 2016). Defects in mitophagy have been linked to neurodegeneration in Parkinson's disease. The E3 ligase Parkin and the mitochondrial kinase PINK1, both of which are causal gene products of familial Parkinson disease (Kitada et al, 1998; Valente et al, 2004), coordinately function to generate poly-ubiquitin (Ub) chains on damaged mitochondria. Under healthy mitochondrial

[1]Department of Biomolecular Pathogenesis, Medical Research Institute, Tokyo Medical and Dental University, 1-5-45 Yushima, Bunkyo-ku, Tokyo 113-8510, Japan. [2]Ubiquitin Project, Tokyo Metropolitan Institute of Medical Science, 2-1-6 Kamikitazawa, Setagaya-ku, Tokyo 156-8506, Japan. [3]Department of Biomolecular Engineering, Graduate School of Engineering, Nagoya University, Furo-cho, Chikusa-ku, Nagoya 464-8603, Japan. [4]Protein Metabolism Project, Tokyo Metropolitan Institute of Medical Science, 2-1-6 Kamikitazawa, Setagaya-ku, Tokyo 156-8506, Japan. ✉E-mail: kojibiom@tmd.ac.jp

conditions, PINK1 is constitutively and rapidly degraded by proteasomes after cleaving the mitochondrial targeting sequence and the associated hydrophobic transmembrane segments in the mitochondria (Deas et al, 2011; Greene et al, 2012; Jin et al, 2010; Meissner et al, 2011; Sekine et al, 2019; Shi et al, 2011; Yamano and Youle, 2013). In contrast, when mitochondria are damaged by aging and/or exposure to toxic compounds and lose their membrane potential, PINK1 accumulates on the outer mitochondrial membrane (OMM) via associations with the TOMM complex (Hasson et al, 2013; Lazarou et al, 2012; Okatsu et al, 2013). PINK1 kinase activity is essential for Parkin recruitment to damaged mitochondria as well as activation of Parkin E3 activity (Matsuda et al, 2010; Narendra et al, 2008; Narendra et al, 2010). PINK1 phosphorylates Ub and the Ub-like (UBL) domain of Parkin at Serine 65 (Kane et al, 2014; Kazlauskaite et al, 2014; Kondapalli et al, 2012; Koyano et al, 2014; Shiba-Fukushima et al, 2012). Binding phosphorylated Ub and UBL phosphorylation promote conformational changes in the Parkin structure that convert the inactivated form to the activated form (Gladkova et al, 2018; Gundogdu et al, 2021; Sauve et al, 2018; Yamano et al, 2015). Consequently, PINK1, Parkin, Ub, and various E2 enzymes form a positive feedback loop of Ub amplification on damaged mitochondria (Hayashida et al, 2022; Okatsu et al, 2015; Ordureau et al, 2015; Ordureau et al, 2014). Ub-coated mitochondria are then recognized by autophagy adaptors (Lazarou et al, 2015; Yamano and Kojima, 2021). Autophagy adaptors (also known as autophagy receptors) are defined as proteins containing both a Ub-binding domain and an LC3 interacting region (LIR). Five different adaptors, OPTN, NDP52, TAX1BP1, p62, and NBR1, have been identified to date in mammals (Adriaenssens et al, 2022). Autophagy adaptors thus function as a molecular bridge linking ubiquitinated cargo with the autophagy machinery. In Parkin-mediated mitophagy, OPTN and NDP52 are required for autophagosome formation (Lazarou et al, 2015). NDP52 is composed of several functional domains such as SKICH for interactions with AZI2 and TBK1BP1, LIR for LC3 binding, coiled-coil for homo-dimerization, and two C-terminal zinc fingers for interactions with the Ub-chain (Yamano and Kojima, 2021). In addition to LC3 binding, NDP52 also directly binds FIP200, which promotes recruitment of the ULK complex to damaged mitochondria (Fu et al, 2021; Ravenhill et al, 2019; Shi et al, 2020; Vargas et al, 2019).

OPTN is another essential mitophagy adaptor composed of long coiled-coil regions, a leucin zipper and an LIR as well as UBAN and zinc finger domains (Yamano and Kojima, 2021). Recent experimental evidence has shown that OPTN interacts with multiple autophagy components during Parkin-mediated mitophagy. In addition to LC3 binding via the LIR, OPTN can recruit ATG9A vesicles through its leucine zipper motif (Yamano et al, 2020). Furthermore, the LIR in OPTN can interact with FIP200 when OPTN S177 is phosphorylated (Zhou et al, 2021). Given the multiple interactions linking NDP52 and OPTN with autophagy components, they are essential adaptors for Parkin-mediated mitophagy.

In addition to autophagy adaptors, TBK1-mediated phosphorylation regulates Ub-dependent selective autophagy (Wild et al, 2011). TBK1 interacts directly with OPTN and indirectly with NDP52 and TAX1BP1 via AZI2 and TBKBP1 (Li et al, 2016; Thurston et al, 2009). During Parkin-mediated mitophagy,

autophosphorylation of TBK1 at S172 converts the protein to its activated form (Heo et al, 2015; Zhang et al, 2019). Activated TBK1 subsequently phosphorylates multiple Ser/Thr residues in the autophagy adaptors (Richter et al, 2016). Furthermore, it also phosphorylates RAB7A, which promotes ATG9A recruitment to damaged mitochondria (Heo et al, 2018), and LC3C and GABARAPL2 to expand the isolation membranes (Herhaus et al, 2020). While TBK1 autophosphorylation is a prerequisite for TBK1 activation, the mechanism underlying this mitophagy-driven step remains unknown.

In this study, we found that *TBK1* deletion inhibits the localization of OPTN, but not NDP52, at the autophagosome formation site during Parkin-mediated mitophagy, and that deletion of *OPTN* inhibits TBK1 autophosphorylation. We also found that activated TBK1 is selectively and rapidly degraded via autophagy. Two different interaction axes, OPTN-Ub and OPTN-autophagy, enable the formation of a contact site between the isolation membrane and damaged mitochondria that supports autophosphorylation of TBK1 by neighboring TBK1. In addition, monobodies engineered in this study against OPTN inhibited formation of the OPTN-dependent contact site, which suppressed both TBK1 autophosphorylation and the elimination of damaged mitochondria. These results indicate that OPTN-TBK1 catalyzes a unique autophagy initiation step that is different from that mediated by NDP52.

# Results

## OPTN is required for TBK1 phosphorylation and subsequent autophagic degradation

Although autophosphorylation of TBK1 at S172 was reported to increase during Parkin-mediated mitophagy (Heo et al, 2015), how it is regulated remains largely unknown. Immunoblots of HeLa cells stably expressing Parkin and which had been treated with valinomycin for varying lengths of time revealed phosphorylation of S172 in TBK1. As shown in Fig. 1A, this phosphorylation event was observed after 30 min valinomycin treatment, with the levels gradually decreasing over time. While phosphatase-mediated de-phosphorylation has been suggested for the OPTN-TBK1 axis (Kachaner et al, 2012), we found that reduction of the TBK1 pS172 signal was blocked when cells were treated with the lysosomal inhibitor bafilomycin A1 (Fig. 1A,B). Concanamycin A, a different lysosomal inhibitor, similarly enhanced the TBK1 pS172 signal, whereas the proteasomal inhibitors epoxomicin and MG132 had no effect (Fig. 1C,D), indicating that the TBK1 phosphorylation triggered by Parkin-mediated OMM ubiquitination is rapidly degraded in lysosomes by the autophagy pathway. Consequently, to limit autophagic degradation of activated TBK1 during Parkin-mediated mitophagy, we used bafilomycin A1 for subsequent analyses. TBK1 phosphorylates S177 in OPTN, which undergoes ubiquitination and autophagic degradation (Fig. 1A and Appendix Fig. S1). In response to Parkin-mediated mitophagy, endogenous TBK1 and OPTN were recruited and subsequently phosphorylated at autophagosome formation sites that were associated with WIPI1 and in close proximity to mitochondria (Figs. EV1,EV2). This is consistent with the immunoblot data (Fig. 1A,B). Furthermore, we found that Parkin expression levels

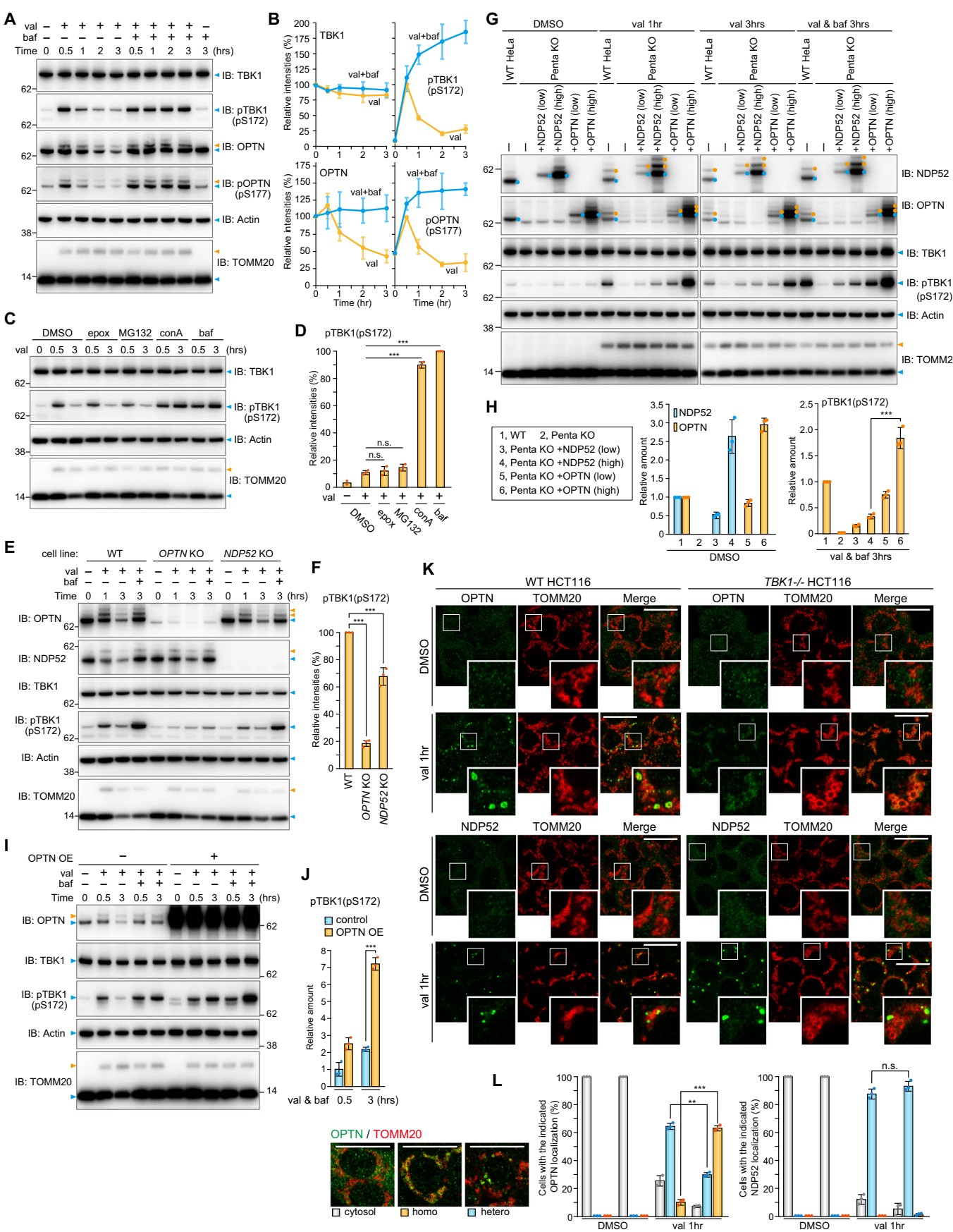

**Figure 1. Activated TBK1 induced by OPTN during mitophagy is rapidly degraded via autophagy.**

(A) HeLa cells stably expressing Parkin were treated with valinomycin (val) and bafilomycin (baf) for the indicated times. Total cell lysates were analyzed by immunoblotting (IB). (B) The levels of proteins indicated in (A) were quantified. TBK1 and OPTN levels without valinomycin were set to 100. pTBK1(pS172) and pOPTN(pS177) levels after 0.5 h val were set to 100. Error bars represent mean ± s.d. of three independent experiments. (C) Parkin-expressing HeLa cells treated with the indicated inhibitors in the presence of val were analyzed by IB. Epox; Epoxomicin, conA; Concanamycin A. (D) pTBK1(pS172) levels after 3 h val treatment in (C) were quantified. Error bars represent mean ± s.d. of three independent experiments. (E) WT, *OPTN* KO, and *NDP52* KO HeLa cells stably expressing Parkin were treated with val and baf for the indicated times. Total cell lysates were analyzed by IB. (F) pTBK1(pS172) levels after 3 h val and baf treatment in (E) were quantified. Error bars represent mean ± s.d. of three independent experiments. (G) Parkin-expressing WT HeLa, Parkin-expressing Penta KO HeLa, or those expressing 3FLAG-NDP52 and 3FLAG-OPTN at different levels (indicated as low and high) were treated with DMSO and val with or without baf for the indicated times, and then analyzed by IB. (H) NDP52 and OPTN in cells treated with DMSO and pTBK1(pS172) in cells treated with val and baf for 3 h in (G) were quantified. Protein levels in WT cells were set to 1. Error bars represent mean ± s.d. of three independent experiments. (I) Parkin-expressing HeLa cells with or without 3FLAG-OPTN overexpression (OE) were treated with val and baf for the indicated times and analyzed by IB. (J) pTBK1(pS172) levels in Parkin-expressing HeLa cells with or without OPTN OE in (I) were quantified. Error bars represent mean ± s.d. of three independent experiments. pTBK1 levels in control cells after 0.5 h val and baf treatment were set to 1. (K) WT and *TBK1-/-* HCT116 cells stably expressing Parkin were treated with DMSO or val for 1 h and then immunostained with the indicated antibodies. Bars, 10 μm. (L) OPTN and NDP52 localization in cells in (K) were quantified manually. Heterogenous denotes that the protein is assembled on mitochondria and unevenly distributed, whereas homogenous denotes that the protein localizes uniformly on mitochondria. The examples of OPTN distribution are shown in the left. Bars, 10 μm. Error bars represent mean ± s.d. of three independent experiments. Data information: n.s. not significant, **P < 0.01, ***P < 0.001 by two-tailed Student's *t*-test (H,J,L) and two-tailed Dunnett's test (D,F). The light blue and orange arrowheads/dots indicate unmodified and ubiquitinated bands, respectively (A,C,E,G,I). Source data are available online for this figure.

moderately affect TBK1 phosphorylation as higher Parkin levels induced more rapid phosphorylation of TBK1 (Appendix Fig. S2). Since OPTN and NDP52 are essential adaptors for mitophagy and TBK1 is either directly (OPTN) or indirectly (NDP52) associated with both, we next examined the requirement of the two adaptors for TBK1 autophosphorylation. *OPTN* KO drastically reduced pS172 in TBK1 (Fig. 1E,F). *NDP52* KO also reduced TBK1 phosphorylation, but the effect was more moderate than that observed with *OPTN* deletion (Fig. 1E,F). Similar results were observed when different levels of either OPTN or NDP52 were recombinantly expressed in Penta KO HeLa cells lacking all five autophagy adaptors (Fig. 1G,H). Furthermore, there was a significant increase in the TBK1 pS172 signal in cells over-expressing OPTN, which had been treated with valinomycin in the presence of bafilomycin A1 for 3 h, as compared to cells that lacked OPTN overexpression (Fig. 1I,J). These results indicate that OPTN is a crucial rate-limiting factor for the autophosphorylation of TBK1 that occurs in response to Parkin-mediated mitophagy.

We next examined the recruitment of OPTN and NDP52 to damaged mitochondria in cells lacking TBK1. In WT cells, both OPTN and NDP52 formed cup-shaped and/or sphere-like structures on damaged mitochondria (Fig. 1K) that were previously reported to co-localize with LC3B-labeled autophagic membranes (Lazarou et al, 2015; Padman et al, 2019; Yamano et al, 2020). In *TBK1-/-* HCT116 cells, the NDP52 structures were larger with more pronounced signals (Fig. 1K). In sharp contrast, OPTN was homogenously distributed on mitochondria rather than being heterogeneously distributed in concentrated spots (Fig. 1K,L). Thus, although NDP52 and OPTN are essential mitophagy adaptors that interact with TBK1, their recruitment to the autophagosome formation site is differently regulated by TBK1.

## OPTN association with the autophagy machinery is required for TBK1 activation

We next investigated how OPTN induces TBK1 autophosphorylation during mitophagy. Previously, OPTN binding of Ub was shown to be a requirement for TBK1 autophosphorylation (Heo et al, 2015). Here, we found that OPTN association with the autophagy machinery is also required for TBK1 activation. The OPTN leucine zipper and LIR motifs are important for the

recruitment of ATG9A vesicles, FIP200, and LC3 family proteins as 4LA mutations in the leucine zipper motif disrupted ATG9A vesicle interactions (Yamano et al, 2020) and an F178A mutation in the LIR inhibited interactions with FIP200 and LC3 family proteins (Johansen and Lamark, 2020; Richter et al, 2016; Zhou et al, 2021). Penta KO HeLa cells stably expressing OPTN WT or the previously defined OPTN mutations along with Parkin were treated with valinomycin and the phosphorylation state of TBK1 was assessed. First, TBK1 phosphorylation was not apparent in the Penta KO cells (none in Fig. 2A) even after 3 h with valinomycin (Fig. 2A), indicating that autophagy adaptors are essential for TBK1 activation. OPTN WT expression in Penta KO cells induced autophosphorylation of TBK1 at S172 after 1 h valinomycin treatment (Fig. 2A). This signal decreased after 3 h because of lysosomal degradation, but was enhanced in the presence of bafilomycin A1 (Fig. 2A). In contrast, the F178A, 4LA, and double 4LA/F178A OPTN mutants reduced TBK1 phosphorylation as compared to cells expressing OPTN WT, albeit with more moderate effects observed with the F178A mutations (Fig. 2A–D). When the intracellular localization was observed, OPTN WT signals were visible as cup-shaped and/or sphere-like structures on the mitochondria that colocalized with WIPI2, an autophagosome formation site marker (Fig. 2E,F). Despite OPTN association with damaged mitochondria, the WIPI2 foci associated with formation of the isolation membrane were reduced in the F178A mutant and severely inhibited by the 4LA and 4LA/F178A double mutants (Fig. 2E,F). Furthermore, while OPTN WT was heterogeneously recruited to mitochondria where WIPI2 was frequently colocalized, the OPTN mutants, in particular 4LA and 4LA/F178A, were uniformly (i.e., homogenously) distributed to mitochondria (Fig. 2G). These results strongly suggest that association of OPTN with mitochondria alone is not sufficient, and that recruitment of OPTN to the autophagosome formation site, where isolation membranes are synthesized and expanded, is required for full activation of TBK1.

Next, to confirm that autophagy machinery is required for TBK1 activation during Parkin-mediated mitophagy, autophagy gene KO (*FIP200* KO, *ATG5* KO and *ATG9A* KO) cells were used (Fig. EV3A). The levels of FIP200, ATG5, ATG9A, and lipidated LC3B in Penta KO cells are comparable to those in WT cells (Fig. EV3A). Since LC3B lipidation is catalyzed by ATG5, but not

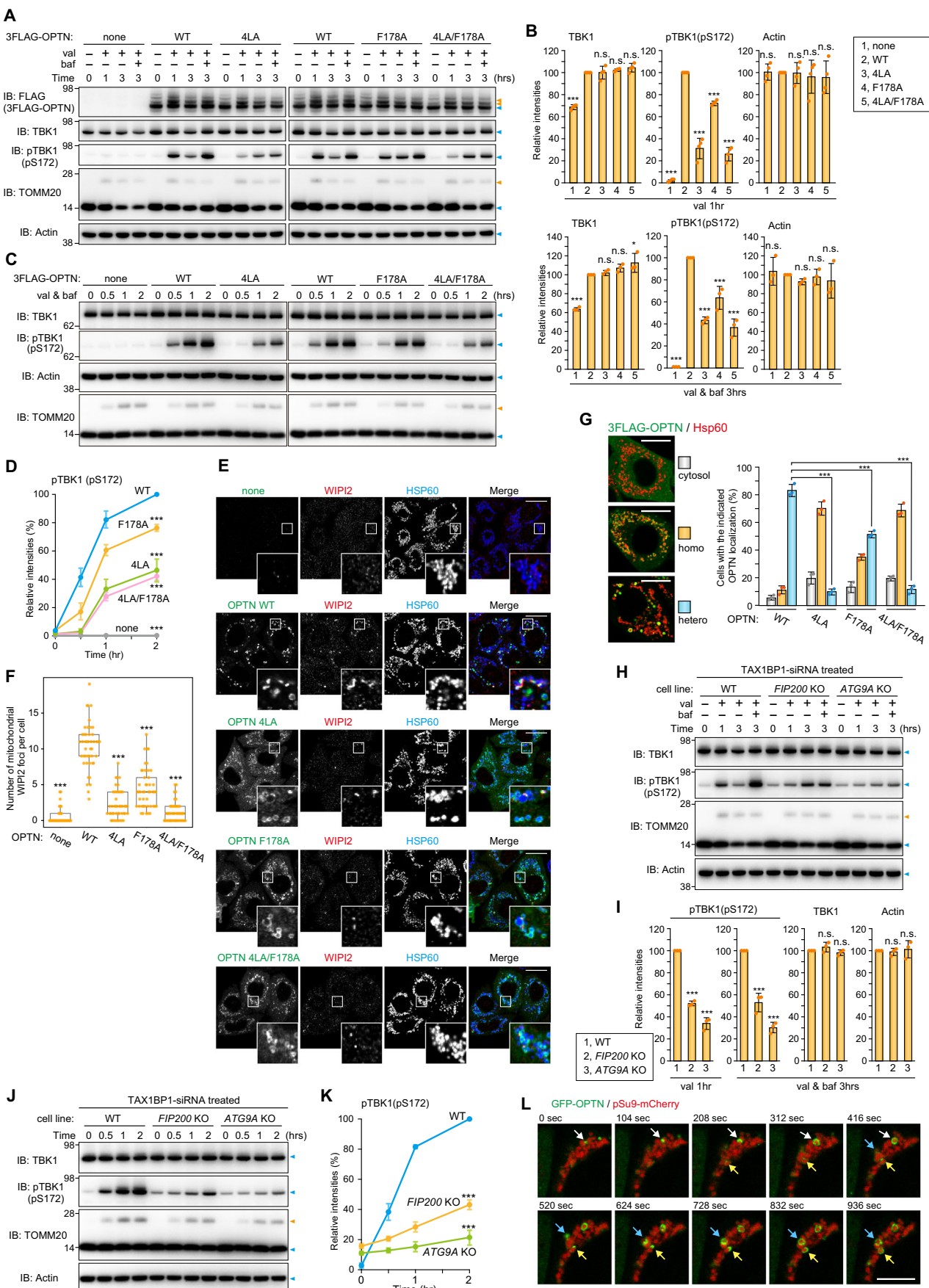

**Figure 2. TBK1 activation requires OPTN-autophagic machinery associations.**

(A) Penta KO HeLa cells stably co-expressing Parkin and 3FLAG-OPTN WT or the indicated mutants were treated with valinomycin (val) and bafilomycin (baf) for the indicated times. Total cell lysates were analyzed by immunoblotting (IB). (B) The levels of proteins indicated in (A) were quantified. The protein levels in cells expressing OPTN WT were set to 100. Error bars represent mean ± s.d. of three independent experiments. (C) The cells in (A) were treated with val in the presence of baf for the indicated times. Total cell lysates were analyzed by IB. (D) pTBK1(pS172) levels in (C) were quantified. pTBK1 levels in cells expressing OPTN WT after 2 h val and baf treatment were set to 100. Error bars represent mean ± s.d. of three independent experiments. (E) The cells in (A) treated with val for 1 h were immunostained. OPTN was detected with an anti-FLAG antibody. Bars, 10 µm. (F) The number of WIPI2 foci near mitochondria in (E) per cell were quantified manually. The data is shown as a box plot, with the box indicating the inter-quarter range (IQR), the whiskers showing the range of values that are within 1.5 × IQR and a horizontal line indicating the median. More than 45 cells in each cell line were examined in two independent experiments. (G) OPTN localization in cells in (E) was quantified manually. Heterogenous denotes that the protein is assembled on mitochondria and unevenly distributed, whereas homogenous denotes that the protein localizes uniformly on mitochondria. The examples of 3FLAG-OPTN distribution are shown in the left. Bars, 10 µm. Error bars represent mean ± s.d. of three independent experiments. (H) TAX1BP1 siRNA-treated WT, *FIP200* KO and *ATG9A* KO HeLa cells stably expressing Parkin were treated with val and baf for the indicated times. Total cell lysates were analyzed by IB. (I) The levels of proteins indicated in (H) were quantified. The protein levels in WT cells were set to 100. Error bars represent mean ± s.d. of three independent experiments. (J) The cells in (H) were treated with val in the presence of baf for the indicated times. Total cell lysates were analyzed by IB. (K) pTBK1(pS172) levels in (J) were quantified. pTBK1 levels in WT cells after 2 h val and baf treatment were set to 100. Error bars represent mean ± s.d. of three independent experiments. (L) WT HeLa stably expressing GFP-OPTN and pSu9-mCherry were treated with val and analyzed by time-lapse microscopy. Selected frames at 15 min after val treatment (set to 0 s, time-interval every 104 s) are shown. OPTN recruitment to mitochondria is indicated by arrows. Bars, 10 µm. Data information: n.s. not significant, *P < 0.05, ***P < 0.001 by two-tailed Dunnett's test (B,D,F,G,I,K). The light blue and orange arrowheads indicate unmodified and ubiquitinated bands, respectively (A,C,H,J). Source data are available online for this figure.

by FIP200 or ATG9A, the lipidated form is absent only in *ATG5* KO cells (Hanada et al, 2007). When Parkin-mediated ubiquitination was induced, phosphorylation of TBK1 at S172 was observed in WT cells after 1 h valinomycin and was significantly enhanced after 3 h valinomycin in the presence of bafilomycin A1 (Fig. EV3B). Although autophagy gene KO cells possess a certain amount of phosphorylated TBK1 even under basal growing conditions (i.e., without valinomycin treatment), the levels of TBK1 pS172 increased slowly during mitophagy (Fig. EV3B). To measure the amounts of newly generated TBK1 pS172, we subtracted the basal phosphorylation signal from the signal generated post-valinomycin in the presence of bafilomycin A1 (3 h). In WT cells, the newly generated TBK1 pS172 score was ~95 but was ~26 in *ATG9A* KO cells (Fig. EV3C). It is assumed that autophagy gene KOs constitutively inhibit autophagy, thereby causing phosphorylated TBK1 to accumulate under basal conditions (Fig. EV3B,C). Therefore, to simultaneously inhibit autophagy and induce Parkin-mediated ubiquitination, we next knocked down FIP200 and ATG9A by siRNA before inducing mitochondrial damage. As shown in Fig. EV3D, two different siRNAs efficiently knocked down FIP200 and ATG9A. While there was a slight increase in TBK1 phosphorylation in the FIP200 and ATG9A siRNA-treated cells under basal conditions, the TBK1 pS172 signal following Parkin-mediated ubiquitination was only moderately elevated when compared to that in control cells (Fig. EV3E,F). These results demonstrate that both ubiquitination and the autophagy core subunits are required for TBK1 autophosphorylation of S172 during Parkin-mediated mitophagy. Because TAX1BP1 mediates pTBK1 (pS172) accumulation in both *FIP200* KO and *ATG9A* KO cells under basal conditions (see Fig. 3F for the detail later), the introduction of TAX1BP1 siRNA should decrease pTBK1 basal levels and allow for discrimination of mitophagy-dependent TBK1 phosphorylation. WT, *FIP200* KO, and *ATG9A* KO cells were treated with TAX1BP1 siRNA prior to initiation of Parkin-mediated ubiquitination with valinomycin. pTBK1 (pS172) levels in WT HeLa cells were elevated in response to valinomycin treatment in the presence of bafilomycin for 3 h, whereas pTBK1 levels were much lower in the *FIP200* KO and *ATG9A* KO cell lines (Fig. 2H,I). Furthermore, the pTBK1 levels only increased slightly over time in the autophagy knockout lines, in particular *ATG9A* KO cells (Fig. 2J,K). These results demonstrate that, in addition to

mitochondrial ubiquitination, TBK1 autophosphorylation of S172 during Parkin-mediated mitophagy is dependent on FIP200 and ATG9A activity in isolation membrane formation.

Time-lapse microscopy showed that during Parkin-mediated mitophagy, GFP-OPTN recruitment to regions of mitochondria labeled by pSu9-mCherry manifested as small dot-like structures that over time expanded to become larger spherical structures (Fig. 2L). This strongly suggests that OPTN is assembled at the contact site between isolation membranes and damaged mitochondria where TBK1 is fully activated following autophosphorylation.

## TAX1BP1, but not OPTN, mediates TBK1 phosphorylation when basal autophagy is impaired

Because cells with autophagy gene deletions accumulate phosphorylated TBK1 (Fig. EV3B), we next examined whether OPTN mediates TBK1 autophosphorylation under basal conditions. Protein levels for the autophagy adaptors NDP52, TAX1BP1, p62, and NBR1 increased following KO of the autophagy genes *FIP200*, *ATG5* and *ATG9A* (Fig. EV4), whereas OPTN levels remained constant (Fig. EV4). Furthermore, TBKBP1, the mediator between TBK1 and NDP52/TAX1BP1, became unstable in Penta KO cells and the electrophoretic migration of AZI2, another TBK1 mediator, was altered following deletion of either *FIP200* or *ATG9A* (Fig. EV4A). Although total TBK1 levels were comparable between WT and autophagy gene KO cells, TBK1 pS172 levels were moderately increased in *ATG5* KO cells and significantly elevated in *FIP200* KO and *ATG9A* KO cells (Fig. EV4). p62 (S403) phosphorylation was also elevated (Fig. EV4). We next examined the subcellular localization pattern of autophagy adaptors in the autophagy KO cells. TAX1BP1, p62, NBR1, and phosphorylated TBK1 strongly colocalized with Ub-positive condensates in *FIP200* KO and *ATG9A* KO cells (Figs. 3A and EV5A). Ferritin was also concentrated on the p62-positive condensates (Fig EV5B), which is consistent with previous findings (Kishi-Itakura et al, 2014). In contrast, the OPTN signal in the Ub condensates was relatively low when compared with the other autophagy adaptors (Fig. 3A).

Biochemical analyses also showed that insoluble forms of the autophagy adaptors were elevated in *FIP200* KO and *ATG9A* KO cells (Fig. 3B,C). After solubilizing WT, *FIP200* KO, and *ATG9* KO cells with 2% TX-100, the soluble and insoluble pellet fractions

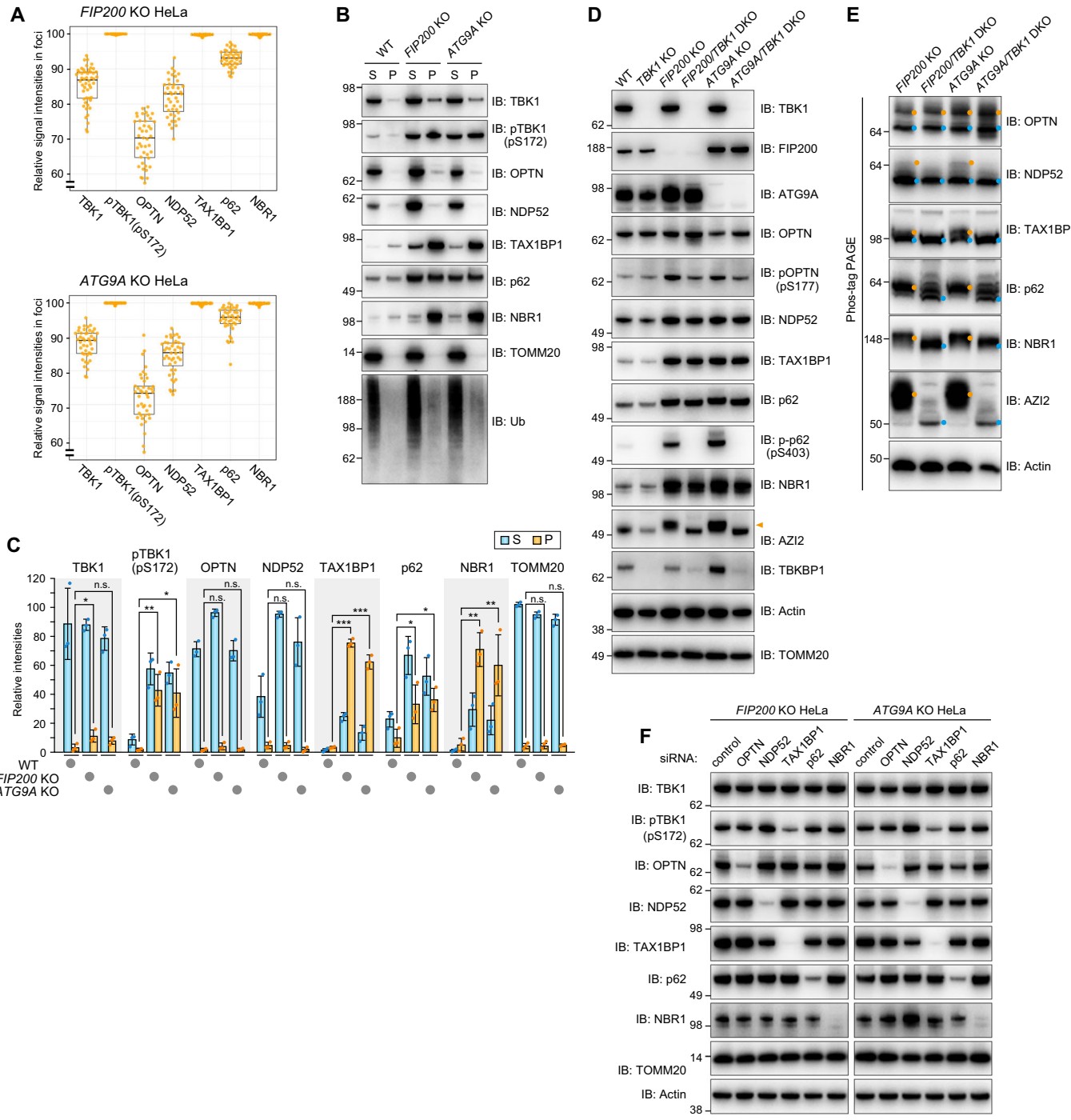

**Figure 3. TAX1BP1, but not OPTN, increases TBK1 phosphorylation under basal conditions in autophagy gene KO cells.**

(A) WT, *FIP200* KO, and *ATG9A* KO HeLa cells were immunostained (see the raw data in Fig. EV5A). The ratios of the signal intensity for each protein in the Ub-condensates relative to those in the cytosol in the *FIP200* KO and *ATG9A* KO cells were calculated as "relative signal intensities in foci". The data is shown as a box plot, with the box indicating the inter-quarter range (IQR), the whiskers showing the range of values that are within 1.5 × IQR and a horizontal line indicating the median. More than 40 foci (more than 20 cells) in each condition were examined in two independent experiments. (B) WT, *FIP200* KO, and *ATG9A* KO HeLa cells were solubilized using 2% Triton X-100. The supernatants (S) and pellets (P) were separated by centrifugation and then analyzed by immunoblotting (IB). (C) The amount of protein recovered in S and P in (B) was quantified. The total signal intensity (S + P) in *FIP200* KO cells for each protein was set to 100. Error bars represent mean ± s.d. of three independent experiments. n.s. not significant, *P < 0.05, **P < 0.01, ***P < 0.001 by two-tailed Dunnett's test. (D) The indicated proteins in WT, *TBK1* KO, *FIP200* KO, *FIP200/TBK1* DKO, *ATG9A* KO, and *ATG9A/TBK1* DKO HeLa cells were analyzed by IB. The orange arrowhead indicates a slower migrating AZI2 bands. (E) Total cell lysates prepared in (D) were analyzed by Phos-tag PAGE followed by IB. The light blue and orange dots indicate bands for unmodified proteins and proteins phosphorylated by TBK1, respectively. (F) *FIP200* KO and *ATG9A* KO cells treated with the indicated siRNA were analyzed by IB. Source data are available online for this figure.

were separated by centrifugation. We found that autophagy KO resulted in insoluble Ub smears and a prominent accumulation of TAX1BP1, p62, and NBR1 in the insoluble fraction (Fig. 3B,C). TBK1 accumulation in the insoluble fraction was slightly increased in the *FIP200* KO and *ATG9A* KO cells (Fig. 3B,C). Furthermore, the ratio of phosphorylated TBK1 in the insoluble fraction was significantly higher than total TBK1 (Fig. 3B,C), suggesting that the phosphorylated form is specifically targeted to the insoluble fraction. However, a large percentage of OPTN was maintained in the soluble fraction even in the autophagy KO cells (Fig. 3B,C). Since phosphorylated TBK1 (activated TBK1) accumulates with autophagy adaptors in Ub-condensates, it may induce phosphorylation of the adaptor. To test this possibility, we knocked out *TBK1* in WT, *FIP200* KO, and *ATG9A* KO cells (Fig. 3D). Although autophagy gene deletion promoted an increase in NDP52, TAX1BP1, p62, and NBR1 (Fig. EV4A), their levels were not affected with the concomitant deletion of *TBK1* (Fig. 3D). However, it completely impeded p62 (S403) phosphorylation and altered the electrophoretic migration AZI2 (Fig. 3D). To clarify the phosphorylation state of other autophagy adaptors/mediators, we performed Phos-tag analysis (Fig. 3E). Although electrophoretic migration of OPTN was not affected by *TBK1* deletion, which was previously shown by Heo et al (Heo et al, 2015), the other autophagy adaptors in the *FIP200* and *ATG9A* KO cells resolved as several distinct bands and the slower migrating bands (denoted by orange-colored dots in Fig. 3E) disappeared and/or were only present in their unphosphorylated state (denoted by blue-colored dots in Fig. 3E). These results strongly suggest that, except for OPTN, TBK1 phosphorylates autophagy adaptors when autophagy is impaired.

All autophagy adaptors possess a ubiquitin-binding domain and several adaptors interact with TBK1 either directly or indirectly. Therefore, the autophosphorylation of S172 in TBK1 that is induced in *FIP200* KO or *ATG9A* KO cells could be due to autophagy adaptor-mediated TBK1 recruitment to Ub-condensates. Therefore, we next tested whether S172 phosphorylation is affected by autophagy adaptor knockdown. For this purpose, *FIP200* KO and *ATG9A* KO cells were treated with control siRNA or siRNA for each autophagy adaptor. As shown in Fig. 3F, when TAX1BP1 was knocked down, the level of pTBK1 (pS172) was significantly reduced in both KO cells. These results indicate that TAX1BP1, but not OPTN, mediates TBK1 phosphorylation when autophagy is impaired, and that OPTN-mediated activation of TBK1 is specific to Parkin-mediated mitophagy.

## Engineered multimeric OPTN-Ub condensates bypass the autophagy requirement for TBK1 activation

The above results suggest that formation of the isolation membrane provides a structural support that brings OPTN-TBK1 into close proximity to each other. To examine this, we next sought to engineer OPTN multimerization in a way that bypassed the autophagic machinery requirement for TBK1 autophosphorylation. For this purpose, we used the fluoppi technique (Fig. 4A). Fluoppi can create LLPS (liquid-liquid phase separation) via multimeric interactions (Watanabe et al, 2017; Yamano et al, 2015). An HA-Ash tag that forms a homo-oligomer was fused to a linear six-tandem Ub chain to yield HA-Ash-6Ub, and a homotetrameric humanized Azami-Green (hAG) was fused to OPTN to make hAG-OPTN. Co-expression of HA-Ash-6Ub and hAG-OPTN in cells

induces multimeric interactions (Ash-Ash, hAG-hAG, and OPTN-Ub) that result in the formation of fluoppi foci in the cytosol (Fig. 4A). We and others previously showed that TBK1 rather than the ULK complex is recruited to OPTN-Ub fluoppi foci (Yamano et al, 2020) and that M44Q/L5Q mutations in OPTN impede OPTN-TBK1 interactions (Li et al, 2016). Here, we found that phosphorylation of OPTN (S177) and TBK1 (S172) was induced by the OPTN-Ub fluoppi (Fig. 4B) and that neither OPTN nor TBK1 were phosphorylated following *TBK1* deletion (*TBK1-/-*) (Fig. 4B). Further, signals for phosphorylated TBK1 (S172) and OPTN (S177) were exclusively localized in the foci (Fig. 4C,D), suggesting that TBK1 autophosphorylation occurs in the OPTN condensates. Although OPTN-Ub foci formation still occurred in *TBK1-/-* cells, the phosphorylated OPTN (S177) signal was absent (Fig. 4C–F). These results indicate that OPTN can induce TBK1 autophosphorylation when multimeric OPTNs are sequestered within a particular cellular localization. In other words, Parkin-mediated mitophagy promotes interactions between OPTN and the autophagy components that contribute to OPTN multimerization and concomitant TBK1 autophosphorylation.

## TBK1-ALS-associated mutations affect mitophagy through phosphorylation of TBK1 and OPTN

Gene mutations in both *OPTN* and *TBK1* have been found in patients suffering from amyotrophic lateral sclerosis (ALS) and frontotemporal dementia (FTD), suggesting physiological and molecular links between OPTN and TBK1 (Cirulli et al, 2015; Evans and Holzbaur, 2019). Several ALS-associated mutations have been shown to affect the structure, dimerization state, and kinase activity of TBK1 (Ye et al, 2019). Although mitochondrial recruitment of the TBK1 mutants during mitophagy has been assessed (Harding et al, 2021), the mechanisms underlying how ALS-associated mutations impact mitochondrial degradation remain to be elucidated. TBK1 dimerization is of great interest since the above findings indicate that assembly of the multimeric OPTN-TBK1 complex between the isolation membrane and damaged mitochondria is critical for TBK1 autophosphorylation. We thus examined the effects of various mutations on TBK1 function. R47H is an ALS-associated mutation of the kinase domain. D135N and S172A are engineered kinase inactive mutations, whereas G217R, R357Q, and M559R are ALS-associated mutations that impede dimerization (Ye et al, 2019). To monitor mutational effects on TBK1 stability, we used *TBK1-/-* HCT116 cells stably expressing YFP-P2A-TBK1 in which the P2A self-cleaving peptide was inserted between YFP and TBK1. Post-translation cleavage at this site would yield equivalent amounts of YFP and TBK1. Furthermore, we used a human codon-optimized TBK1 sequence for better expression (see Methods for details). Initially, cells expressing YFP-P2A-TBK1 WT and the various mutants were immunoblotted. As shown in Fig. 5A, comparable YFP signals were detected among the YFP-P2A-TBK1 constructs, but the protein levels of the G217R, R357Q, and M559R mutants were reduced ~50% compared to that of WT TBK1 (Fig. 5A,B left graph), suggesting that dimerization-deficient mutations destabilize TBK1. When inducing Parkin-mediated mitochondrial ubiquitination, OPTN in *TBK1-/-* cells and cells expressing either TBK1 WT or the mutants was efficiently ubiquitinated (Fig. 5A), indicating that its association with damaged mitochondria in which Parkin is

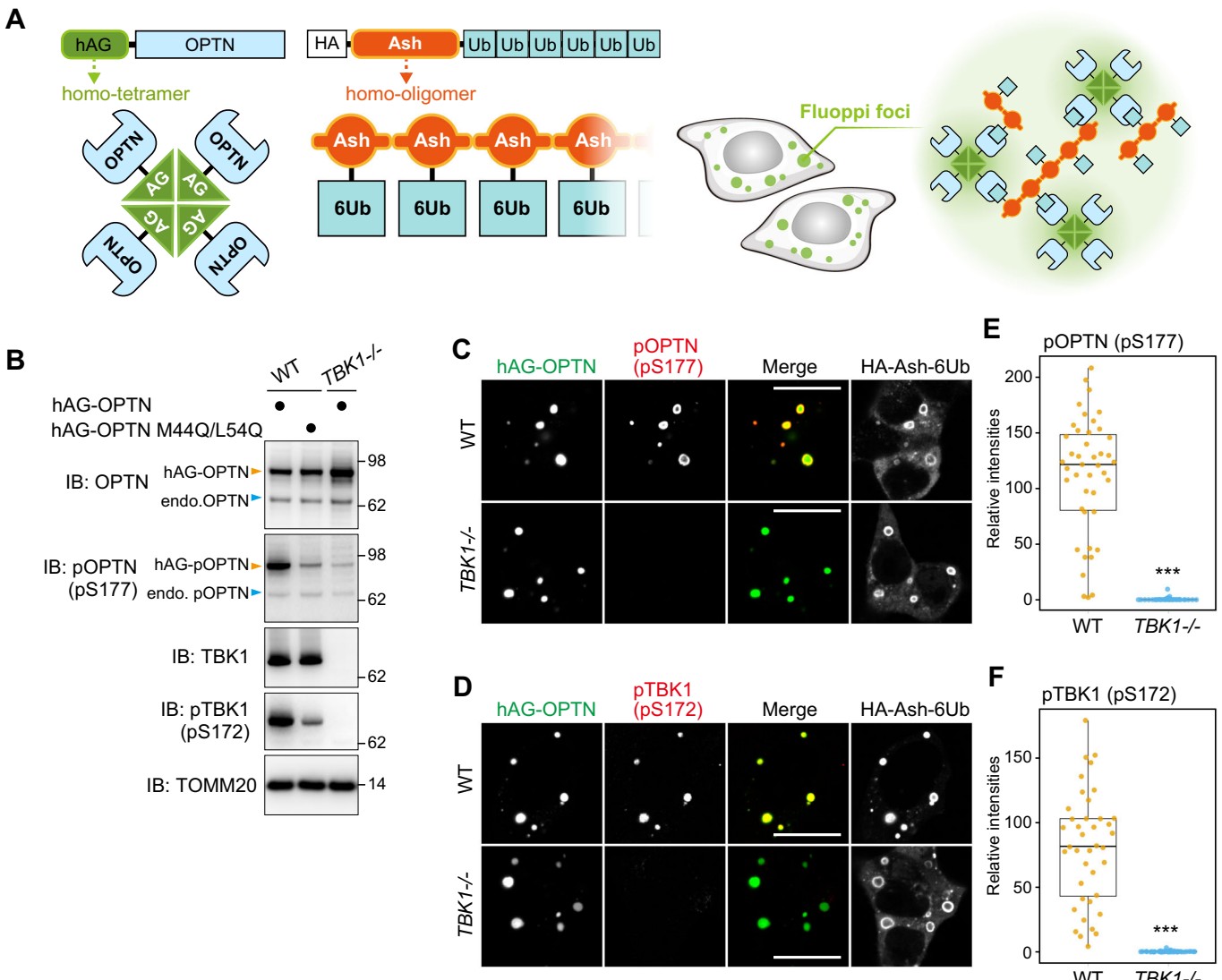

**Figure 4. Artificial OPTN multimerization compensates for the autophagy requirement following TBK1 activation.**

(A) Schematic representation of fluoppi foci. (B) The indicated hAG-OPTN constructs were expressed with HA-Ash-6Ub in WT and *TBK1-/-* HCT116 cells. Total cell lysates were analyzed by immunoblotting (IB). The light blue and orange arrowheads indicate endogenous and hAG-tagged OPTN bands, respectively. (C,D) hAG-OPTN and HA-Ash-6Ub were expressed in WT and *TBK1-/-* HCT116 cells. The cells were immunostained with antibodies against HA, pOPTN(pS177) for (C), and pTBK1(pS172) for (D). Bars, 10 μm. (E,F) The intensities of pOPTN(pS177) and pTBK1(pS172) in the fluoppi foci in (C) and (D), respectively, were quantified and shown as a box plot, with the box indicating the inter-quarter range (IQR), the whiskers showing the range of values that are within 1.5 × IQR and a horizontal line indicating the median. Error bars represent mean ± s.d. with 50 fluoppi foci quantified in two independent experiments. ***P < 0.001 by two-tailed Student's *t*-test. Source data are available online for this figure.

activated occurs regardless of TBK1. However, the levels of phosphorylated TBK1 (S172) and OPTN (S177) differed depending on the TBK1 mutant. In addition to S172A, the D135N, G217R, and M559R mutations were almost devoid of TBK1 phosphorylation (Fig. 5A). While the R47H and R357Q mutations reduced TBK1 phosphorylation when compared to WT, roughly 50% of the TBK1 R357Q mutant, which is only about half as abundant as WT TBK, was phosphorylated during Parkin-mediated mitophagy (Fig. 5B middle graph). The R357Q mutant, however, efficiently phosphorylated S177 in OPTN when compared to the other TBK1 mutants. Normalizing TBK1 WT phosphorylation of OPTN during Parkin-mediated mitophagy to 100% yields an ~60%

phosphorylation rate by R357Q. In contrast, the level of phosphorylated OPTN in cells expressing the other TBK1 mutants was similar to *TBK1-/-* cells (Fig. 5B right graph). Thus, monomeric TBK1 is unstable, whereas the R357Q mutant retains the ability to phosphorylate OPTN via TBK1 activation.

We next tested the effect of TBK1 mutations on mitochondrial degradation using Keima, a fluorescent protein with pH-dependent excitation spectra (Katayama et al, 2011), as a reporter of mitophagy-induced delivery of mitochondria to the acidic lysosome lumen (Lazarou et al, 2015; Yamano et al, 2020). The fluorescence profile of cells expressing mitochondrial-targeted Keima was assessed via FACS after inducing mitophagy with antimycin A

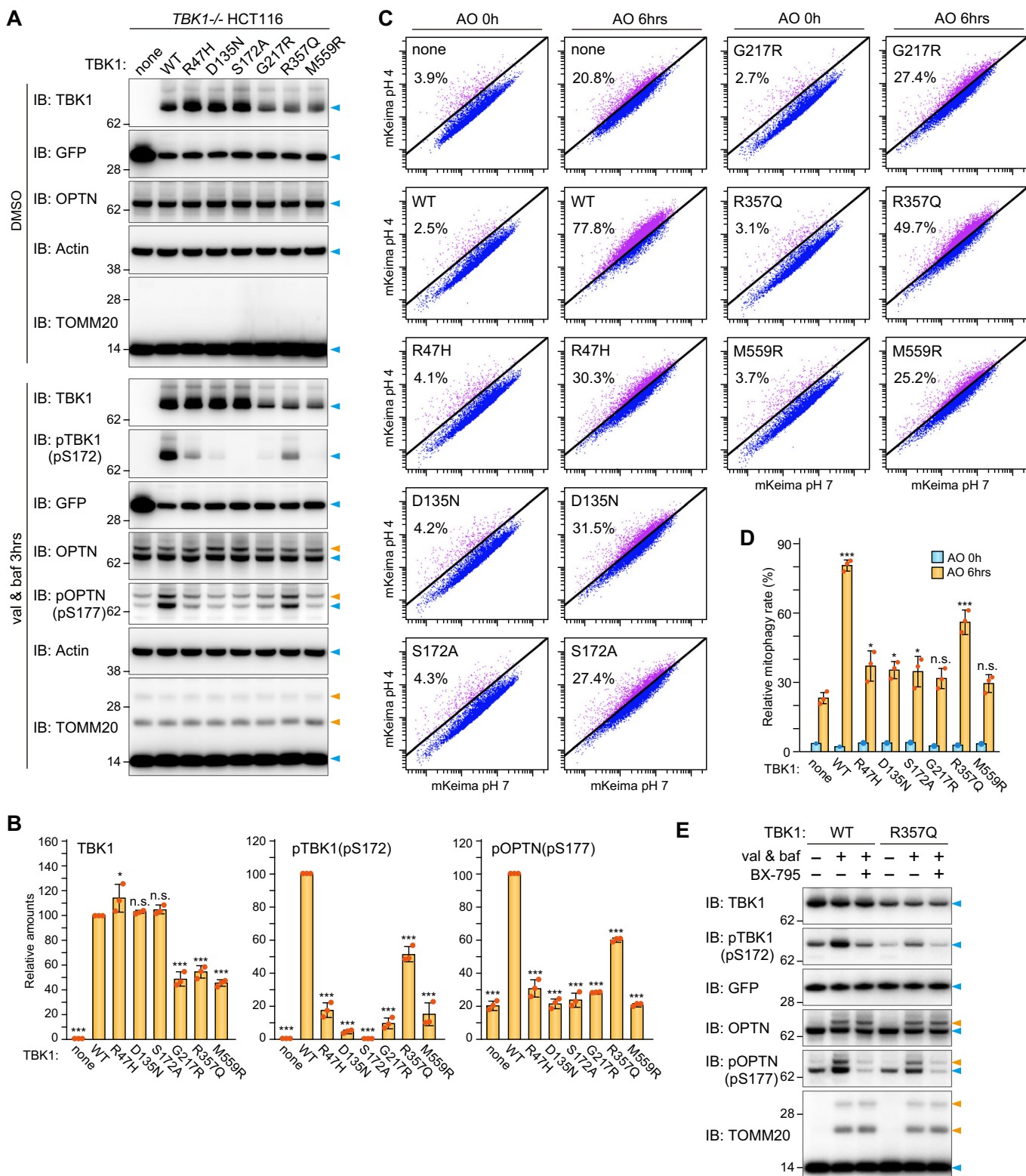

and oligomycin (AO) treatment (Fig. 5C). Mitophagy was evident in 78% of *TBK1-/-* HCT116 cells expressing YFP-P2A-TBK1 WT after 6 h with AO (Fig. 5C,D). In contrast, 20% of the *TBK1-/-* cells and 30–40% of the cells with the R47H, D135N, S172A, G217R, or M559R TBK1 mutants were mitophagy positive. Highest

mitophagy induction among the mutants, however, was observed with in cells expressing the R357Q mutant (Fig. 5C,D). These results strongly suggest that the efficiency of Parkin-mediated mitochondrial degradation is correlated with the degree of S172 phosphorylation in TBK1.

Figure 5.   Impact of ALS-associated TBK1 mutations on Parkin-mediated mitophagy.

(A) *TBK1-/-* HCT116 cells stably expressing Parkin, mt-Keima, and YFP alone (indicated as "none") or YFP-P2A-TBK1 (WT or the indicated mutants) were treated with DMSO or valinomycin (val) and bafilomycin (baf) for 3 h and then analyzed by immunoblotting (IB). (B) The levels of TBK1, pTBK1(pS172), and pOPTN(pS177) after 3 h of val and baf treatment in (A) were quantified. The protein levels in cells expressing WT TBK1 were set to 100. Error bars represent mean ± s.d. of three independent experiments. (C) Cells in (A) were treated with antimycin A and oligomycin (AO) for 0 or 6 h and analyzed by FACS. Representative FACS data with the percentage of cells exhibiting lysosomal positive mt-Keima are shown. (D) Mitophagy rate (percentage of cells having lysosomal positive Keima) in (C). Error bars represent mean ± s.d. of three independent experiments. (E) *TBK1-/-* HCT116 cells stably expressing Parkin and YFP-P2A-TBK1 (WT or the R357Q mutant) were treated with val and baf for 3 h with or without 2 µM BX-795. Total cell lysates were analyzed by IB. Data information: n.s. not significant, *$P < 0.05$, **$P < 0.01$, ***$P < 0.001$ by two-tailed Dunnett's test (B,D). The light blue and orange arrowheads indicate unmodified and ubiquitinated bands, respectively (A,E). Source data are available online for this figure.

There is a possibility that ULK1/2 may phosphorylate OPTN at S177 when TBK1 is mutated (Harding et al, 2021). To determine if this phosphorylation event is mediated by TBK1 R357Q, we induced mitophagy in the presence of BX-795, a commonly used TBK1 kinase inhibitor. In *TBK1-/-* HCT116 cells expressing WT TBK1 in the presence of BX-795, the phosphorylation signals for OPTN S177 and TBK1 S172 were comparable to basal levels, suggesting complete inhibition (Fig. 5E). Similar results were observed in *TBK1-/-* cells expressing the TBK1 R357Q mutant (Fig. 5E). While we cannot exclude the possibility that BX-795 also inhibits other kinase activities, the results suggest that during mitophagy the phosphorylations of TBK1 S172 and OPTN are primarily driven by TBK1, even when TBK1 is monomeric.

## Creation of monobodies against OPTN

The results described above suggest that OPTN forms a contact site between isolation membranes and damaged mitochondria during Parkin-mediated mitophagy that supports TBK1 autophosphorylation. To determine if OPTN accumulation at the mitophagic contact site is required for TBK1 activation, we next sought to create monobodies that bind OPTN and physically prevent it from accumulating. Monobodies are non-immunogloblin-based proteins derived from the 10th type III domain of human fibronectin that serve as backbone proteins that are sufficiently small for expression in both bacterial and mammalian cells (Koide et al, 1998). We previously developed a high-speed in vitro selection method termed TRAP (transcription-translation coupled with association of puromycin-linker) display for generating a monobody against a target protein (Kondo et al, 2020). As shown in Fig. 6A, monobody/mRNA complexes were synthesized from DNA pools in which the coding region for the BC-FG or CD-FG loops of a monobody were randomized. After reverse transcription, monobody/cDNA complexes were incubated with biotinylated recombinant OPTN fragments and treated with streptavidin-coated magnetic beads, such that complexes with affinities for OPTN were selected. The amplified monobody DNAs were re-introduced into the TRAP system to yield a monobody library. We targeted two different N-terminal OPTN regions (26–196 aa and 133–196 aa), which were conjugated to biotin using an in vitro sortase reaction (Fig. 6A and see Methods). After 7–8 rounds of selection, the recovered monobody cDNAs were sequenced (Supplementary Table 1). We selected 12 enriched clones from the pool for further analyses. YFP-fused monobodies were initially expressed in *NDP52* KO and *OPTN* KO HeLa cells, and a Keima-based mitophagy assay was performed. A previous study showed that both NDP52 and OPTN function as essential mitophagy adaptors and that either is sufficient for mitophagy (Lazarou et al, 2015), which means that

*NDP52* KO or *OPTN* KO HeLa cells can drive Parkin-mediated mitophagy. Indeed, 79.3% of *NDP52* KO HeLa cells, in which OPTN is the lone mitophagy adaptor, were mitophagy-positive after 5 h with AO (Fig. 6B,C). Interestingly, the expression of several OPTN monobody clones (clones #2, 3, 4, 5, 6, 8, 9) in *NDP52* KO cells inhibited mitophagy (Fig. 6B,C). The inhibitory effects were specific to monobody-OPTN interactions as no mitophagy defects were observed in *OPTN* KO HeLa cells, which have NDP52 as the lone essential mitophagy adaptor (Fig. 6D). To test if the monobodies can bind to OPTN in cells, *NDP52* KO HeLa cells stably expressing YFP alone or a YFP-monobody were co-immunoprecipitated. We confirmed that all of the YFP-monobodies tested were expressed (Fig. 6E) and, with the exception of clones #1 and 11, that all interacted with endogenous OPTN (Fig. 6F). Next, we characterized the OPTN monobodies in vitro. To verify the region of OPTN that interacts with the monobodies, the recombinant monobodies were pulled down using three different GST-OPTN constructs corresponding to OPTN aa 26–196, 26–119, or 133–196. All of the monobodies bound aa 26–196 of OPTN and all but clone #7 bound aa 26–119 (Fig. 6G). The N-terminal region of OPTN (26–119 aa) interacts with the C-terminal region (677–729 aa) of TBK1 (Li et al, 2016). GST-tagged TBK1 (677–729 aa) was incubated with EGFP or OPTN (26–196)-EGFP in the presence of the recombinant monobodies and the GST-TBK chimera was pulled down using glutathione sepharose (Fig. 6H). For three clones (#3, 6, and 8), both OPTN (26–196)-EGFP and the monobodies were present in the eluant, whereas no monobodies and reduced levels of OPTN (26–196)-EGFP were in the eluant for clones #4 and #9. These results indicate that the first group of clones (i.e., #3, 6, and 8) can form a ternary complex with OPTN and TBK1 but that clones #4 and #9 competitively prevent TBK1 from binding OPTN (Fig. 6H,I).

## Monobodies against OPTN inhibit both the assembly and accumulation of OPTN at the mitophagic contact site

We selected monobody clones #3 and #4 for further analysis. First, the binding constants for OPTN and the monobodies were assessed using biolayer interferometry (BLI). OPTN (26–196 aa) was immobilized on the streptavidin-coated sensor and various concentrations of the monobodies were applied to determine the kinetic parameters based on global fitting (Appendix Fig. S3). Both clones exhibited nanomolar affinities (clone #3, Kd = 4.56 nM; clone #4, Kd = 6.19 nM) for the OPTN N-terminus, which are much lower (i.e., stronger) than the 2.5 µM reported for TBK1 (677–729 aa) and OPTN (26–119 aa) (Li et al, 2016). Next, we examined how the monobodies block Parkin-mediated mitophagy. For these assays, the two monobodies were stably expressed in *NDP52* KO HeLa cells and the cellular localization of

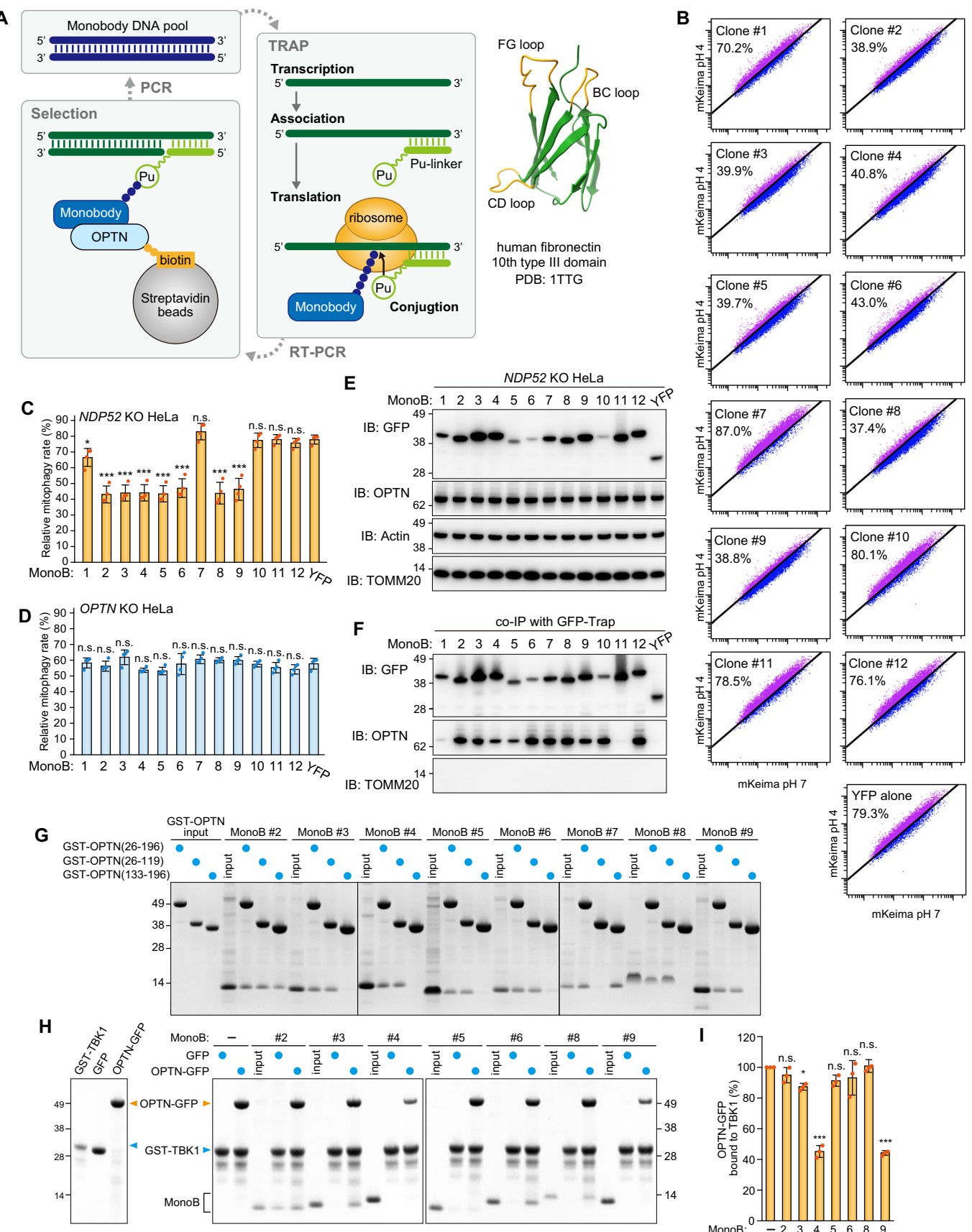

**Figure 6.   Selection and validation of OPTN monobodies.**

(A) Schematic representation of the process to select monobodies against OPTN using TRAP display. The BC, CD, and FG loops in the monobody structure (PDB: 1TTG) are shown in orange. (B) *NDP52* KO HeLa cells expressing Parkin, mt-Keima, and YFP or YFP-tagged monobodies (MonoB) were treated with antimycin A and oligomycin (AO) for 5 h and analyzed by FACS. Representative FACS data with the percentage of cells exhibiting lysosomal positive mt-Keima are shown. (C) Mitophagy rate (percentage of cells having lysosomal positive Keima) in (B) were quantified. Error bars represent mean ± s.d. of four independent experiments. (D) Mitophagy rate (percentage of cells having lysosomal positive Keima) using *OPTN* KO HeLa cells were quantified. Error bars represent mean ± s.d. of four independent experiments. (E) *NDP52* KO HeLa cells expressing Parkin and YFP or YFP-tagged MonoB were analyzed by immunoblotting (IB). (F) The cells in (E) were co-immunoprecipitated (co-IP) using GFP-Trap. The bound fractions were analyzed by IB. (G) Recombinant GST-OPTN (26–196 aa, 26–119 aa, or 133–196 aa) were incubated with recombinant MonoB. The glutathione sepharose bound fractions were analyzed by CBB staining. (H) Recombinant GST-TBK1 (677–729 aa) and MonoB were incubated with either recombinant GFP or OPTN (26–196 aa)-GFP. GST-TBK1 (677–729 aa) was pulled down with glutathione sepharose and the bound fractions were analyzed by CBB staining. (I) The levels of OPTN (26–196 aa)-GFP pulled down with GST-TBK1 (677–729 aa) in (H) were quantified. The level of OPTN (26–196 aa)-GFP pulled down without MonoB was set to 100. Error bars represent mean ± s.d. of three independent experiments. Data information: n.s. not significant, *P < 0.05, ***P < 0.001 by two-tailed Dunnett's test (C,D,I). Source data are available online for this figure.

endogenous OPTN during Parkin-mediated mitophagy was determined (Fig. 7A). As a control, we also examined the localization of OPTN in *NDP52* KO cells expressing GFP. In these cells, OPTN formed dots and sphere-like structures on damaged mitochondria. In sharp contrast, this OPTN distribution was not observed in cells expressing the GFP-tagged monobodies; however, mitochondrial recruitment of OPTN was still observed (Fig. 7A). Furthermore, immunoblots showed that expression of the monobodies abolished both autophosphorylation of S172 in TBK1 and S177 phosphorylation in OPTN (Fig. 7B,C). Mitophagy-induced ubiquitination of OPTN was not disrupted by the presence or absence of the monobodies (Fig. 7B), which is consistent with mitochondrial recruitment of OPTN regardless of monobody expression.

## Discussion

TBK1 is a key regulatory kinase in the cell signaling pathways underlying innate immune responses and the elimination of invader pathogens and damaged mitochondria (Alam et al, 2021). To modulate this signaling, TBK1 can bind to different adaptor proteins. These interactions facilitate local clustering of TBK1 and allow the interdimer kinase domains of TBK1 to activate via trans-autophosphorylation (Helgason et al, 2013). TBK1 is a multimeric domain protein consisting of a kinase domain, a ubiquitin-like domain, a scaffold dimerization domain, and C-terminal domain that can bind various adaptor proteins. TBK1 forms a rigid dimer in which the two kinase catalytic sites are oriented away from one another. Thus, TBK1 autophosphorylation requires formation of homogenous TBK1 dimers. In the STING pathway, the binding of cGAMP by STING induces the oligomerization of STING on post-Golgi membranes (Cai et al, 2014; Shang et al, 2019). Subsequent higher-order STING oligomers arranged in a linear manner and a C-terminal STING segment bind TBK1 dimers without inducing structural changes in TBK1. Therefore, STING oligomers provide a platform for TBK1 oligomerization, which induces trans-autophosphorylation of adjacent TBK1 proteins. Thus, in the STING pathway, STING oligomerization directly induces TBK1 activation.

Previous findings showed that TBK1 promotes Parkin-mediated mitophagy by phosphorylating NDP52 and OPTN to increase their affinity for ubiquitin, ATG8s, and FIP200 (Heo et al, 2015; Richter et al, 2016; Wild et al, 2011), and that OPTN recruits TBK1 onto damaged mitochondria (Moore and Holzbaur, 2016). Since TBK1 autophosphorylation is a prerequisite for TBK1 activation, OPTN

recruitment to damaged mitochondria was thought to be sufficient for inducing TBK1 autophosphorylation. However, in this study, we showed that OPTN recruitment to damaged mitochondria in the absence of the autophagy machinery reduced TBK1 autophosphorylation (Figs. 2 and EV3). Although OPTN forms a dimer through its coiled-coil domains, OPTN is unlikely to oligomerize on its own even when bound to ubiquitin (Nakazawa et al, 2016). In this study, we revealed that TBK1 autophosphorylation requires OPTN for interactions with both the autophagy machinery and the ubiquitin-coated cargo. Previously, it was thought that TBK1 activates via autophosphorylation before phosphorylating OPTN to recruit autophagic membranes to mitochondria. However, our results show that formation of the isolation membrane itself is involved in TBK1 autophosphorylation (Fig. 2). We show that OPTN interacts with both autophagy machinery and ubiquitin to form contact sites that accumulate TBK1. Further, our results show that autophagic membrane formation affects TBK1 autophosphorylation, which must occur for TBK1 activation, thereby negating the unidirectional model in which TBK1 activation results in the phosphorylation of OPTN S177 and the subsequent recruitment of autophagic membranes to mitochondria. Signaling cascades for most kinases are unidirectional (i.e., the transfer of signals from upstream components to those downstream). In response to PINK1/Parkin-mediated mitophagy, the initial TBK1 autophosphorylation step requires downstream interactions between OPTN and the autophagy components, suggesting a positive feedback loop. TBK1 phosphorylation of the OPTN UBAN domain (e.g. S473) promotes tight association of OPTN with ubiquitin-coated mitochondria, whereas TBK1 phosphorylation of S177 in the LIR motif induces translocation of mitochondria-associated OPTN to the autophagosome formation site. This accumulation of OPTN at a contact site between the autophagy machinery and ubiquitin-coated mitochondria provides a platform for TBK1 hetero-autophosphorylation by adjacent TBK1 at the contact site (Fig. 7D). Activated TBK1 then phosphorylates other OPTN-TBK1 pairs, which are newly recruited to damaged mitochondria, and the next phosphorylation cycle proceeds. Thus, OPTN-mediated TBK1 phosphorylation and TBK1-mediated OPTN phosphorylation appear to form a self-propagating positive feedback loop, that leads to the elongation of isolation membranes on damaged mitochondria. Our OPTN monobody data also support this model. Although monobody clone #3 formed a ternary complex with OPTN and TBK1 (Fig. 6H), it prevented OPTN accumulation, which abolished formation of the mitophagic contact site and subsequent TBK1 autophosphorylation (Fig. 7).

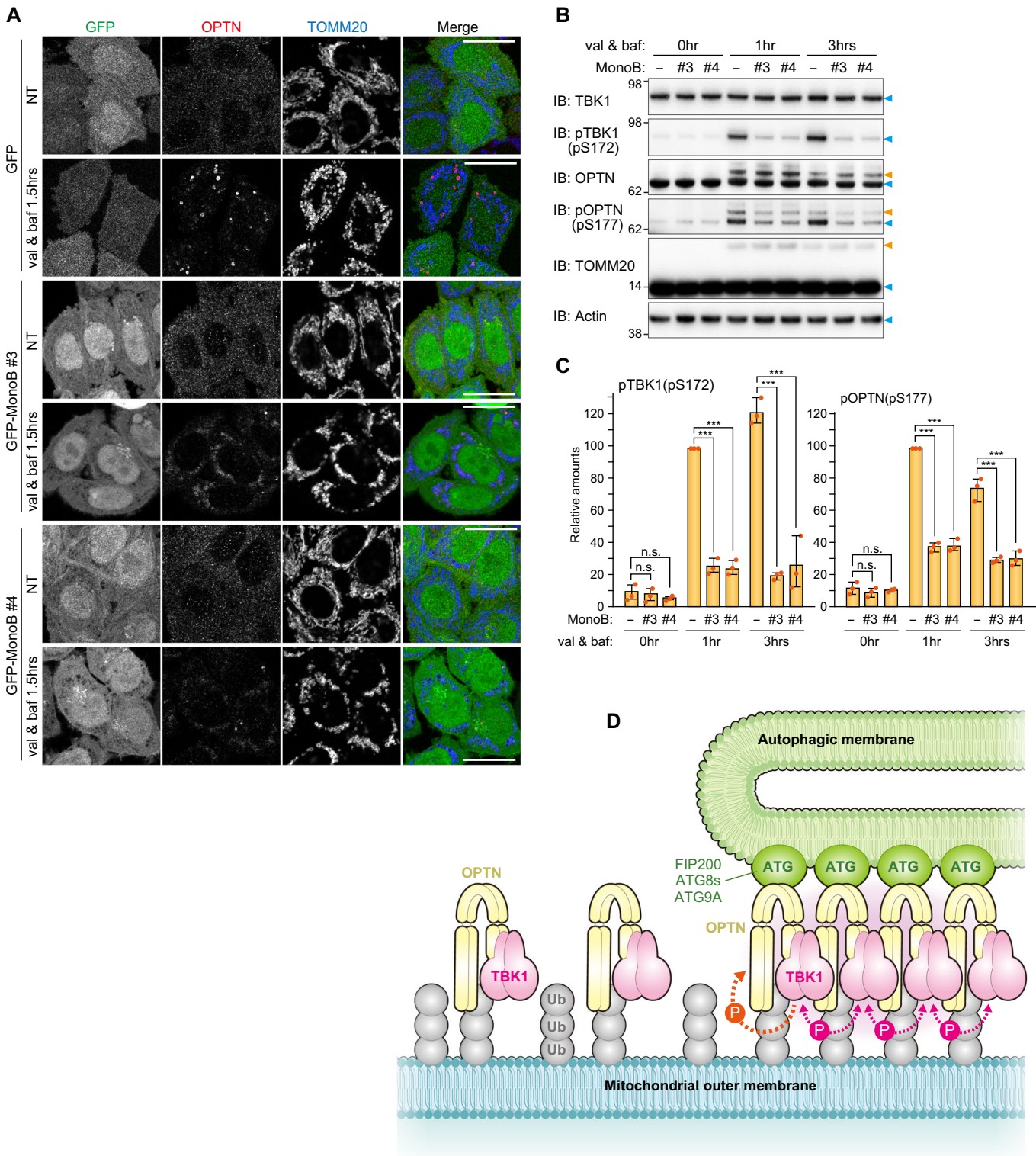

**Figure 7. OPTN monobodies inhibit OPTN assembly at the mitophagy-contact sites.**

(A) *NDP52* KO HeLa cells expressing Parkin, GFP alone, or GFP-tagged monobodies (MonoB) were treated with or without valinomycin (val) and bafilomycin (baf) for 1.5 h. The cells were immunostained with anti-OPTN and TOMM20 antibodies. Bars, 10 μm. (B) The cells in (A) were analyzed by immunoblotting (IB). The light blue and orange arrowheads indicate unmodified and ubiquitinated protein bands, respectively. (C) The levels of pTBK1 and pOPTN in (B) were quantified. The protein levels in cells without MonoB after 1 h val and baf treatment were set to 100. Error bars represent mean ± s.d. of three independent experiments. n.s. not significant, ***P < 0.001 by two-tailed Dunnett's test. (D) Proposed model of TBK1 activation during PINK1/Parkin-mediated mitophagy. Source data are available online for this figure.

Our findings led us to question if freely diffusing OPTN on the outer mitochondrial membrane could activate TBK1 without autophagy machinery present. The OPTN UBAN domain binds ubiquitin chains composed of two ubiquitin molecules (Nakazawa et al, 2016), and during Parkin-mediated mitophagy only shorter length poly-ubiquitin chains are generated on the mitochondrial surface (Swatek et al, 2019). Based on these studies, it is apparent that it is difficult for multiple OPTN molecules to bind single ubiquitin chains. However, we cannot rule out the possibility that TBK1 autophosphorylation does not occur on mitochondria in the absence of autophagy machinery. While full activation of TBK1 requires OPTN association with the isolation membrane, initial TBK autophosphorylation at the onset of mitophagy might proceed based only on OPTN-ubiquitin interactions.

Some of our OPTN monobodies (such as clones #4 and #9) directly disturbed the OPTN-TBK1 interactions. Interestingly, an OPTN E50K mutation was identified in familial primary open-angle glaucoma (Rezaie et al, 2002) that promotes stronger interactions with TBK1 (Li et al, 2016), which in turn triggers an accumulation of insoluble OPTN and constriction of the Golgi body (Minegishi et al, 2013). Our monobodies could potentially be used to suppress the disease-related phenotype by weakening the interaction between OPTN and TBK1. This hypothesis, however, will need to be tested in future studies.

TBK1 mediates phosphorylation of various Ser/Thr sites on at least four autophagy adaptors, OPTN, NDP52, TAX1BP1, and p62 (Richter et al, 2016). Although phosphorylated S177 and S473 in OPTN enhance interactions with LC3 and K63-linked ubiquitin chains, respectively, other sites, including S262, appear to be randomly phosphorylated. Furthermore, p62, which is recruited to mitochondria but does not accumulate at the mitophagic contact site (Yamano et al, 2020), was also highly phosphorylated by TBK1 during mitophagy. During manuscript preparation, Nguyen et al, reported that direct interactions between TBK1 and the PI3K complex facilitate the initiation of mitophagy by OPTN (Nguyen et al, 2023). Our results support this finding as recruitment of OPTN, but not NDP52, to the autophagosome formation site strictly depends on TBK1 (Fig. 1K,L). Furthermore, that study showed that OPTN-mediated mitophagy does not require the ULK1/2 kinases and that TBK1 compensates for them. Based on our findings, and those of other researchers, TBK1 does not appear to have strict substrate specificity requirements. Rather, the most critical aspect of TBK1 is the spatial location of activation (i.e., at the autophagosome formation site on mitochondria). Since OPTN-TBK1 localizes to the mitophagic contact site where autophagy core components also accumulate, it seems to follow that activated TBK1 would be able to non-discriminately phosphorylate autophagy core components in close proximity.

# Methods

Reagents including cell lines, antibodies, and plasmid DNAs including siRNA used in this study are listed in Appendix Tables S1–S3.

## Plasmid construction

Human codon-optimized *TBK1* was synthesized by eurofins genomics and was inserted into the EcoRI site of pMXs-puro_EYFP-P2A-EcoRI to make pMXs-puro_YFP-P2A-hOpt-TBK1. Mutations were introduced by PCR-based DNA mutagenesis.

## Cell culture and transfection

HeLa and HEK293T cells were cultured in DMEM supplemented with 10% (v/v) FBS, 1 mM sodium pyruvate, nonessential amino acids, and PSG. HCT116 cells were cultured in McCoy's 5A medium supplemented with 10% (v/v) FBS, nonessential amino acids, and 2 mM GlutaMax. The Penta KO HeLa (Lazarou et al, 2015), *NDP52* KO HeLa (Lazarou et al, 2015), *OPTN* KO HeLa (Lazarou et al, 2015), *FIP200* KO HeLa (Vargas et al, 2019), *ATG9A* KO HeLa (Nezich et al, 2015), *ATG5* KO HeLa (Nezich et al, 2015), and *TBK1-/-* HCT116 (Yamano et al, 2020) cell lines were described previously. Cell lines used in this study were authenticated and tested for mycoplasma contamination.

The HeLa AAVS-Parkin cell line was established as follows. A donor Parkin plasmid was constructed using a pZDonor-AAVS Puromycin Vector Kit. The resultant donor plasmid was transfected with AAVS TALEN plasmids into HeLa cells using FuGENE6. The cells were grown in the presence of 1 µg/ml puromycin and single clones were isolated into 48-well plates. Insertion of the *PARKIN* gene into the AAVS locus was confirmed by isolation of genomic DNA followed by PCR. Parkin expression and mitochondrial recruitment were confirmed by immunostaining valinomycin-treated cells with an anti-Parkin antibody. The HeLa AAVS-Parkin cell line was used for Figs. 1A–D,I,J,2L,EV1–EV3D–F, Appendix S1,S2.

Stable cell lines were made by recombinant retrovirus infection (Yamano et al, 2020). FuGENE6 and FuGENE HD reagents were used for plasmid transfection according to the manufacturer's instructions. RNAiMAX was used for siRNA transfection.

Final concentrations of reagents used were: 10 µM valinomycin, 10 µM oligomycin, 4 µM antimycin A, 100 nM bafilomycin A1, 1 µM epoxomicin, 10 µM MG132, and 100 nM concanamycin A. To block apoptotic cell death in response to Parkin-mediated ubiquitination, 5 µM Q-VD-OPH was added.

## CRISPR/Cas9-edited gene knockout

*TBK1* KO, *FIP200/TBK1* DKO, and *ATG9A/TBK1* DKO HeLa cell lines were established via CRISPR/Cas9-based genome editing. The three gRNA target sequences (5′-GAC CCT TTG AAG GGC CTC GTA GG-3′, 5′-ATT CCT ACG AGG CCC TTC AAA GG-3′ and 5′-GGC CCT TCA AAG GGT CTA AAT GG-3′) for *TBK1* exon 6 were designed using CRISPOR (http://crispor.tefor.net/). Each of the oligonucleotide pairs were annealed and introduced into the BpiI site of the PX459 vector to yield PX459-TBK1-ex6-1, PX459-TBK1-ex6-2, and PX459-TBK1-ex6-3. The plasmids were transfected into WT HeLa, *FIP200* KO HeLa, and *ATG9A* KO HeLa cell lines. Puromycin-resistant cells were seeded into 96 well plates, and single clones were analyzed by immunoblotting to confirm *TBK1* knockout.

## Immunoblotting

Cells grown in 6-well plates were washed twice with Phosphate Buffered Saline (PBS) and solubilized with 2% CHAPS buffer (25 mM HEPES-KOH pH 7.5, 300 mM NaCl, 2% [w/v] CHAPS, cOmplete) on ice for 10 min. To detect phosphorylated proteins, PhosSTOP was

added to the 2% CHAPS buffer. After centrifugation at $12,000 \times g$ for 2 min at 4 °C, the supernatants were collected, and protein concentrations were determined on a DS-11+ spectrophotometer (DeNovix). SDS-PAGE sample buffer with DTT was added to the supernatants, which were then incubated at 42 °C for 5 min. To detect hAG constructs, the samples were boiled at 95 °C for 5 min. Total cell lysates were loaded on NuPAGE 4–12% Bis-Tris gels and electrophoresed using MES running buffer. Proteins were transferred to PVDF membranes that were blocked with 2% (w/v) skim-milk/TBS-T and then incubated with primary and HRP-conjugated secondary antibodies. Proteins were detected using a Western Lighting Plus-ECL Kit on an ImageQuant LAS4000 (GE Healthcare) or a FUSION SOLO S system (VILBER). ImageJ was used to quantify protein bands. Since the same amounts of total cell lysates were loaded for immunoblotting, the phosphorylated forms pTBK1(pS172), pOPTN(pS177), and p-p62(pS403) were quantified against the total protein.

## Immunostaining and immunofluorescence microscopy

Cells grown on glass bottom 35-mm dishes (MatTek) were fixed with 4% PFA solution for 20 min at room temperature, permeabilized with 0.15% (v/v) Triton X-100 in PBS for 20 min, and preincubated with 0.1% (w/v) gelatin in PBS for 30 min. The cells were incubated with primary antibodies diluted in 0.1% (w/v) gelatin for 2 h and Alexa Fluor-conjugated secondary antibodies diluted in 0.1% (w/v) gelatin for 1 h. Microscopy images were captured using inverted LSM710 and LSM780 confocal microscopes (CarlZeiss) with a Plan-Apochromat 63x/1.4 oil DIC lens or an FV3000 (Olympus) with a PlanApo N 60x/1.4 oil objective lens. For image analysis, ZEN microscope software and Photoshop (Adobe) were used.

## In vitro deubiquitinase assay

After mitophagy induction, cells were washed with PBS twice and solubilized with 2% CHAPS buffer (25 mM HEPES-KOH pH 7.5, 150 mM NaCl,1 mM DTT, 1 µM MLN7243) on ice for 10 min. After centrifugation at $12,000 \times g$ for 2 min at 4 °C, the supernatants were collected. Cell lysates (40 µg) were incubated with 2 µg of recombinant human USP2 catalytic domain (Boston Biochem) at 37 °C. The reaction was stopped by the addition of SDS-PAGE sample buffer.

## Phos-tag PAGE

Phos-tag PAGE analysis was described previously (Yamano et al, 2015).

## Selection of monobodies against OPTN

*Escherichia coli* codon-optimized *OPTN* DNA encoding aa 26–196, 26–119, or 133–196 was synthesized by eurofins genomics and then inserted into the BamHI site of pGEX6P1 together with the sortase recognition sequence and a His6-tag sequence to generate PGEX6P1_Opt-OPTN (26–196 aa, 26–119 aa, or 133–196 aa)-Sortase-His6-STOP. The plasmids were introduced into *E. coli* BL21-CodonPlus(DE3)-RIL competent cells, and the transformants were grown in LB medium supplemented with 100 µg/ml ampicillin

and 25 µg/ml chloramphenicol at 37 °C. OPTN-sortase-His6 was expressed at 18 °C overnight by adding 200 µM IPTG. Bacterial cell pellets were resuspended in TBS buffer (50 mM Tris–HCl pH 7.5, 120 mM NaCl) supplemented with lysozyme, DNAse I, DTT, $MgCl_2$, and cOmplete and stored at −20 °C. Frozen cell suspensions were thawed and sonicated (Advanced-Digital Sonifer, Branson) and insoluble proteins removed by centrifugation. The supernatants were mixed with equilibrated glutathione sepharose for 40 min at 4 °C and then loaded onto a column and washed with TBS buffer. The sepharose columns were incubated overnight at 4 °C with Prescission protease. GST-cleaved OPTN-sortase-His proteins were subsequently collected. Biotin-modified OPTN was prepared as follows. OPTN-sortase-His was incubated with $NH_2$-GGG(Lys[Biotin])-$CONH_2$ (eurofins) and His-tagged Sortase 5Y (a kind gift from Dr. Yasushi Saeki) in TBSC buffer (50 mM Tris–HCl pH 7.5, 120 mM NaCl, 5 mM $CaCl_2$) for 1.5 h at 37 °C. Unmodified OPTN-sortase-His and His-tagged Sortase 5Y were removed by TALON resin and GGGK-biotin was removed using PD Miditrap G-25.

The monobody mRNA libraries were prepared using a similar procedure as reported before (Kondo et al, 2020) and the detailed procedure will be reported elsewhere. For the first round of selection, 1 µM mRNA library/HEX-mPuL was added to a reconstituted translation system, and the reaction mixture (500 µL) was incubated at 37 °C for 10 min. After the reaction, 41.7 µL of 200 mM EDTA (pH 8.0) was added to the translation mixture. Reverse transcription mixture (144.3 µL; 224 mM Tris–HCl, pH 8.4, 336 mM KCl, 96 mM $MgCl_2$, 22 mM DTT, 2.3 mM each dNTP, 6.9 µM MoS-G5R-T.R28 [5′-ACT ATC GGC CTC CTC CTC CAC CTT GAC T-3′] primer, and 5.5 µM MMLV) was added to the translation mixture, and the resulting solution was incubated at 42 °C for 15 min. The buffer was changed to HBST buffer (50 mM Hepes-KOH pH 7.5, 300 mM NaCl, and 0.05% [v/v] Tween 20) using Zeba Spin Desalting Columns. To remove the bead binders, the resulting solution was mixed with 100 µL of Dynabeads M-280 streptavidin (Thermo Fisher Scientific) at 25 °C for 10 min. The supernatant was mixed with 28.8 µL of target protein mixture containing 0.5 µM each biotinylated OPTN fragment at a final concentration of 20 nM. The resulting solution was incubated at 25 °C for 10 min. The target proteins were collected by mixing with 41.1 µL of Dynabeads M-280 streptavidin for 1 min. The collected beads were washed twice with 500 µL of the HBST buffer for 1 min, and 690 µL of PCR premix (10 mM Tris–HCl pH 8.4, 50 mM KCl, 0.1% [v/v] Triton X-100, 2 mM $MgCl_2$, and 0.25 mM each dNTP) was added. The beads were heated at 95 °C for 5 min, and the amount of eluted cDNA was quantified by SYBR green-based quantitative PCR using T7SD8M2.F44 (5′-ATA CTA ATA CGA CTC ACT ATA GGA TTA AGG AGG TGA TAT TTA TG-3′) and MoS-RealTime.R20 (5′-AGC ATC CCA GCT GAT CTG AA-3′) as primers. The eluted cDNA was PCR-amplified using T7SD8M2P.F47 (5′-ATA CTA ATA CGA CTC ACT ATA GGA TTA AGG AGG TGA TAT TTA TGC CT-3′), Mos-G5R-T-an21-3.R45 (5′-CCC GCC TCG CGC CCG CCG TCC ACT ATC GGC CTC CTC CTC CAC CTT-3′) and Pfu-S DNA polymerase and purified by phenol/chloroform extraction and isopropanol precipitation. From the following selection, the resulting DNA (1.25 nM final concentration) was added to the TRAP system, and the reaction mixture (10 µL) was incubated at 37 °C for 30 min. The other procedure was similar to

the above description. After the final round of selection, the sequences of the recovered DNA were analyzed using an Ion Torrent instrument (Thermo Fisher Scientific).

DNAs encoding the monobodies were synthesized (eurofin genetics) and subcloned into pMXs-puro_YFP-TEV, pMXs-puro_GFP-TEV, and pET21a(+) vectors.

## Mitophagy assay using mito-Keima and FACS

FACS-based mitophagy measurements were performed as follows. For *TBK1-/-* HCT116 cells, untagged Parkin, YFP-P2A-TBK1, and mt-Keima were stably expressed via retrovirus infections. For *OPTN* KO and *NDP52* KO HeLa cells, untagged Parkin, YFP-monobody, and mt-Keima were stably expressed. Cells were grown in a 6-well plate and treated with 4 μM antimycin A and 10 μM oligomycin (AO) for varying lengths of time. Trypsinized cells were resuspended in PBS containing 2.5% FBS. FACS analysis was performed using a BD LSRFortessa X-20 cell sorter (BD Biosciences) with FACSDiva software. Keima fluorescence was measured as a dual-excitation ratiometric pH system using 405-nm (pH 7) and 561-nm (pH 4) lasers and 610/20-nm emission filters. Each sample measured consisted of 10,000 YFP/Keima double positive cells.

## Preparation of recombinant proteins

Recombinant OPTN monobodies, OPTN (26–196aa)-EGFP, EGFP, and GST-TBK1 (677–729aa) were obtained as follows. pET21a(+)_OPTN MonoS, pET16b_opt-OPTN (26–196aa)-EGFP, pET16b_EGFP and pGEX6P1_opt-TBK1 (677–729aa) were introduced into *E. coli* BL21-CodonPlus(DE3)-RIL competent cells, and the transformants were grown in LB medium supplemented with 100 μg/ml ampicillin and 25 μg/ml chloramphenicol at 37 °C. The E50K mutation was introduced into OPTN (26–196aa) to enhance interactions with TBK1 (Li et al, 2016). His6-tagged monobody, His10-tagged OPTN (26–196aa)-EGFP, His10-tagged EGFP, and GST-TBK1 (677–729aa) were expressed overnight at 20 °C with 100 μM IPTG. Bacterial cell pellets were resuspended in TBS buffer with lysozyme, DNAse I, DTT, MgCl₂, and cOmplete and stored at −20 °C until used. The frozen cell suspension was thawed, sonicated, and insoluble proteins were removed by centrifugation. His-tagged protein supernatants were mixed with equilibrated Ni-NTA, whereas GST-tagged proteins supernatants were mixed with equilibrated glutathione Sepharose. Protein supernatant mixtures were incubated for 30 min at 4 °C and then washed with TBS buffer. His-tagged proteins were eluted stepwise with imidazole. GST-tagged proteins were eluted with 20 mM glutathione. The elution buffers were exchanged for 50 mM Tris–HCl pH 7.5, 120 mM NaCl, 1 mM TCEP, 10%(w/v) glycerol, and the proteins were stored at −80 °C.

## Pull down assay using recombinant proteins

Recombinant proteins GST-TBK1 (677–729 aa), OPTN (26–196 aa)-EGFP, EGFP alone, and monobodies were incubated as 4.5 μM aliquots with equilibrated glutathione Sepharose in 100 μL binding buffer (50 mM Tris–HCl pH 7.5, 120 mM NaCl, 1 mM TCEP, 10% [w/v] glycerol) for 30 min at room temperature. The resin was washed three times with 1 mL of binding buffer containing 0.1% (v/v) Triton X-100 and bound proteins were eluted with SDS-PAGE sample buffer.

## Kd determination

Protein affinities were determined for biotinylated OPTN (26–196 aa) immobilized on a streptavidin biosensor (ForteBio) using the Octet system (ForteBio) according to the manufacturer's instructions. The analyte monobody was initially dissolved in 50 mM Tris–HCl pH 7.5, 120 mM NaCl, 1 mM TCEP, and 200 mM imidazole and then was exchanged to buffer D (50 mM Hepes-KOH, pH 7.5, 300 mM NaCl, 0.05% [v/v] Tween 20, and 0.1% [w/v] PEG6000) using Zeba Spin Desalting Columns. The protein concentration was measured via A280 according to the molar extinction coefficient estimated from the amino acid composition. The binding assay was performed at 30 °C in buffer D. The binding assay steps consisted of equilibration for 150 s, association for 600 s, and dissociation for 600 s.

## Coimmunoprecipitation using a GFP-Trap

*NDP52* KO HeLa stably expressing YFP alone or YFP-tagged monobodies grown on 35-mm dishes were washed with PBS and solubilized with 0.2% TX-100 buffer (50 mM Tris–HCl pH 7.5, 150 mM NaCl, 0.2% (v/v) Triton X-100, cOmplete) on ice for 20 min. After centrifugation (16,000 × g for 10 min at 4 °C), the supernatants were incubated with equilibrated GFP-Trap agarose for 1 h at 4 °C. The agarose was three times washed with 0.2% TX-100 buffer and the proteins were eluted with SDS-PAGE sample buffer.

## Statistical analysis

Statistical analysis was performed using data obtained from three or more biologically independent experimental replicates. Student's *t*-test was used for comparisons between two groups and Dunnett's test was used for multiple comparisons using GraphPad Prism (n.s., not significant; *$P < 0.05$; **$P < 0.01$; ***$P < 0.001$).

# Data availability

This study includes no data deposited in external repositories.

# Peer review information

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

## Acknowledgements

We thank Drs. Richard J. Youle and Chunxin Wang for the *FIP200* KO, *ATG5* KO, *ATG9A* KO and Penta KO HeLa cells, Dr. Masaaki Komatsu for the rabbit anti-NBR1 antibodies, and Dr. Yasushi Saeki's group for the in vitro sortase reaction. This work was supported by JSPS KAKENHI Grants JP18H05500, JP18K06237, 22H02577 and 23H04923 (to KY); by JSPS KAKENHI Grants JP22J00707ZA, JP22K15045ZA (to WK); JSPS KAKENHI grant JP19H05712, AMED CREST grant JP23gm1410004h0004, Takeda Science Foundation, Nanken-Kyoten, TMDU, and Joint Usage and Joint Research Programs, Institute of Advanced Medical Sciences, Tokushima University (to NM); by JSPS KAKENHI Grants JP22H00419 and Takeda Science Foundation (to KT); by JSPS KAKENHI Grants 19H05287, 21H00278, and JST PRESTO JPMJPR19K6 (to GH); by AMED Grants JP21zf0127004 (to HM).

## Author contributions

**Koji Yamano**: Conceptualization; Supervision; Funding acquisition; Investigation; Methodology; Writing—original draft; Writing—review and editing. **Momoha Sawada**: Validation; Investigation. **Reika Kikuchi**: Validation; Investigation. **Kafu Nagataki**: Investigation. **Waka Kojima**: Funding acquisition; Validation; Investigation. **Ryu Endo**: Investigation. **Hiroki Kinefuchi**: Investigation. **Atsushi Sugihara**: Investigation. **Tomoshige Fujino**: Investigation. **Aiko Watanabe**: Validation; Investigation. **Keiji Tanaka**: Funding acquisition; Writing—original draft; Writing—review and editing. **Gosuke Hayashi**: Conceptualization; Funding acquisition; Investigation; Methodology; Writing—original draft; Writing—review and editing. **Hiroshi Murakami**: Supervision; Funding acquisition; Writing—original draft; Writing—review and editing. **Noriyuki Matsuda**: Supervision; Funding acquisition; Writing—original draft; Writing—review and editing.

## Disclosure and competing interests statement

The authors declare no competing interests.

# Expanded View Figures

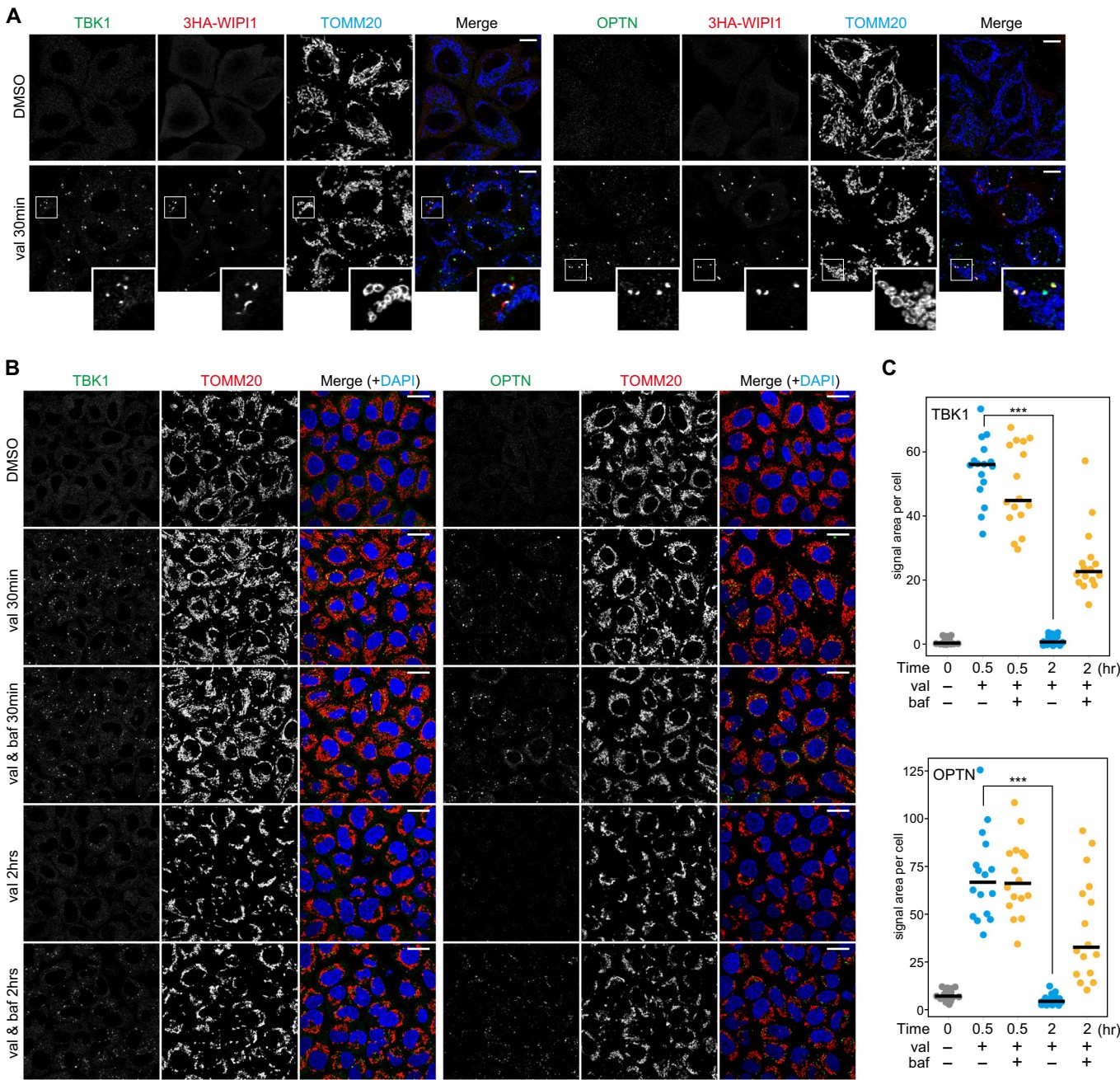

**Figure EV1. TBK1 and OPTN recruitments to damaged mitochondria upon Parkin-mediated mitophagy.**

(A) HeLa cells stably expressing Parkin and 3HA-WIPI1 were treated with DMSO or valinomycin (val) for 30 min and immunostained with the indicated antibodies. Magnified images are also shown for cells treated with val. Bars, 10 μm. (B) HeLa cells stably expressing Parkin were treated with DMSO or valinomycin (val) with or without bafilomycin for the indicated times and immunostained with the indicated antibodies. Nuclei were stained with DAPI. Bars, 20 μm. (C) The relative areas of TBK1 (upper) and OPTN (lower) foci on mitochondria per cell were quantified. Each dot represents the mean value determined from 18–36 cells, and the horizontal lines indicate the median. ***$P < 0.001$ by two-tailed Student's $t$-test. Source data are available online for this figure.

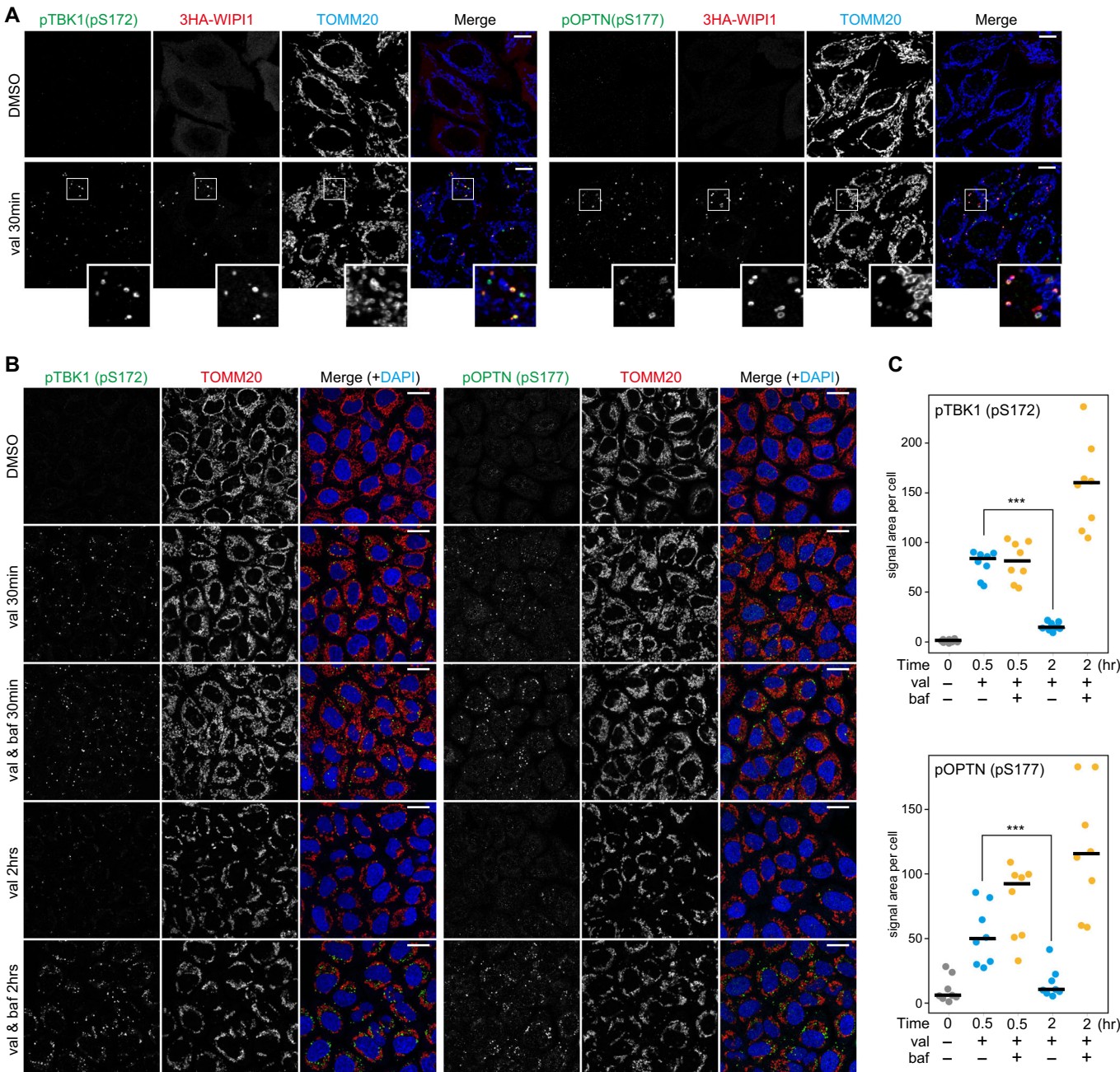

**Figure EV2. phosphorylated TBK1 and phosphorylated OPTN productions on damaged mitochondria upon Parkin-mediated mitophagy.**

(A) HeLa cells stably expressing Parkin and 3HA-WIPI1 were treated with DMSO or valinomycin (val) for 30 min and immunostained with the indicated antibodies. Magnified images are also shown for cells treated with val. Bars, 10 μm. (B) HeLa cells stably expressing Parkin were treated with DMSO or valinomycin (val) with or without bafilomycin for the indicated times and immunostained with the indicated antibodies. Nuclei were stained with DAPI. Bars, 20 μm. (C) The relative areas of pTBK1(pS172) (upper) and pOPTN(pS177) (lower) foci on mitochondria per cell were quantified. Each dot represents the mean value determined from 18–36 cells, and the horizontal lines indicate the median. ***$P < 0.001$ by two-tailed Student's $t$-test. Source data are available online for this figure.

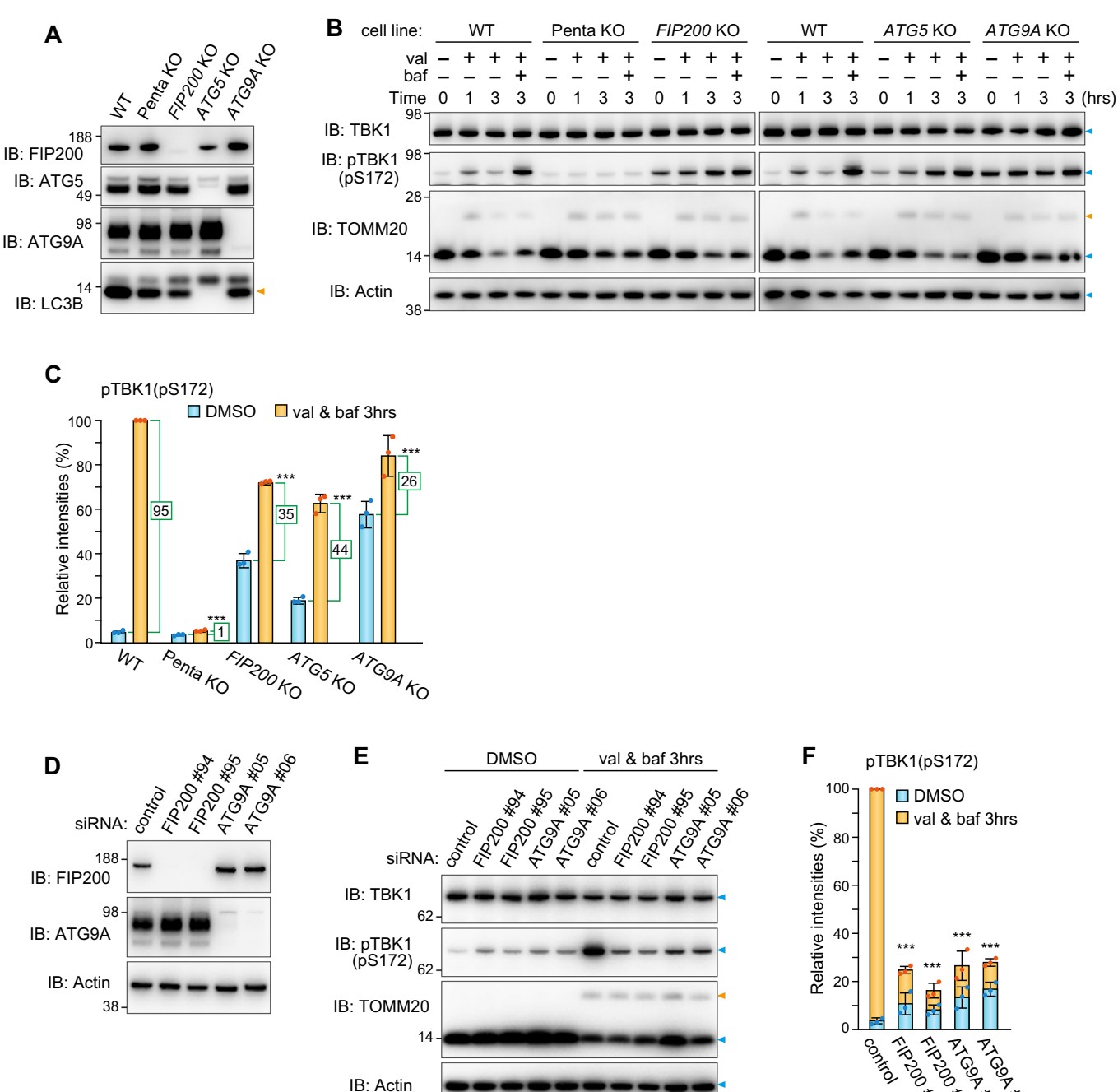

**Figure EV3. TBK1 activation during Parkin-mediated mitophagy requires autophagy core components.**

(A) Depletion of the indicated proteins was confirmed by immunoblotting (IB). The orange arrowhead denotes lipidated forms of LC3B. (B) The indicated cells stably expressing Parkin were treated with val and baf for the indicated times. Total cell lysates were analyzed by IB. (C) The levels of pTBK1(pS172) in (B) were quantified. The level of pTBK1 in WT cells treated with val and baf for 3 h was set to 100. Error bars represent mean ± s.d. of three independent experiments. Signal intensities for pTBK1(pS172) specifically generated during Parkin-mediated mitophagy were determined by subtracting signals for pTBK1 in DMSO from those following val and baf for 3 h (green lines). The difference scores are indicated in the green boxes. (D) HeLa cells stably expressing Parkin treated with the indicated siRNA were analyzed by IB. (E) HeLa cells stably expressing Parkin pre-treated with the indicated siRNA were treated with val and baf for 3 h and analyzed by IB. (F) The levels of pTBK1(pS172) in (E) were quantified. The pTBK1 level in control cells was set to 100%. Error bars represent mean ± s.d. of three independent experiments. Data information: ***$P < 0.001$ by two-tailed Dunnett's test (C,F). The light blue and orange arrowheads indicate unmodified and ubiquitinated bands, respectively (B,E). Source data are available online for this figure.

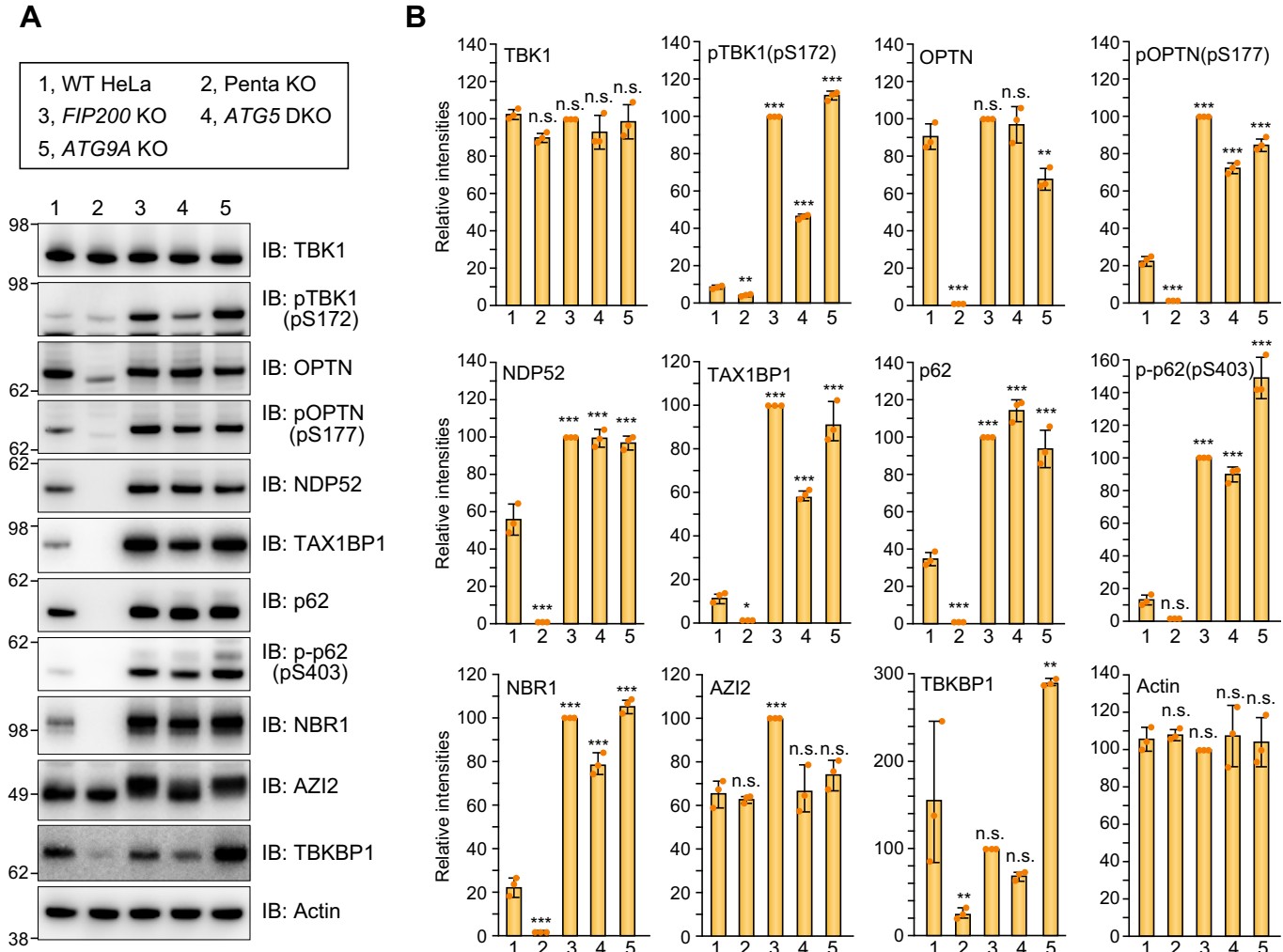

**Figure EV4. Accumulations of autophagy adaptors by loss of autophagy core components.**

(A) The indicated proteins in WT, Penta KO, *FIP200* KO, *ATG5* KO, and *ATG9A* KO HeLa cells were analyzed by immunoblotting. (B) Protein levels in (A) were quantified. Protein levels in *FIP200* KO cells were set to 100. Error bars represent mean ± s.d. of three independent experiments. n.s. not significant, *$P < 0.05$, **$P < 0.01$, ***$P < 0.001$ by two-tailed Dunnett's test. Source data are available online for this figure.

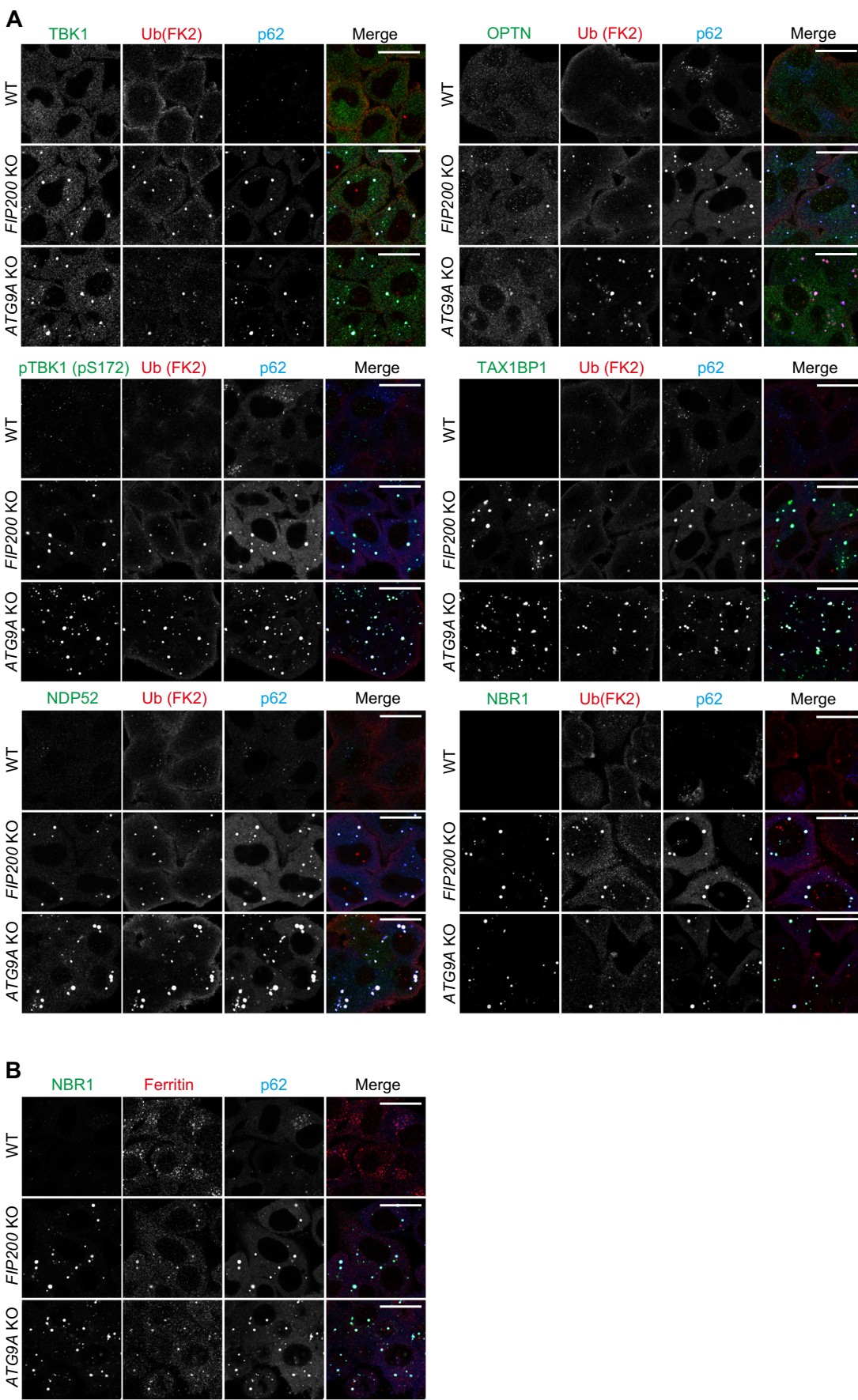

**Figure EV5.  Accumulations of autophagy adaptors in ubiquitin-positive condensates by loss of autophagy core components.**

(**A,B**) WT, *FIP200* KO, and *ATG9A* KO HeLa cells were immunostained with the indicated antibodies. Bars, 10 μm. Source data are available online for this figure.

