## [Peer Review File · The EMBO Journal]

Optineurin provides a mitophagy contact site for TBK1 activation

Koji Yamano, Momoha Sawada, Reika Kikuchi, Kafu Nagataki, Waka Kojima, Ryu Endo, Hiroki Kinefuchi, Atsushi Sugihara, Tomoshige Fujino, Aiko Watanabe, Keiji Tanaka, Gosuke Hayashi, Hiroshi Murakami, and Noriyuki Matsuda

Corresponding author(s): Koji Yamano (kojibiom@tmd.ac.jp)

Review Timeline:

Transfer from Review Commons:	9th Nov 23
Editorial Decision:	19th Dec 23
Revision Received:	9th Jan 24
Accepted:	11th Jan 24

Review
COMMONS

Editor: William Teale

Transaction Report: This manuscript was transferred to The EMBO JOURNAL following peer review at Review Commons.

Review #1

1. Evidence, reproducibility and clarity:

Evidence, reproducibility and clarity (Required)

In their study, Yamanato et al. dissect the mechanism of TBK1 activation and downstream effects, especially in its relation to mitophagy adaptor OPTN. The authors find that OPTN's interaction with ubiquitin and the autophagy machinery, forming contact sites between mitochondria and autophagic membranes, results in TBK1 accumulation and subsequent autophosphorylation. Based on these findings, the authors propose a self-propagating feedback loop wherein OPTN phosphorylation by TBK1 promotes recruitment and accumulation of OPTN to damaged mitochondria and specifically the autophagosome formation site. This formation site is then involved in TBK1 autophosphorylation, and the activated TBK1 can then further phosphorylate other pairs of OPTN and TBK1. A OPTN monobody investigation strengthens their findings.

Critique:

- It would be helpful if the authors could more clearly highlight the previous findings in OPTN-TBK1 relationship and which gaps in the understanding their study addresses.
- It is not always clear whether experiments have been replicated sufficiently; this should be indicated in the figure descriptions.
- During the discussion, references to the figures that indicate conclusions should be added where appropriate.

Figure 1 / Result "OPTN is required for TBK1 phosphorylation and subsequent autophagic Degradation":

- In a) the TBK1 and TOMM20 blots feature an image artefact that makes it appear like the blots are stitched together or there was a problem with the digital imager. The quantification in b) seems to be missing replications.
- g) should feature the wt cell line on the same blot for better comparability as well as quantification and replication like done in f)
- h) is missing the blots for controls actin and TOMM20
- In the text to e/f), the authors write that NDP52 KO effect on pS172 are comparable to controls, though the quantitation in f) indicates that pS172 signal is indeed significantly reduced compared to wt

- In the text to h/i), the authors write "there was a significant increase in the TBK1 pS172 signal in cells overexpressing OPTN", though the quantification in i) does not indicate significance levels

Figure 2 / Result "OPTN association with the autophagy machinery is required for TBK1 activation":

- In b), pTBK1 at val 1 hr only features one dot/experiment per cell line
- In the text to c), the authors claim that the mutants reduce/abolish the recruitment of OPTN to the autophagosome site. A costain for LC3, as done for SupFig 1b, would be necessary to support that specific claim.
- d) and g) as simple confirmations of KO/KD efficiency might be better suited for the supplemental part, or blots for FIP/ATG be included with the blots in e) and h)
- In the text to e), the authors claim that the levels of pS172 in the KO cell lines did not increase during mitophagy, though the blot and quantification in f) seem to indicate an increase. The results therefore don't seem to align completely with the claims that pS172 generation in response to mitophagy requires the autophagy machinery, or that FIP200 and ATG9A rather than ATG5 are critical for TBK1 phosphorylation.
- f) is missing significance indications. Its description has a typo: "bad" instead of "baf"

Figure 3 / Result "TBK1 activation does not require OPTN under basal autophagy conditions":

- In the text to SupFig2, the authors claim that pS172 levels are significantly elevated, but no significance levels are indicated
- In the text to a), NBR1 is claimed to colocalize with Ub, but no costaining with Ub is shown. The claimed lacking colocalization of OPTN with Ub is not obvious from the images; a quantification might be appropriate.
- In the text to b), the authors make reference to significant changes, but replication/quantification/significance testing is missing.

Figure 4b) is missing the pTBK1 data that is referenced in the text.

In the text to figure 5 c/d), the authors claim that certain mutants have no significant effect on mitophagy, though d) is missing significance testing

Figure 6 c/d/i) appear to be missing replication.

2. Significance:

Significance (Required)

Removal of damaged mitochondria by the mitophagy pathway provides an important safeguarding mechanism for cells. The Pink1/Parkin mechanism linked to numerous modulators and adaptor proteins ensures an efficient targeting of damaged mitochondria to the phagophore. The Ser/Thr kinase TBK1, in addition of multiple roles in innate immunity, is a major mitophagy regulator as has been revealed by the Dikic and Youle groups in 2016 (Richter et al., PNAS). The mechanistic insights provided by this manuscript add to a growing body of studies of how the autophagy machinery interconnects with cellular signalling networks. Although parts of the results need to be further validated, the data shown is of high quality, revealing an important conceptual advance. The paper is interesting and of general relevance beyond the signalling and autophagy community.

3. How much time do you estimate the authors will need to complete the suggested revisions:

Estimated time to Complete Revisions (Required)

(Decision Recommendation)

Between 3 and 6 months

No

Review #2

1. Evidence, reproducibility and clarity:

Evidence, reproducibility and clarity (Required)

Summary

In this manuscript, Yamano and colleagues show that as for Sting-mediated TBK1 activation, Optn provides a platform for TBK1 activation by autophosphorylation and that TBK1 is activated after the interaction of Optn with the autophagy machinery and ubiquitin and not before. They show that TBK1 phosphorylation is blocked by bafilomycin A1, an inhibitor of vacuolar ATPases that blocks the late phase of autophagy. Furthermore, they demonstrate that Optn is required for TBK1 phosphorylation since variation of Optn expression regulates TBK1 phosphorylation in response to PINK/Parkin-mediated autophagy. Interestingly, using immunofluorescence microscopy, they show that Optn forms sphere-like structures at the surface of damaged mitochondria which are more dispersed in the absence of TBK1. In addition, TBK1 is also recruited at the surface of damaged mitochondria and as Optn and NDP52 (but not p62) colocalize with LC3B in response to PINK/Parkin-mediated mitophagy. Next, it is demonstrated that the Leucine zipper and LIR domains of Optn (which modulate Optn interaction with autophagosome) play an important role for TBK1 activation. Additionally, the autophagy core is shown to be required for TBK1 activation. Under basal conditions, depletion of the autophagosome machinery leads to an increase in autophagy receptors (except Optn) and TBK1 phosphorylation which colocalize with ubiquitin in insoluble moieties. In contrast, Optn remains cytosolic and is dispensable for TBK1 activation in these conditions.

Then, using the fluoppi technique, the authors demonstrate that the generation of Optn-Ubiquitin condensates recruits and activates TBK1. They express in HCT116 TBK1-deficient cells engineered or pathological ALS mutations of TBK1 that affect ubiquitin interaction, structure, dimerization and kinase activity of TBK1. The expression level of TBK1 was only affected by the dimerization-deficient mutations. None of the mutations impaired Optn and TBK1 ubiquitination. Interestingly, some ALS-associated mutations affect TBK1 activity and it is said in the text that the dimerization-deficient mutations of TBK1 affect its activity proportionally to their level of expression, which is not really correct (the expression level of the mutants is very heterogeneous and not always correlates to their activity). Regarding their effect on mitophagy, the authors claim that the phosphorylation of TBK1 correlates with mitophagy which is not really the case. By using TBK1 inhibitor or TBK1-depleted cells, the authors conclude that TBK1 is the only kinase phosphorylating Optn.

However, BX-795 is not completely specific to TBK1. Finally, the authors use monoclonal antibodies against Optn effective in inhibiting mitophagy in NDP52 KO cells. Some of the monoclonal antibodies have been shown to form a ternary complex with Optn and TBK1, while others compete for the interaction between Optn and TBK1 which involves the amino-terminal region of Optn and the C-terminal region of TBK1. Monoclonal antibodies that compete for the interaction of Optn with TBK1 could alter the cellular distribution of Optn and inactivate TBK1, but they do not alter the ubiquitination of Optn. Finally, these monoclonal antibodies inhibit 50% of mitophagy.

****Major and minor points:****

Introduction

The first paragraph of the Introduction section is confused and difficult to read. First and second paragraphs (page 3 and top of page 4) are dedicated to macroautophagy processes but ended with one sentence on Parkin-mediated autophagy without further introduction, while all processes regarding mitophagy are detailed in the next paragraph.

Links between ideas developed are also somewhat missing. For example, in page 6, the three last sentences detailed the phosphorylation of autophagosome component, the fact that Optn and TBK1 genes are involved in neurodegenerative diseases and autophosphorylation of TBK1 as a pre-requirement for TBK1 activation without evident links between them, except "interestingly".

Results

****Major points:****

1. Results are often over-interpreted regarding data obtained leading to inadequate conclusions (see below for details);
2. Quantification of protein levels detected by western blot are provided as "relative intensities" without referring to specific loading control or to total protein when - phosphorylated forms are quantified (Fig. 1b, 1d, 1f, 1i, 2b, 2f, 2i, 5b, 7b, supplemental figures 2b).
3. In western blotting experiments, authors described slower migrating bands as "ubiquitinated" forms of detected proteins, but never provided experimental evidences that it could be the case. Use of non-specific deubiquitinase incubation of extracts prior to western blot could help to correctly identify ubiquitination versus other post-translational modifications such as phosphorylation, glycosylation, acetylation etc...

4. Conclusions from data obtained by immunofluorescent imaging are often drawn from only one image presented without further statistical analysis.

Page 7:

- authors referred to TBK1 phosphorylation induced by mitophagy induction as "TBK1 phosphorylation induced by Parkin-mediated ubiquitination" while mitophagy can be induced independently of Parkin (ex: via mitochondrial receptors) and without any evidence (according to referee's knowledge) of a link between ubiquitination by Parkin and TBK1 phosphorylation.

Fig 1g: Western blots performed in Penta KO cells without exogene expression of any autophagy receptors should be provided as control. Furthermore, lower expression of NDP52 relative to that of Optn (using flag antibodies) should be discussed as it can explained the differential levels in TBK1 phosphorylation observed.

Page 8:

Supplemental Fig 1a:

- The inability of authors to observe TBK1 endogenous signal in HeLa cells using commercially available antibodies is surprising as many publications reported successful staining (see Figure 1 of Suzuki et al. 2013 Cell type-specific subcellular localization of phospho-TBK1 in response to cytoplasmic viral DNA. PLoS One. 8:e83639 among others) as well as commercial promotion (see Anti-NAK/TBK1 antibody from Abcam reference: ab235253).

- Conclusions of the localization of signal on mitochondria (dispersed, in the periphery or at contact sites) are clearly over-interpreted in the absence of other membrane or autophagosome specific labeling and statistical colocalization analyses of multiple images. It is particularly difficult to assess any difference between Tax1BP1, p62 and NBR1 localization on mitochondria subdomains.

Page 9:

- First part of results ended without any conclusions.

- The observation that "TBK1 phosphorylation was not apparent in the Optn mutant cell lines, even after 3 hrs of valinomycin, ..." is inconsistent with detection of bands with anti-pS172-TBK1 antibodies in Fig 2a detected at 1hr (with F178A) and 3 hrs (4LA, F178A, and 4LA/F178A mutants) of treatment.

- Similarly, decreased levels of phosphorylated TBK1 stated for F178A mutant was only observed at 1 but not 3hrs or at 3hrs in the presence of bafilomycin.

Page 10:

The results and their repartition between figure 2 d, e, f, g, h, I and figure 3 is a bit confusing. In these experiments, it is shown Figure 2 that the absence or depletion of the autophagy machinery increase the phosphorylation of TBK1 and in Figure 3 it is shown that not only the phosphorylation of TBK1 accumulate but also the expression of NDP52, Tax1BP1 and p62. Is it because their degradation by autophagy is blocked (like for phosphoTBK1)?

Fig 2c: conclusions on the reduction of recruitment of Optn mutants on autophagosome formation seem over-interpreted as:

- 1- no labeling with LC3 has been used to identified autophagosome,
- 2- immunofluorescent signals observed with mutants are dispersed throughout the entire mitochondria network (see the merged images) rendering impossible to distinguish between autophagosome-associated mitochondria and others.

The following conclusive sentence stating that association of Optn to damaged mitochondria is not sufficient for TBK1 activation based solely on IF of figure 2c seems therefore unrelated to the obtained data.

Fig 2d: authors should explain why ATG KO cells displayed lipidated LC3B in the absence of efficient autophagy processes.

Fig 2e: despite authors statement that TBK1 phosphorylation did not increase during mitophagy in ATG KO cells, increased pS172-TBK1 is visible in FIP200 and ATG5 KO cells especially between 1 and 3 hrs of stimulation, leading to inaccurate conclusions that TBK1 phosphorylation requires the autophagy machinery. Therefore, overall assumption that both ubiquitination and autophagy subunits are required for TBK1 autophosphorylation appears erroneous.

Page 12:

- Fig 3a: conclusion that Optn signal is more cytosolic and did not localize with Ub condensates seems speculative as based on:

- 1- only one immunofluorescence image without statistical analysis
- 2- Optn and Ub signals are lower in images with Optn is analyzed compared to other

images in which NDP52, TAX1BP1 and NBR1 are detected.

Fig 3b: interpretation of western blot data is uncertain due to lack of appropriate loading control, especially with pellets (P) extracts. In addition, it is not clear how to conclude from the experiments in Fig 3b that autophagy adaptors other than Optn mediate TBK1 phosphorylation.

Minor point: reference is missing in the last sentence of the paragraph stating that K48-linked chains dominate when autophagy pathways are impaired.

Page 13:

Conversely to Optn, they find that the other autophagic receptors localize in insoluble fractions (what does it mean?) independently of TBK1 expression (experiments with DKO cells) and also independently of Optn (where is this shown?). Altogether, these experiments are far from the message of the manuscript. The title of the paragraph "TBK1 activation does not require Optn under basal autophagy conditions" is not correct because even if the level of expression of autophagic receptors and TBK1 phosphorylation are increase in response to the depletion of the autophagy machinery, it does not increase autophagy.

Fig 3d: authors should mention the nature of the upper band observed in Optn western blot and show the same experiment in since solely TBK1 depleted cells since they stated that "electrophoretic migration of Optn was not affected by TBK1 deletion". In addition, suggesting from these sole experiments that "NP52, TAX1BP1, p62, NBR1 and AZI2 form Ub-positive condensates where TBK1 is activated" seems over-interpreted.

Page 14:

- Fig 4: TBK1 phosphorylation was analyzed in Fig4d and not in Fig4b as stated. In addition, it is rather difficult to conclude from artificial multimerization experiments, as the authors have done, that interaction between Optn and autophagy components contributes to Optn multimerization in genuine conditions.

Page 15:

This work could have therapeutic consequences but the pathological mutants of TBK1 used affect ALS (Figure 5) while in the discussion it is proposed that monobodies could have a therapeutic interest in familial forms of glaucoma due to the E50K mutation of Optn. It should be better to target only one pathology.

Fig 5c, d: Authors stated that degree of TBK1 autophosphorylation correlated with OPTN phosphorylation at S177 whereas phosphorylated TBK1 is unaffected by L693Q and V700Q mutants that display decreased phosphorylated Optn. In addition, authors' interpretation of Figure 5 data is clearly problematic as they stated that:

1- neither 693Q and V700Q mutants had "significant effect on mitophagy", while decreasing efficiency from 78% to 37-51%

2- but conclude that 49.7% mitophagy levels of R357Q mutant is significant mitochondrial degradation.

Overall conclusion that mitophagy efficiency is correlated with phosphorylated TBK1 levels is therefore inaccurate.

Discussion

****Minor points:****

page 20: - reference is missing in the sentence "Optn cannot oligomerize on its own on ubiquitin-decorated mitochondria".

****Major points:****

Authors stated that they showed that Optn recruitment to damaged mitochondria, itself, is insufficient for TBK1 autophosphorylation, but did not show experiment of Optn recruitment to mitochondria and its consequences on TBK1 phosphorylation in the absence of mitophagy induction signal. Authors could for example target HA-Ash-6Ub to mitochondria in order to artificially recruit hAG-Optn to "ubiquitinated" mitochondria in the absence of mitophagy signal.

Similarly, experimental approaches used by authors lack dynamics parameters to conclude on formation and elongation of isolation membranes and contacts sites that could be probably obtained through video microscopy.

Finally, the model proposed by the authors does not take into account data showing that Optn basally interacts with ubiquitinated mitochondria and LC3 family members (see Wild et al., Phosphorylation of the autophagy receptor optineurin restricts Salmonella growth. *Science*. 2011 333:228-33), although at lower levels compared to induced conditions, relativizing the impact of the proposed model.

In conclusion, this manuscript represents a lot of work but the experiments often lack controls and are over-interpreted.

****Referees cross-commenting****

In my opinion, what emerges from these 3 reviews is that the results lack controls or have not been repeated enough to support the message that the interaction of Optn with ubiquitin and the ubiquitination machinery is sufficient to activate TBK1. In particular, as reviewer 1 says, the phosphorylation kinetics shown in Figure 1a are not consistent with TBK1 phosphorylation following the interaction of Optn with the ubiquitination machinery and ubiquitin. In Figure 1e, there is a decrease in TBK1 phosphorylation in contrast to WT cells as mentioned by Reviewer 1. In agreement with Reviewer 1, we believe that the WT cells are missing in Figure 1g.

With regard to Figure 2c, we agree with reviewer 1 that an LC3 label is missing in order to be able to interpret the data. In Figure 2e and f, we agree with reviewer 1 that it is difficult to understand why TBK1 phosphorylation increases in the absence of the autophagy machinery (FIP200 KO and ATG5KO). In Figure 3, loading controls are missing for 3b and c. The TBK1 KO cells alone are missing in Fig 2d. In Figure 2b, pTBK1 is missing. In agreement with reviewer 3, we believe that the data with fluoppi contradict the message of the manuscript since they show that TBK1 can be phosphorylated by ubiquitin in the absence of the ubiquitination machinery. In agreement with reviewer 3, we believe that the experiments in Figure 5 are very difficult to interpret. The first reviewer is right to ask the question of the replicates for figures 6c and d.

2. Significance:

Significance (Required)

This manuscript represents a lot of work but the experiments often lack controls and are over-interpreted. The manuscript is for a broad audience.

3. How much time do you estimate the authors will need to complete the suggested revisions:

Estimated time to Complete Revisions (Required)

(Decision Recommendation)

More than 6 months

Yes

Review #3

1. Evidence, reproducibility and clarity:

Evidence, reproducibility and clarity (Required)

The authors investigated the mechanisms by which TBK1 is phosphorylated and thus activated in PINK1/Parkin-mediated mitophagy. They show data indicating that OPTN, by interacting both with ubiquitin-coated mitochondria and with the autophagy machinery, provides a platform where OPTN-bound TBK1 can be hetero-autophosphorylated by adjacent TBK1.

According to the prevailing model (prior to this manuscript), TBK1 activation via autophosphorylation leads to TBK1-mediated phosphorylation of OPTN S177 and subsequent pOPTN-mediated recruitment of autophagic isolation membranes to the mitochondria. However, based on the model provided in this manuscript, OPTN needs to interact first with both autophagic membranes and ubiquitin before TBK1 can become activated.

This is an important topic. Overall, the experimental data are of high scientific quality. For the most part, the manuscript is clearly written. The figures have been made with great care. The novel insights are relevant. However, a number of issues need to be addressed or clarified.

****Major comments:****

1. Fig. 1a-b shows that pTBK1 (pS172) formation already peaks after 30 min of valinomycin. Even when bafilomycin is added, pTBK1 level already reaches a near maximum after 30 min of valinomycin. If the model proposed by the authors is correct and pTBK1 (pS172) formation requires extensive interaction of OPTN with both ubiquitin and autophagic isolation membranes, they should be able to show (by immunostaining) that OPTN already extensively forms peri-mitochondrial cup/sphere-shaped structures that colocalize with isolation membrane markers after only 30 min of valinomycin. In the present manuscript, they only show formation of such structures after 1-3 h of valinomycin.

2. The authors propose that OPTN needs to interact both with ubiquitin on mitochondria and with isolation membrane proteins such as ATG9A to allow TBK1 phosphorylation. However, their fluoppi experiments in Fig. 4 seem to contradict this. In the fluoppi experiments, the authors generate multimeric OPTN-Ub foci and this is apparently sufficient to induced TBK1 phosphorylation at S172 (shown in 4d,f). In this experiment, there is no induction of autophagy or formation of isolation membranes, and TBK1 nevertheless gets activated.

3. Can the authors be more concrete/specific in the discussion about the molecular mechanisms that explain why this 'platform' that is created by OPTN-autophagy machinery interactions is so crucial for TBK1 activation? If I understand the model in Fig. 7D correctly, the OPTN-autophagy machinery interactions are mainly important because they reduce the distance between OPTN-bound TBK1 molecules so that they can trans-phosphorylate each other. But if TBK1 autophosphorylation was just a matter of proximity between OPTN-bound TBK1 molecules, interaction of OPTN with densely ubiquitinated mitochondria should already be sufficient for TBK1 phosphorylation. When multiple OPTN molecule bind to one ubiquitin chain or to closely adjacent ubiquitin chains (similar to the fluoppi experiments), TBK1 molecules binding to OPTN would be expected to be already closely enough to one another for trans-autophosphorylation.

4. Fig. 5c,d and P. 16: the mitophagy experiments in TBK1^{-/-} cells expressing the different mutant forms of TBK1 are hard to interpret because it is not clear which mitophagy differences are statistically significant. The main text about this part (p. 16) is also confusing.

5. Many graphs lack statistics: Fig. 2b (pTBK1), Fig. 2f, Fig. 5b, Fig. 5d, Fig. 6c.

****Other comments:****

1. Fig. 1a: how do they know that the upper OPTN band is ubiquitinated OPTN?

2. Fig. 1a,b: the bafilomycin stabilization of pTBK1, OPTN and pOPTN indicates that these proteins are substantially degraded by autophagy within 30-60 minutes. This seems extremely fast for mitophagy completion. Please discuss.

3. Fig. 1a and rest of the manuscript: is there a reason why the authors only looked at S177 phosphorylation of OPTN and not also at OPTN S473, which is also phosphorylated by TBK1?
4. Fig. 1e-f: the main text states that "NDP52 KO effects on the pS172 signal were comparable to controls", but the blot in 1e and the graph in 1f indicate a difference between NDP52KO and WT (significant difference shown in 1f). This is confusing.
5. P. 9: "TBK1 phosphorylation however was not apparent in the OPTN mutant lines, even after 3 hrs with valinomycin, indicating that autophagy adaptors are essential for TBK1 activation (Fig. 2a)". However, the pTBK1 blot in Fig. 1a does show pTBK1 formation in the OPTN mutant (4LA etc.) lines. This is confusing.
6. P. 10: "we subtracted the basal phosphorylation signal from that generated post-valinomycin (1 hr) and bafilomycin (3 hr)". Do they mean "from that generated post-valinomycin (3 hr) and bafilomycin (3 hr)"?
7. P. 10, same paragraph: "the phosphorylation signal was ~90 but was less than 30 in ATG9A KO cells." Unclear what they mean by 90 and 30. 90% and 30%? 90-fold and 30-fold?
8. Fig. 3a: Do they have an idea what kind of ubiquitinated substrates are contained in the ubiquitin-positive condensates that accumulate in FIP200 KO and ATG9A KO cells (i.e. without valinomycin treatment)?
9. P. 12 and Fig. 3a: please explain why they look at ferritin, to improve readability.
10. Fig. 3a: please also include Ub stain for NBR1.
11. Fig. 3d: the OPTN blot shows 2 OPTN bands. What does the upper OPTN band represent here?
12. P. 14 and Fig. 4b: "Here, we found that phosphorylation of ... TBK1 (S172) was induced by the OPTN-ub fluoppi (Fig. 4b)." However, Fig 4b does not show a pTBK1 blot.

2. Significance:

Significance (Required)

The novel insights are relevant.

According to the prevailing model (prior to this manuscript), TBK1 activation via autophosphorylation leads to TBK1-mediated phosphorylation of OPTN S177 and subsequent pOPTN-mediated recruitment of autophagic isolation membranes to the mitochondria. However, based on the model provided in this manuscript, OPTN needs to interact first with both autophagic membranes and ubiquitin before TBK1 can become activated.

3. How much time do you estimate the authors will need to complete the suggested revisions:

Estimated time to Complete Revisions (Required)

(Decision Recommendation)

Between 1 and 3 months

No

Full Revision

Manuscript number: RC-2023-01953

Corresponding author(s): Koji Yamano

1. General Statements [optional]

We would like to thank the editor and reviewers for investing tremendous efforts and times for our manuscript. We sincerely appreciate the constructive and valuable comments. Our point-by-point responses are indicated below.

Point-by-point description of the revisions

Reviewer #1 (Evidence, reproducibility and clarity (Required)):

In their study, Yamano et al. dissect the mechanism of TBK1 activation and downstream effects, especially in its relation to mitophagy adaptor OPTN. The authors find that OPTN's interaction with ubiquitin and the autophagy machinery, forming contact sites between mitochondria and autophagic membranes, results in TBK1 accumulation and subsequent autophosphorylation. Based on these findings, the authors propose a self-propagating feedback loop wherein OPTN phosphorylation by TBK1 promotes recruitment and accumulation of OPTN to damaged mitochondria and specifically the autophagosome formation site. This formation site is then involved in TBK1 autophosphorylation, and the activated TBK1 can then further phosphorylate other pairs of OPTN and TBK1. A OPTN monobody investigation strengthens their findings.

Critique:

- It would be helpful if the authors could more clearly highlight the previous findings in OPTN-TBK1 relationship and which gaps in the understanding their study addresses.*

We thank the reviewer for this comment. As suggested, we have highlighted previous findings and detailed in the Discussion how the study advances our understanding of TBK1 activation.

- It is not always clear whether experiments have been replicated sufficiently; this should be indicated in the figure descriptions.*

In the original manuscript, most of the data shown was derived from duplicated experiments. For the revised version, we repeated experiments as needed to generate the replication necessary (i.e, N = 3) for determining statistical significance. Error bars and statistical significance have been added to the graphs and figure legends accordingly.

- During the discussion, references to the figures that indicate conclusions should be added where appropriate.*

We thank the reviewer for the suggestion. References to figures have been added were appropriate to the Discussion.

Figure 1 / Result "OPTN is required for TBK1 phosphorylation and subsequent autophagic Degradation":

o In a) the TBK1 and TOMM20 blots feature an image artefact that makes it appear like the blots are stitched together or there was a problem with the digital imager. The quantification in b) seems to be missing replications.

We found that the artifact came from an automatic pixel interpolation process in Adobe Photoshop when the image was rotated by a small angle. We have provided the original immunoblotting data below as evidence that the data were not stitched from separate images. More accurate representations of the images without the artifact are now shown in Fig1 A of the revised manuscript.

For Fig 1b, the experiment was independently replicated three times with error bars added to each plot on the graph.

o g) should feature the wt cell line on the same blot for better comparability as well as quantification and replication like done in f)

As suggested, we have included the WT cell line in the immunoblot (See Fig 1g). In addition, Reviewer 2 asked that we provide data for Penta KO cells without exogenous expression of the autophagy adaptors and expressed concern regarding the lower expression of NDP52 relative to OPTN. To address these issues, we repeated the mitophagy experiments and detected phosphorylated TBK1 in six different cell lines: WT, Penta KO, Penta KO stably expressing OPTN at both low and high expression levels, and Penta KO stably expressing NDP52 at low and high expression levels. Immunoblots of phos-TBK1(pS172), TBK1, OPTN, NDP52, TOMM20, and actin were generated under four different conditions (DMSO, valinomycin for 1 hr, valinomycin for 3 hrs, and valinomycin in the presence of bafilomycin for 3 hrs). In addition, phos-TBK1 abundance in the six cell lines was determined in response to val and baf for

3 hrs and the expression levels of NDP52 and OPTN were similarly determined in response to DMSO. Error bars based on three independent experiments have been incorporated into the data, which are shown in Figure 1g and 1h of the revised manuscript.

o h) is missing the blots for controls actin and TOMM20

Immunoblots for actin and TOMM20 have been added, please see Fig 1i in the revised manuscript.

o In the text to e/f), the authors write that NDP52 KO effect on pS172 are comparable to controls, though the quantitation in f) indicates that pS172 signal is indeed significantly reduced compared to wt

The reviewer is correct, the phos-TBK1 (pS172) signal in *NDP52* KO cells is reduced compared to that in WT cells, but is only moderately lower in *NDP52* KO cells relative to *OPTN* KO. We regret the error, which has been corrected in the revised manuscript.

o In the text to h/i), the authors write "there was a significant increase in the TBK1 pS172 signal in cells overexpressing OPTN", though the quantification in i) does not indicate significance levels

We performed statistical analyses on the phos-TBK1 (pS172) levels between cells with or without OPTN overexpression and have added the degree of significance to Fig 1j. As indicated in the original manuscript, there was a significant increase in phos-TBK1 (pS172) levels when OPTN was overexpressed.

*Figure 2 / Result "OPTN association with the autophagy machinery is required for TBK1 activation":
o In b), pTBK1 at val 1 hr only features one dot/experiment per cell line*

Three independent replicates of the experiment (val 1 hr) were performed. The levels of phos-TBK1 (pS172), total TBK1, and actin were quantified, and the graph was remade with error bars and statistical significance incorporated. Please see Fig 2b in the revised manuscript.

o In the text to c), the authors claim that the mutants reduce/abolish the recruitment of OPTN to the autophagosome site. A costain for LC3, as done for SupFig 1b, would be necessary to support that specific claim.

To address the reviewer's concern regarding the recruitment of OPTN mutants to the autophagosomal formation site, we performed two different experiments. First, when OPTN WT is recruited to the contact site between the autophagosomal formation site and damaged mitochondria, it should be heterogeneously distributed across mitochondria. In contrast, OPTN mutants that are unable to associate with the autophagosome formation sites should be largely localized to damaged mitochondria since the mutants are still capable of binding ubiquitin. When we examined the mitochondrial distribution of OPTN WT following valinomycin treatment for 1 hr, more than 80% of the Penta KO cells exhibited a heterogeneous distribution, whereas only 10% of the cells showed a similar distribution for OPTN 4LA or OPTN 4LA/F178A (please see Fig 2g in the revised manuscript). Although the OPTN F178A mutant exhibited 50% heterogeneous distribution (Fig 2g), this may be because OPTN F178A retains the ability to interact with ATG9A vesicles. In fact, our previous mitophagy analyses (Keima-based FACS analysis, Yamano et al 2020 JCB), which are strongly correlated with OPTN mitochondrial distribution, showed that the OPTN F178A mutant moderately (~ 60%) induced mitochondrial degradation. This degradation effect

was slightly higher (80%) with OPTN WT but significantly lower (9%) with the 4LA/F178A mutant. In the second experiment, Penta KO cells expressing either OPTN WT or the OPTN mutants were immunostained for exogenous FLAG-tagged OPTN, endogenous WIPI2, and HAP60 (a mitochondrial marker) after valinomycin treatment for 1 hr (see Fig 2e and 2f in the revised manuscript). Because LC3B is assembled on the autophagosomal formation site as well as completed autophagosomes, we detected endogenous WIPI2 because WIPI2 is only recruited to autophagosomal formation sites (Dooley et al. 2014 Mol Cell). Confocal microscopy images and their associated quantification data indicate that WIPI2 foci formation during mitophagy was reduced in Penta KO cells expressing the OPTN mutants (4LA, F178A and 4LA/F178A) as compared to Penta KO cells expressing OPTN WT.

o d) and g) as simple confirmations of KO/KD efficiency might be better suited for the supplemental part, or blots for FIP/ATG be included with the blots in e) and h)

Based on the reviewer comments, we performed additional experiments related to Figure 2 and have incorporated the new data into the revised figure. The original Figure 2d, e, f, g, h, and I have been moved to supplemental Figure 5.

o In the text to e), the authors claim that the levels of pS172 in the KO cell lines did not increase during mitophagy, though the blot and quantification in f) seem to indicate an increase. The results therefore don't seem to align completely with the claims that pS172 generation in response to mitophagy requires the autophagy machinery, or that FIP200 and ATG9A rather than ATG5 are critical for TBK1 phosphorylation.

Although newly generated pS172 TBK1 was reduced in *FIP200* KO and *ATG9A* KO cells relative to WT cells, the signals gradually increased. In the autophagy KO cell lines (*FIP200* KO and *ATG9A* KO), phos-TBK1 accumulates prior to mitophagy stimulation. Although suggesting it is mitophagy-independent, phos-TBK1 accumulation prior to mitophagy stimulation in autophagy KO cell lines complicated interpretation of the results. To avoid this issue, we used siRNA to transiently knock down *FIP200* and *ATG9A*. As shown in the original manuscript (Fig 2g, h, I in the original manuscript, supplementary Fig 5d, e, f in the revised manuscript), knockdown of *FIP200* and *ATG9A* prior to mitophagy induction allowed us to observe mitophagy-dependent phosphorylation of TBK1. This result strongly suggests that the autophagy machinery does induce TBK1 phosphorylation in response to Parkin-mediated mitophagy. However, TBK1 phosphorylation still increases, albeit very slightly, in the *FIP200* and *ATG9A* knock down cells. Thus, it may be reasonable to assume that OPTN-dependent phosphorylation of TBK1 can occur to a certain degree even in the absence of autophagy components. We have noted this in the Discussion. While conducting experiments for the revised manuscript, we determined that TAX1BP1 is responsible for the accumulation of phos-TBK1 in the autophagy KO cell lines under basal conditions. When TAX1BP1 is knocked down in *FIP200* KO or *ATG9A* KO cells, the basal accumulation of phos-TBK1 was eliminated and then we could observe mitophagy-specific TBK1 phosphorylation (please see Fig 2h, i, j, k in the revised manuscript). These results showed that mitophagy-dependent phos-TBK1 is largely attenuated in *FIP200* KO and was almost completely eliminated in *ATG9A* KO cells (Fig 2k in the revised manuscript).

o f) is missing significance indications. Its description has a typo: "bad" instead of "baf"

Newly synthesized pTBK1 (pS172) during mitophagy was quantified and statistical significance incorporated into the figure (please see supplementary Fig 5c). The identified typo has been corrected.

Figure 3 / Result "TBK1 activation does not require OPTN under basal autophagy conditions":

o In the text to SupFig2, the authors claim that pS172 levels are significantly elevated, but no significance levels are indicated

Statistical significance was determined for all proteins shown in original supplementary Fig 2 and the results have been incorporated into the relevant figure. The original supplementary Fig 2 is now supplementary Fig 6.

o In the text to a), NBR1 is claimed to colocalize with Ub, but no costaining with Ub is shown. The claimed lacking colocalization of OPTN with Ub is not obvious from the images; a quantification might be appropriate.

Since the anti-NBR1 antibody used in the original manuscript is derived from mouse, we were unable to use it in conjunction with the mouse ubiquitin antibody. Because ubiquitin-positive foci and NBR1-positive foci contain p62 (original Fig 3a) and NBR1 and p62 are known to tightly interact each other (Kirkin et al. 2009 Mol Cell and Sanchez-Martin et al. 2020 EMBO Rep), we stated that "NBR1 colocalizes with Ub". However, the reviewer is correct. To remedy this confusion, we obtained a rabbit anti-NBR1 antibody (a gift from the Masaaki Komatsu group) and used it to co-immunostain with anti-Ub antibodies (please see supplementary Fig 7a of the revised manuscript). NBR1 foci colocalize with both ubiquitin and p62 in *FIP200* KO and *ATG9A* KO cells. Further, based on comments from Reviewer 2, we purchased several anti-TBK1 antibodies and identified one that was able to detect endogenous TBK1 by immunostaining (see Figure 1 for reviewers in our response to Reviewer 2 below). Using this anti-TBK1 antibody, we showed that a part of TBK1 also associates with ubiquitin and p62-positive aggregates.

o In the text to b), the authors make reference to significant changes, but replication/ quantification/ significance testing is missing.

We independently performed the same experiments three times. The levels of TBK1, phos-TBK1 (pS172), all five autophagy adaptors, and TOMM20 in both the supernatants and pellets have been quantified with error bars and statistical significance indicated. These results have been incorporated into Figure 3c in the revised manuscript.

Figure 4b) is missing the pTBK1 data that is referenced in the text. In the text to figure 5 c/d), the authors claim that certain mutants have no significant effect on mitophagy, though d) is missing significance testing

Figure 6 c/d/i) appear to be missing replication.

For Figure 4b, phos-TBK1 was immunoblotted (See Fig 4b of the revised manuscript). For Figure 5b and d, statistical significance was determined for the effect of TBK1 mutations on autophosphorylation and OPTN phosphorylation and the effect of the TBK1 mutants on Parkin-mediated mitophagy. For Figure 6 c/d/i, the experiment was repeated; error bars and statistical significance have been added to the associated graphs.

Reviewer #1 (Significance (Required)): Removal of damaged mitochondria by the mitophagy pathway provides an important safeguarding mechanism for cells. The Pink1/Parkin mechanism linked to numerous modulators and adaptor proteins ensures an efficient targeting of damaged mitochondria to the phagophore. The Ser/Thr kinase TBK1, in addition of multiple roles in innate immunity, is a major

mitophagy regulator as has been revealed by the Dikic and Youle groups in 2016 (Richter et al., PNAS). The mechanistic insights provided by this manuscript add to a growing body of studies of how the autophagy machinery interconnects with cellular signalling networks. Although parts of the results need to be further validated, the data shown is of high quality, revealing an important conceptual advance. The paper is interesting and of general relevance beyond the signalling and autophagy community.

We would like to thank Reviewer 1 for the comments and suggestions, many of which improved our manuscript. We hope that the reviewer's comments have been adequately addressed in the revised manuscript.

Reviewer #2 (Evidence, reproducibility and clarity (Required)): Summary In this manuscript, Yamano and colleagues show that as for Sting-mediated TBK1 activation, Optn provides a platform for TBK1 activation by autophosphorylation and that TBK1 is activated after the interaction of Optn with the autophagy machinery and ubiquitin and not before. They show that TBK1 phosphorylation is blocked by bafilomycin A1, an inhibitor of vacuolar ATPases that blocks the late phase of autophagy. Furthermore, they demonstrate that Optn is required for TBK1 phosphorylation since variation of Optn expression regulates TBK1 phosphorylation in response to PINK/Parkin-mediated autophagy. Interestingly, using immunofluorescence microscopy, they show that Optn forms sphere like structures at the surface of damaged mitochondria which are more dispersed in the absence of TBK1. In addition, TBK1 is also recruited at the surface of damaged mitochondria and as Optn and NDP52 (but not p62) colocalize with LC3B in response to PINK/Parkin-mediated mitophagy. Next, it is demonstrated that the Leucine zipper and LIR domains of Optn (which modulate Optn interaction with autophagosome) play an important role for TBK1 activation. Additionally, the autophagy core is shown to be required for TBK1 activation. Under basal conditions, depletion of the autophagosome machinery leads to an increase in autophagy receptors (except Optn) and TBK1 phosphorylation which colocalize with ubiquitin in insoluble moieties. In contrast, Optn remains cytosolic and is dispensable for TBK1 activation in these conditions. Then, using the fluoppi technique, the authors demonstrate that the generation of Optn-Ubiquitin condensates recruits and activates TBK1. They express in HCT116 TBK1-deficient cells engineered or pathological ALS mutations of TBK1 that affect ubiquitin interaction, structure, dimerization and kinase activity of TBK1. The expression level of TBK1 was only affected by the dimerization-deficient mutations. None of the mutations impaired Optn and TBK1 ubiquitination. Interestingly, some ALS-associated mutations affect TBK1 activity and it is said in the text that the dimerization-deficient mutations of TBK1 affect its activity proportionally to their level of expression, which is not really correct (the expression level of the mutants is very heterogeneous and not always correlates to their activity). Regarding their effect on mitophagy, the authors claim that the phosphorylation of TBK1 correlates with mitophagy which is not really the case. By using TBK1 inhibitor or TBK1-depleted cells, the authors conclude that TBK1 is the only kinase phosphorylating Optn. However, BX-795 is not completely specific to TBK1. Finally, the authors use monoclonal antibodies against Optn effective in inhibiting mitophagy in NDP52 KO cells. Some of the monoclonal antibodies have been shown to form a ternary complex with Optn and TBK1, while others compete for the interaction between Optn and TBK1 which involves the amino-terminal region of Optn and the C-terminal region of TBK1. Monoclonal antibodies that compete for the interaction of Optn with TBK1 could alter the cellular distribution of Optn and inactivate TBK1, but they do not alter the ubiquitination of Optn. Finally, these monoclonal antibodies inhibit 50% of mitophagy.

Major and minor points: Introduction The first paragraph of the Introduction section is confused and difficult to read. First and second paragraphs (page 3 and top of page 4) are dedicated to macroautophagy processes but ended with one sentence on Parkin-mediated autophagy without further introduction, while all processes regarding mitophagy are detailed in the next paragraph. Links between

ideas developed are also somewhat missing. For example, in page 6, the three last sequences detailed the phosphorylation of autophagosome component, the fact that Optn and TBK1 genes are involved in neurodegenerative diseases and autophosphorylation of TBK1 as a pre-requirement for TBK1 activation without evident links between them, except "interestingly".

In response to the reviewer's suggestion, we have rewritten the Introduction. The first paragraph focused on introducing the molecular mechanism underlying macroautophagy and the second paragraph focused on Parkin-mediated mitophagy. As the reviewer indicated, the ALS mutations and TBK1 phosphorylation during Parkin-mediated mitophagy are not well related, so we moved the background material on the relationship between OPTN and TBK1 in neurodegenerative diseases to the beginning of the section describing Figure 5. We believe these changes have made the Introduction easier to read and understand.

Results

Major points:

1- Results are often over-interpreted regarding data obtained leading to inadequate conclusions (see below for details);

We regret the reviewer's concerns regarding over-interpretation. To address this issue, we have carefully considered the data, performed additional experiments where necessary, and rewritten the results accordingly. Please see our point-by-point responses below.

2- Quantification of protein levels detected by western blot are provided as "relative intensities" without referring to specific loading control or to total protein when -phosphorylated forms are quantified (Fig. 1b, 1d, 1f, 1i, 2b, 2f, 2i, 5b, 7b, supplemental figures 2b).

For the immunoblots, we loaded the same amount of total cell lysate and the phosphorylated forms were quantified relative to the total protein input. This has been mentioned in the Materials and Methods.

3- In western blotting experiments, authors described slower migrating bands as "ubiquitinated" forms of detected proteins, but never provided experimental evidences that it could be the case. Use of non-specific deubiquitinase incubation of extracts prior to western blot could help to correctly identified ubiquitination versus other post-translational modifications such as phosphorylation, glycosylation, acetylation etc...

We appreciate the reviewer's suggestion. The cell lysates after mitophagy induction were incubated *in vitro* with a recombinant USP2 core domain (non-specific DUB), and then immunoblotted. As shown in supplemental Fig 1 of the revised manuscript, the slower migrating OPTN bands disappeared in a USP2-dependent manner. The slower migrating NDP52 and TOMM20 bands likewise disappeared. These results confirm that the slower migrating OPTN, NDP52, and TOMM20 bands are ubiquitinated.

4- Conclusions from data obtained by immunofluorescent imaging are often drawn from only one image presented without further statistical analysis.

Statistical significance was determined for the immunofluorescent data (original figures 1j, 2c and 3a). Please see Fig 1l, 2f, 2g, and 3a in the revised manuscript.

Page 7: - authors referred to TBK1 phosphorylation induced by mitophagy induction as "TBK1 phosphorylation induced by Parkin-mediated ubiquitination" while mitophagy can be induced independently of Parkin (ex: via mitochondrial receptors) and without any evidence (according to referee's knowledge) of a link between ubiquitination by Parkin and TBK1 phosphorylation.

As the reviewer indicated, Parkin-independent and ubiquitination-independent mitophagy pathways are also known (i.e. receptor-mediated mitophagy driven by NIX, BNIP3, BCL2L13, FKBP8, FUNDC1, or Atg32). Therefore, references to "mitophagy" in our manuscript were reworded as "Parkin-mediated mitophagy". Since TBK1 phosphorylation is observed before mitochondria are degraded and is dependent on Parkin-mediated ubiquitin (for example, see Fig 1c), we use the phrase "TBK1 phosphorylation triggered by Parkin-mediated OMM ubiquitination".

Fig 1g: Western blots performed in Penta KO cells without exogene expression of any autophagy receptors should be provided as control. Furthermore, lower expression of NDP52 relative to that of Optn (using flag antibodies) should be discussed as it can explained the differential levels in TBK1 phosphorylation observed.

As suggested, we repeated the experiment using Penta KO cells in the absence of exogenous autophagy adaptor expression. Furthermore, we expressed different amounts of NDP52 and OPTN (indicated as low and high in the figure) in Penta KO cells to rule out the possibility that higher TBK1 phosphorylation is induced by simple overexpression of autophagy adaptor (please see Fig 1g and h in the revised manuscript). At high NDP52 expression (2.5-3.0-fold higher than endogenous NDP52), phosphorylated TBK1 was reduced to ~30% the level of that observed in WT cells after 3 hrs with val and baf. In contrast, Penta KO cells with higher OPTN expression (3.0-fold higher than endogenous OPTN) had phosphorylated TBK1 signals that were 2-fold higher than those in WT cells. Based on these results, we concluded that OPTN is an important adaptor for TBK1 activation during Parkin-mediated mitophagy.

Page 8: Supplemental Fig 1a: - The inability of authors to observe TBK1 endogenous signal in HeLa cells using commercially available antibodies is surprising as many publications reported successful staining (see Figure 1 of Suzuki et al. 2013 Cell type-specific subcellular localization of phospho-TBK1 in response to cytoplasmic viral DNA. PLoS One. 8:e83639 among others) as well as commercial promotion (see Anti-NAK/TBK1 antibody from Abcam reference: ab235253).

For the original manuscript, anti-TBK1 antibodies purchased from abcam (ab235253), CST (#3013S), Proteintech (28397-1-AP), and GeneTex (GTX12116) for immunostaining were unable to yield TBK1-positive signals (please see Fig 1 for reviewers below). WT and *TBK1*^{-/-} HCT116 cells stably expressing Parkin were treated with valinomycin for 1 hr and immunostained with the indicated antibodies. Anti-phospho-TBK1 antibody (CST, #5483) was used as a positive control. Based on these results, we stated in the original manuscript that the "endogenous TBK1 signal could not be observed using commercially available antibodies". At the reviewer's suggestion, we purchased anti-TBK1 antibodies from abcam (ab40676) and CST (#38066). As shown in the figure below, the immunofluorescent signals generated by these antibodies were detected in WT, but not in *TBK1*^{-/-} cells. The CST (#38066) antibody yielded a stronger signal, most of which was on damaged mitochondria. Thanks to this suggestion, we repeated the experiment using the new anti-TBK1 antibody. Furthermore, based on a suggestion from Reviewer 3, we detected mitochondrial recruitment of TBK1 during mitophagy stimulation (valinomycin for 30 min or 2 hrs in the presence and absence of bafilomycin; supplemental Fig 2 in the revised manuscript). We also

detected association of endogenous TBK1 with ubiquitin-positive condensates in WT, *FIP200* KO, and *ATG9A* KO cells (Fig 3a and supplementary Fig 7a in the revised manuscript).

Figure 1 for reviewers

WT and *TBK1*^{-/-} HCT116 cells stably expressing Parkin were treated with valinomycin for 1hr. The cells were immunostained with anti-TOMM20 and the indicated anti-TBK1 antibodies.

- Conclusions of the localization of signal on mitochondria (dispersed, in the periphery or at contact sites) are clearly over-interpreted in the absence of other membrane or autophagosome specific labeling and

statistical colocalization analyses of multiple images. It is particularly difficult to assess any difference between Tax1BP1, p62 and NBR1 localization on mitochondria subdomains.

We previously expressed each FLAG-tagged autophagy adaptor in Penta KO cells and observed their localization during Parkin-mediated mitophagy and found that exogenous FLAG-tagged OPTN and NDP52, but not p62, colocalized with LC3B (Yamano et al 2020 JCB). No one has assessed and compared the localization of all five endogenous autophagy adaptors. Although we still believe that the results (supplemental Fig1 in the original manuscript) are informative for researchers in the autophagy field, we decided to remove that data from the revised manuscript since they are not the main focus of the study. We will consider publishing those data elsewhere in the future after co-staining with autophagosome markers and assessing the statistical significance of colocalization as the reviewer suggested.

Page 9:

- First part of results ended without any conclusions.

As detailed in the previous response, we have removed results for mitophagic recruitment of autophagy adaptors (supplementary Figure 1 in the original manuscript).

- The observation that "TBK1 phosphorylation was not apparent in the Optn mutant cell lines, even after 3 hrs of valinomycin, ..." is inconsistent with detection of bands with anti-pS172-TBK1 antibodies in Fig 2a detected at 1hr (with F178A) and 3 hrs (4LA, F178A, and 4LA/F178A mutants) of treatment.

We apologize for the confusion. This statement was clearly our mistake. We had intended to state when "all autophagy adaptors are deleted" no phosphorylated TBK1 was observed. We have rewritten this part as "TBK1 phosphorylation was not apparent in the Penta KO cells even after 3 hrs with valinomycin".

- Similarly, decreased levels of phosphorylated TBK1 stated for F178A mutant was only observed at 1 but not 3hrs or at 3hrs in the presence of bafilomycin.

Based on the mitophagy assay previously reported (Yamano et al 2020 JCB), the F178A mutant only moderately inhibited mitophagy (60% mitophagy with the F178A mutant vs 80% mitophagy with OPTN WT). Conversely, the 4LA mutant and 4LA/F178A double mutant had stronger inhibitory effects on mitophagy (35% for 4LA and 9% mitophagy for 4LA/F178A). Therefore, the levels of phos-TBK1 after 1 hr with valinomycin or 3 hrs with valinomycin in the presence of bafilomycin are consistent with mitophagy progression. When mitophagy proceeds efficiently, the amount of phos-TBK1 in the 1 hr val samples is reduced relative to the 3 hr val samples due to autophagic degradation.

To more clearly observe and compare the levels of mitophagy-dependent phos-TBK1 among Penta KO cells expressing OPTN WT and the mutants, we treated cells with valinomycin in the presence of bafilomycin for 0, 0.5, 1, and 2 hrs and quantified phos-TBK1. The results are shown in Fig 2c and d in the revised manuscript. The phos-TBK1 signal increased over time with val and baf treatment in all OPTN expressing cells. Cells with OPTN WT generated the most phos-TBK1, whereas the signal generated by the F178A mutant was 75% that of the OPTN WT-expressing cells and the 4LA and 4LA/F178A mutants were about 40%. The experiments were independently replicated three times and error bars and statistical significance were incorporated into the associated graph. These results indicate that OPTN association with the autophagy machinery, in particular ATG9A vesicles, is important for TBK1 activation.

Page 10:

The results and their repartition between figure 2 d, e, f, g, h, I and figure 3 is a bit confusing. In these experiments, it is shown Figure 2 that the absence or depletion of the autophagy machinery increase the phosphorylation of TBK1 and in Figure 3 it is shown that not only the phosphorylation of TBK1 accumulate but also the expression of NDP52, Tax1BP1 and p62. Is it because their degradation by autophagy is blocked (like for phosphoTBK1)?

The reviewer is correct that autophagy adaptors other than OPTN (especially TAX1BP1, p62 and NBR1) are constantly degraded by macro/micro autophagy (Mejlvang et al. 2018 J Cell Biol and Yamano et al. 2021 BBA Gen Subj). Therefore, these adaptors accumulate in autophagy deficient cell lines (original Fig 3). In this study, we found that in the absence of mitophagy stimulation phos-TBK1 accumulates in autophagy deficient cell lines. This suggests that the accumulated autophagy adaptors induce TBK1 phosphorylation under basal conditions. In the original manuscript, we claimed that TBK1 phosphorylation under basal conditions does not require OPTN since in *FIP200* KO and *ATG9A* KO cells it did not accumulate and did not primarily colocalize with ubiquitin- and TBK1-positive foci (original Fig 3). To gain more direct evidence for the revised manuscript, we performed additional experiments and discovered that TAX1BP1 is the adaptor responsible for TBK1 autophosphorylation under basal autophagy. We treated *FIP200* KO and *ATG9A* KO cells with siRNAs against OPTN, NDP52, TAX1BP, p62, and NBR1, and immunoblotted total cell lysates with an anti-phos-TBK antibody. As shown in Fig 3f in the revised manuscript, TAX1BP1 siRNA treatment decreased phos-TBK1 levels without affecting total TBK1. This result indicates that the accumulation of TAX1BP1 in the *FIP200* KO and *ATG9A* KO cells induced TBK1 autophosphorylation under basal conditions. Considering this result, we treated WT, *FIP200* KO, and *ATG9A* KO cells with TAX1BP1 siRNA, and then induced Parkin-mediated mitophagy with valinomycin in the presence of bafilomycin. This strategy eliminated the basal accumulation of phos-TBK1 and allowed us to focus on mitophagy-dependent TBK1 phosphorylation. Please see revised Fig 2h, I, j, and k. The results showed that mitophagy-dependent phos-TBK1 is predominantly attenuated in *FIP200* KO and *ATG9A* KO cells. In Figs 2 and 3, we would like to emphasize that OPTN is required for TBK1 phosphorylation in response to Parkin-mediated mitophagy, whereas TAX1BP1 is required for TBK1 phosphorylation in basal autophagy. Since Reviewer 3 commented that interpretation of the data in original Figs 2d, e, and f was challenging, we elected to move those results to the supplemental figures. We have incorporated the newly acquired data (mitophagy using *FIP200* KO or *ATG9A* KO with TAX1BP1 siRNA cells) into the main figure. We believe that this makes the text easier for readers to understand.

- Fig 2c: conclusions on

the reduction of recruitment of Optn mutants on autophagosome formation seem over-interpreted as:

1- no labeling with LC3 has been used to identified autophagosome,

2- immunofluorescent signals observed with mutants are dispersed throughout the entire mitochondria network (see the merged images) rendering impossible to distinguish between autophagosome-associated mitochondria and others.

The following conclusive sentence stating that association of Optn to damaged mitochondria is not sufficient for TBK1 activation based solely on IF of figure 2c seems therefore unrelated to the obtained data.

To address concerns about the recruitment of OPTN mutants to the autophagosome formation site, we performed additional experiments. Penta KO cells and those expressing OPTN WT and mutants were treated with valinomycin for 1 hr, and FLAG-tagged OPTN, endogenous WIPI2, and HAP60

(mitochondrial marker) were detected by immunostaining. We detected endogenous WIPI2 because WIPI2 is recruited only to autophagosome formation sites (Dooley et al. 2014 Mol Cell), whereas LC3B assembles on autophagosome formation sites and is also associated with completed autophagosomes. Confocal microscopy images showed that cup-shaped OPTN WT that had been recruited to damaged mitochondria colocalized with WIPI2. Quantification further showed that during mitophagy the number of WIPI2 foci seen in cells expressing OPTN WT decreased in Penta KO cells and cells expressing OPTN mutants (4LA, F178A and 4LA/F178A). These data are shown in Fig 2e and f in the revised manuscript. In addition, we quantified the number of cells that either exhibited heterogeneous or homogeneous recruitment of OPTN to damaged mitochondria after treatment with valinomycin for 1 hr. More than 80% of Penta KO cells with OPTN WT had heterogeneous OPTN recruitment, whereas this distribution was only present in 10% of cells expressing either OPTN 4LA or OPTN 4LA/F178A. Although cells expressing the OPTN F178A mutant exhibited 50% heterogeneous recruitment, this may be because the mutant can interact with ATG9A. As mentioned above, our previous mitophagy analyses (Keima-based FACS analysis, Yamano et al 2020 JCB) showed that the OPTN F178A mutant induced ~60% mitochondrial degradation (which is correlated strongly with OPTN distribution), whereas it was 80% with OPTN WT and 9% with 4LA/F178A.

- Fig 2d: authors should explain why ATG KO cells displayed lipidated LC3B in the absence of efficient autophagy processes.

We thank the reviewer for the suggestion. We added the following sentence to explain the function of ATG5 in LC3B lipidation. "Since LC3B lipidation is catalyzed by ATG5, but not FIP200 and ATG9A, the lipidated form disappears only in ATG5 KO cells (Hanada et al 2007 J Biol Chem). "

- Fig 2e: despite authors statement that TBK1 phosphorylation did not increase during mitophagy in ATG KO cells, increased pS172-TBK1 is visible in FIP200 and ATG5 KO cells especially between 1 and 3 hrs of stimulation, leading to inaccurate conclusions that TBK1 phosphorylation requires the autophagy machinery. Therefore, overall assumption that both ubiquitination and autophagy subunits are required for TBK1 autophosphorylation appears erroneous.

As the reviewer indicated, phos-TBK1 levels gradually increased in ATG KO cells. The main text was rewritten to more accurately reflect this increase. Based on experiments using the monoclonal antibodies and those conducted during the revision process, it is apparent that although the autophagy machinery may not be completely essential for TBK1 phosphorylation, it clearly facilitates TBK1 phosphorylation in response to Parkin-mediated mitophagy.

Page 12:

- Fig 3a: conclusion that Optn signal is more cytosolic and did not localize with Ub condensates seems speculative as based on:

1- only one immunofluorescence image without statistical analysis

2- Optn and Ub signals are lower in images with Optn is analyzed compared to other images in which NDP52, TAX1BP1 and NBR1 are detected.

To address these concerns, we compared and quantified the signal intensities of all endogenous autophagy adaptors in *FIP200* KO and *ATG9A* KO cells. The quantification data are shown in Fig 3a and the immunofluorescence images are shown in supplementary Fig 6a of the revised manuscript.

- Fig 3b: interpretation of western blot data is uncertain due to lack of appropriate loading control, especially with pellets (P) extracts. In addition, it is not clear how to conclude from the experiments in Fig 3b that autophagy adaptors other than Optn mediate TBK1 phosphorylation.

When autophagy is inhibited, p62 accumulates in the cytosol as aggregates (Komatsu et al. 2007 Cell). Therefore, p62 should be a positive control. Indeed, Fig 3b in the original manuscript (Fig 3b and c in the revised manuscript) showed that the amount of p62 in the pellet fraction was elevated in *FIP200* KO and *ATG9A* KO cells. Furthermore, these aggregates were also observed in the imaging data (Fig 3a and supplementary Fig 7 in the revised manuscript). As the reviewer indicated, the original manuscript did not clarify whether autophagy adaptors other than OPTN mediated TBK1 phosphorylation; however, our revised results clearly demonstrate that TAX1BP1 is the adaptor responsible inducing TBK1 autophosphorylation when basal autophagy is impaired (please see Fig 3f in the revised manuscript).

Minor point: reference is missing in the last sentence of the paragraph stating that K48-linked chains dominate when autophagy pathways are impaired.

While several autophagy adaptors preferentially interact with K48-linked ubiquitin chains (Donaldson et al. 2003 PNAS etc), TRAF6 is recruited to ubiquitin-condensates via p62-mediated K63-linked ubiquitination (Linares et al. 2013 Mol Cell). Furthermore, K33-linked ubiquitin chains are also present in p62-positive condensates (Nibe et al. 2018 Autophagy). Because it's not clear which ubiquitin-linkage is dominant in the condensates, we decided to delete the sentence. We regret the confusion.

Page 13:

Conversely to Optn, they find that the other autophagic receptors localize in insoluble fractions (what does it mean?) independently of TBK1 expression (experiments with DKO cells) and also independently of Optn (where is this shown?). Altogether, these experiments are far from the message of the manuscript. The title of the paragraph "TBK1 activation does not require Optn under basal autophagy conditions" is not correct because even if the level of expression of autophagic receptors and TBK1 phosphorylation are increase in response to the depletion of the autophagy machinery, it does not increase autophagy.

According to the suggestion, we changed the title of the paragraph to "TAX1BP1, but not OPTN, mediates TBK1 phosphorylation when basal autophagy is impaired." In addition, we rewrote this section.

- Fig 3d: authors should mention the nature of the upper band observed in Optn western blot and show the same experiment in solely TBK1 depleted cells since they stated that "electrophoretic migration of Optn was not affected by TBK1 deletion". In addition, suggesting from these sole experiments that "NP52, TAX1BP1, p62, NBR1 and AZI2 form Ub-positive condensates where TBK1 is activated" seems over-interpreted.

Reviewer 3 suggested we characterize the upper band in the OPTN blot (Fig 3d in the original manuscript). To determine if the band is genuine OPTN, we used phostag-PAGE to analyze cell lysates from cells treated with control siRNA or OPTN siRNA and found that both the lower and upper bands were OPTN species (please see "Figure 2 for reviewers" in our response to Reviewer 3). The same pattern was reported by the Wade Harper group (Heo et al. 2015 Mol Cell). They showed that the OPTN double band pattern on phos-tag PAGE was not affected by TBK1 deletion. We have cited this literature where appropriate in the revised manuscript. In WT cells, it is difficult to detect phosphorylation of

autophagy adaptors by TBK1 because basal autophagy constantly degrades them. That's why we used autophagy KO cell lines.

Page 14:

- Fig 4: TBK1 phosphorylation was analyzed in Fig4d and not in Fig4b as stated. In addition, it is rather difficult to conclude from artificial multimerization experiments, as the authors have done, that interaction between Optn and autophagy components contributes to Optn multimerization in genuine conditions.

Detection of phos-TBK1 has been corrected to Fig 4b. Although artificial, the fluoppi assay provides insights into how OPTN activates TBK1 and how the autophagy machinery contributes to TBK1 activation via OPTN. To determine if artificial OPTN multimerization could bypass the autophagy machinery requirement, we used the fluoppi assay. This assay was important for us to conclude that the autophagy machinery and Parkin-mediated ubiquitination allow OPTN to be assembled in close proximity to where TBK1 is activated. The main text was rewritten to better convey the benefits of the fluoppi assay.

Page 15:

This work could have therapeutic consequences but the pathological mutants of TBK1 used affect ALS (Figure 5) while in the discussion it is proposed that monobodies could have a therapeutic interest in familial forms of glaucoma due to the E50K mutation of Optn. It should be better to target only one pathology.

Both TBK1 and OPTN are causative genes for ALS and many pathogenic mutations are known to impact their function. In this study, we focused on ALS mutations in TBK1 that affect self-dimerization and investigated their impact in response to Parkin-mediated mitophagy. We created the monobodies as a tool to physically inhibit OPTN assembly at the contact site. Although our monobodies inhibit Parkin-mediated mitophagy, they would not be a useful therapeutic strategy for ALS due to the loss of function with the TBK1 mutations. However, because TBK1 E50K is a glaucomatous mutation that causes OPTN-TBK1 to bind more tightly, our monobodies might be applicable to glaucomatous pathology since they could disrupt this interaction. We thus feel that it is appropriate to mention the potential of the monobodies and their future utility in the Discussion.

- Fig 5c, d: Authors stated that degree of TBK1 autophosphorylation correlated with OPTN phosphorylation at S177 whereas phosphorylated TBK1 is unaffected by L693Q and V700Q mutants that display decreased phosphorylated Optn In addition, authors interpretation of Figure 5 data is clearly problematic as they stated that:

1- neither 693Q and V700Q mutants had "significant effect on mitophagy", while decreasing efficiency from 78% to 37-51%

2- but conclude that 49.7% mitophagy levels of R357Q mutant is significant mitochondrial degradation.

Overall conclusion that mitophagy efficiency is correlated with phosphorylated TBK1 levels is therefore inaccurate.

We regret that this section did not sufficiently describe the data. Reviewer 3 also noted that the text referencing Fig 5 was difficult to interpret. One of the reasons for the complicated data interpretation is the number of TBK1 mutants used. The L693Q and V700Q mutations used by Li et al. (2016 Nat Commun) were expected to inhibit Parkin-mediated mitophagy since those authors reported that the mutations prevented interactions with OPTN. However, our in-cell assay showed that the two mutations only moderately affected Parkin-mediated mitophagy. Furthermore, both the L693Q and V700Q

mutations were engineered based on the X-ray structure, rather than being authentic pathogenic ALS mutations. To avoid any potential confusion, we decided to remove the L693Q and V700A data. We have re-evaluated the other data and have rewritten this section accordingly. Please see the revised main text.

Discussion

Minor points:

page 20: - reference is missing in the sentence "Optn cannot oligomerize on its own on ubiquitin-decorated mitochondria".

We have provided the appropriate reference.

Major points:

Authors stated that they showed that Optn recruitment to damaged mitochondria, itself, is insufficient for TBK1 autophosphorylation, but did not show experiment of Optn recruitment to mitochondria and its consequences on TBK1 phosphorylation in the absence of mitophagy induction signal. Authors could for example target HA-Ash-6Ub to mitochondria in order to artificially recruit hAG-Optn to "ubiquitinated" mitochondria in the absence of mitophagy signal.

We showed that the efficiency of TBK1 autophosphorylation was reduced in cells expressing the OPTN 4LA/F178A mutant, which cannot interact with the autophagy machinery (Fig 2c and d in the revised manuscript). Cells with FIP200 or ATG9A knockdown also have reduced phos-TBK1 (pS172) as shown in supplementary Fig 5e and f. The rate of phos-TBK1 (pS172) generation in ATG9A KO cells during Parkin-mediated mitophagy is reduced relative to that in WT cells (Fig 2j and k). Since a small amount of phos-TBK1 was generated in both ATG9A knockdown and KO cells (supplementary Fig 5e, f, Fig 2j and k), we concur that it would be premature to conclude that phosphorylation of TBK1 does not occur at all when autophagy core components are absent. A small amount of phos-TBK1 may be generated by OPTN that is freely distributed on the outer mitochondrial membrane. In the revised manuscript, we mention the possibility that TBK1 might be phosphorylated by OPTN independent of the autophagy machinery and were careful to avoid over-interpretation.

As shown in Fig 4, fusing OPTN with an Azami-Green tag can induce artificial multimerization and trigger the generation of phos-TBK1 (pS172). Therefore, we expect that mitochondria-targeted HA-Ash-6Ub would induce TBK1 phosphorylation in a hAG-OPTN-dependent manner as was observed with cytosolic HA-Ash-6Ub (Fig 4). The accumulation of OPTN at the contact site in Parkin-mediated mitophagy is important for TBK1 phosphorylation. Even if OPTN is forced to anchor to the mitochondria, this would induce isolation membrane formation and subsequent autophosphorylation of TBK1. Therefore, the utility of forcing OPTN to anchor to mitochondria is questionable.

Similarly, experimental approaches used by authors lack dynamics parameters to conclude on formation and elongation of isolation membranes and contacts sites that could be probably obtained through video microscopy.

Based on the reviewer's comment, we performed time-lapse microscopy to observe OPTN recruitment to the contact site and followed its movement along with the elongation of isolation membranes during Parkin-mediated mitophagy. HeLa cells stably expressing GFP-OPTN and pSu9-mCherry (a mitochondrial marker) were treated with valinomycin (please see Fig 2l in the revised manuscript). Initial recruitment of GFP-OPTN near mitochondria was evident as small dot-like structures that then elongated over time to become cup-shaped structures and culminated in the formation of spherical structures.

Considering the colocalization of OPTN with WIPI1/WIPI2 (markers of autophagosome formation site) in Fig 2e and supplementary Fig 2a, the time-lapse images strongly suggest that OPTN assembles at contact sites followed by elongation in tandem with isolation membranes during Parkin-mediated mitophagy.

Finally, the model proposed by the authors does not take into account data showing that Optn basally interacts with ubiquitinated mitochondria and LC3 family members (see Wild et al., Phosphorylation of the autophagy receptor optineurin restricts Salmonella growth. Science. 2011 333:228-33), although at lower levels compared to induced conditions, relativizing the impact of the proposed model.

According to the Reviewer 2 comment, we again read the Science paper (Wild et al. 2011) but could not find data showing that OPTN basally interacts with ubiquitinated mitochondria. At least, we think that under steady state conditions without mitophagy induction, mitochondrial ubiquitination and mitochondrial localization of OPTN are undetectable as shown in supplementary Figure 2 in our revised manuscript.

In conclusion, this manuscript represents a lot of work but the experiments often lack controls and are over-interpreted.

****Referees cross-commenting****

In my opinion, what emerges from these 3 reviews is that the results lack controls or have not been repeated enough to support the message that the interaction of Optn with ubiquitin and the ubiquitination machinery is sufficient to activate TBK1. In particular, as reviewer 1 says, the phosphorylation kinetics shown in Figure 1a are not consistent with TBK1 phosphorylation following the interaction of Optn with the ubiquitination machinery and ubiquitin. In Figure 1e, there is a decrease in TBK1 phosphorylation in contrast to WT cells as mentioned by Reviewer 1. In agreement with Reviewer 1, we believe that the WT cells are missing in Figure 1g.

With regard to Figure 2c, we agree with reviewer 1 that an LC3 label is missing in order to be able to interpret the data. In Figure 2e and f, we agree with reviewer 1 that it is difficult to understand why TBK1 phosphorylation increases in the absence of the autophagy machinery (FIP200 KO and ATG5KO). In Figure 3, loading controls are missing for 3b and c. The TBK1 KO cells alone are missing in Fig 2d. In Figure 2b, pTBK1 is missing. In agreement with reviewer 3, we believe that the data with fluoppi contradict the message of the manuscript since they show that TBK1 can be phosphorylated by ubiquitin in the absence of the ubiquitination machinery. In agreement with reviewer 3, we believe that the experiments in Figure 5 are very difficult to interpret. The first reviewer is right to ask the question of the replicates for figures 6c and d.

We appreciate the summary of the reviewers' comments. To address their concerns, we have included the appropriate controls and included the results of three independent experiments in the graphs, which now include appropriate error bars and statistical significance. Thus, we believe we have answered the most critical comments concerning the lack of controls.

In Fig 1a, phos-TBK1 was maximal following 30 min of valinomycin treatment. We confirmed using microscopy-based observations that recruitment of endogenous TBK1 and OPTN and the generation of phos-TBK1 and phos-OPTN at contact sites (marked by WIPI1) near damaged mitochondria was also maximal after 30 min of valinomycin treatment (supplementary Fig 2 and 3). Therefore, the kinetics of phos-TBK1 and phos-OPTN generation are consistent with the recruitment of OPTN-TBK1 to the contact site.

The data presented in Fig 2 clearly indicate that the autophagy components are involved in phos-TBK1 generation during Parkin-mediated mitophagy. Therefore, the claim that ubiquitination machinery is sufficient for TBK1 activation is incorrect. Although we agree that the autophagy gene deletions cannot completely inhibit TBK1 autophosphorylation, mitophagy-dependent generation of phos-TBK1 is largely impaired by *ATG9A* KO (Fig 2j and k). Thus, there is no doubt that isolation membrane formation is important for TBK1 activation following Parkin-mediated mitophagy.

Fig 1e - The reviewer is correct that phos-TBK1 is reduced in the NDP52 knockout. We have rewritten the main text. It is also true that NDP52 has a smaller effect on TBK1 autophosphorylation as compared to OPTN.

Fig 1g - Immunoblots using total cell lysates prepared from six different cell lines (WT, Penta KO alone, Penta KO stably expressing low or high OPTN or NDP52) under four different conditions (DMSO, valinomycin 1 hr, valinomycin 3 hrs, valinomycin + bafilomycin 3 hrs) showed that OPTN is a rate-limiting factor for TBK1 phosphorylation. Please see Fig 1g and h in the revised manuscript

Fig 2c - The recruitment of OPTN WT and associated mutants to the contact site was re-examined by immunostaining with WIPI2 labeling. We found that OPTN WT was both efficiently recruited to and formed the contact site. In contrast, the OPTN 4LA/F178A mutant was unable to interact with FIP200/LC3/ATG9A and was uniformly (i.e. homogenously) distributed on damaged mitochondria with the rate of autophagosome site formation reduced. Please see Fig 2e, f, g in the revised manuscript.

Fig 2e and f - KO of the autophagy core components FIP200 and ATG9A increased phos-TBK1 under basal, non-mitophagy-associated conditions (see Fig 3). The levels of autophagy adaptors other than OPTN also increased in *FIP200* KO and *ATG9A* KO cells. Furthermore, as shown in Fig 3a and supplementary Fig 7, both phos-TBK1 and the autophagy adaptors accumulated in Ub-positive condensates. Based on previous reports (Mejlvang 2018 J Cell Biol), TAX1BP1, p62, and NBR1 have short half-lives and are quickly degraded by macro/micro autophagy. The accumulation of phos-TBK1 in the absence of autophagy occurs because autophagy-dependent degradation of TAX1BP1 (and other adaptors) is inhibited. This allows for the formation of Ub-positive condensates, which brings TBK1 into sufficient proximity for activation. This has been noted in the revised manuscript.

Fig 3b and 3c - We wonder if the "loading controls are missing for Fig 3b and 3c" statement might be a misinterpretation by the reviewer as TOMM20 was used as the loading control in the original Fig 3b. It was recovered in the supernatant fractions of WT, *FIP200* KO, and *ATG9A* KO cells, indicating that the accumulation of autophagy adaptors in the pellet fractions depends on autophagy gene deletion. Similarly, actin and TOMM20 were used as loading controls in the original manuscript Fig 3c.

Fig 2d (perhaps meant to be Fig 3d) – A previous study reported that phos-tag PAGE blot of OPTN in TBK1 KO cells alone revealed no differences between WT and *TBK1* KO cells (Heo et al 2015 Mol Cell). We cited this reference in the revised manuscript.

Fig 2b (perhaps meant to be Fig 4b) - Immunoblots of phos-TBK1 have been incorporated into the results of Fig 4b in the revised manuscript.

Fig 4 - We show in Fig 2 that induction of Parkin-mediated mitophagy promotes OPTN accumulation at contact sites formed by isolation membranes and ubiquitinated mitochondria, and that autophagy core subunits are required for efficient generation of phos-TBK1. Fig 3 shows that phos-TBK1 accumulates in Ub-positive condensates with TAX1BP1, rather than OPTN, and that it is responsible for phos-TBK1 accumulation. Together, these results suggest a model in which TBK1 is activated when OPTN and TBK1 are positioned near each other. We hypothesized that if we could force OPTNs into close proximity the autophagy machinery requirement for TBK1 activation might be bypassed. To assess this model, we designed the fluoppi assay shown in Fig 4. This assay was critical in that it provided an important clue for

the molecular mechanism that OPTN and the autophagy machinery use to cooperatively induce TBK1 trans-autophosphorylation. Because the original manuscript may not have sufficiently conveyed our reasoning for the fluoppi analysis, we have rewritten this section. The main point of the fluoppi assay is that engineered OPTN multimerization was able to bypass the autophagy requirement for TBK1 activation. Fig 5 - For easier interpretation, the L693Q and V700Q data, which are not related to ALS pathology, have been removed.

Fig 5d – Statistical significance has been determined for the mitophagy results and the main text has been rewritten for better clarity.

Fig 6c, d, and I – The experiments were independently replicated more than three times with statistical support and error bars incorporated into the associated graphs.

Reviewer #2 (Significance (Required)):

this manuscript represents a lot of work but the experiments often lack controls and are over-interpreted. The manuscript is for a broad audience.

For the revised manuscript, additional experiments were carefully performed with appropriate controls and the manuscript was rewritten to address concerns regarding over-interpretation. We hope that we have adequately addressed the reviewer's comments.

Reviewer #3 (Evidence, reproducibility and clarity (Required)):

The authors investigated the mechanisms by which TBK1 is phosphorylated and thus activated in PINK1/Parkin-mediated mitophagy. They show data indicating that OPTN, by interacting both with ubiquitin-coated mitochondria and with the autophagy machinery, provides a platform where OPTN-bound TBK1 can be hetero-autophosphorylated by adjacent TBK1.

According to the prevailing model (prior to this manuscript), TBK1 activation via autophosphorylation leads to TBK1-mediated phosphorylation of OPTN S177 and subsequent pOPTN-mediated recruitment of autophagic isolation membranes to the mitochondria. However, based on the model provided in this manuscript, OPTN needs to interact first with both autophagic membranes and ubiquitin before TBK1 can become activated.

This is an important topic. Overall, the experimental data are of high scientific quality. For the most part, the manuscript is clearly written. The figures have been made with great care. The novel insights are relevant. However, a number of issues need to be addressed or clarified.

Major comments:

1. Fig. 1a-b shows that pTBK1 (pS172) formation already peaks after 30 min of valinomycin. Even when bafilomycin is added, pTBK1 level already reaches a near maximum after 30 min of valinomycin. If the model proposed by the authors is correct and pTBK1 (pS172) formation requires extensive interaction of OPTN with both ubiquitin and autophagic isolation membranes, they should be able to show (by immunostaining) that OPTN already extensively forms peri-mitochondrial cup/sphere-shaped structures that colocalize with isolation membrane markers after only 30 min of valinomycin. In the present manuscript, they only show formation of such structures after 1-3 h of valinomycin.

We thank the reviewer for the critical comments. Based on the suggestion, we performed immunostaining to observe the recruitment of TBK1 and OPTN to damaged mitochondria as well as the generation of phos-TBK1 (pS172) and phos-OPTN (pS177). HeLa cells stably expressing Parkin and 3HA-WIP1 were treated with valinomycin for 30 min, and then TBK1, OPTN, phos-TBK1, and phos-OPTN were immunostained along with 3HA-WIP1 (a marker of the autophagosome formation site) and TOMM20 (a mitochondria marker). Please see supplementary Fig 2a and 3a in the revised manuscript. The TBK1, OPTN, phos-TBK1, and phos-OPTN signals formed dot-like, cup-shaped, and/or spherical structures, most of which were peri-mitochondrial and colocalized with 3HA-WIP1. In separate experiments, HeLa cells stably expressing Parkin were treated with valinomycin in the presence or absence of bafilomycin for 30 min or 2 hrs and then immunostained. Please see supplementary Fig 2b in the revised manuscript. After 30 min valinomycin in the absence of bafilomycin, many TBK1 and OPTN signals were observed on damaged mitochondria. These signals were quantified from more than 160 cells for each of the four conditions. Each microscopic image generated contained 18-36 cells and corresponds to one dot in supplementary Fig 2c. Based on these results, the abundance of TBK1 and OPTN on mitochondria after 30 min of valinomycin was much higher than that after 2 hrs with valinomycin (supplementary Fig 2c). Similar results were obtained for phos-TBK1 and phos-OPTN (supplementary Fig 3b and c). These results are consistent with the immunoblot data (Fig1a and b).

Furthermore, we show that Parkin expression levels affect the amount of phos-TBK1 generated during mitophagy. Please see supplementary Fig 4 in the revised manuscript. When *PARKIN* was integrated into HeLa cells under a CMV promoter via an AAVS1 (Adeno-associated virus integration site 1)-locus, the resultant cell line (referred to as high-Parkin) had higher Parkin levels than HeLa cells in which *PARKIN* was introduced by retrovirus infection (referred to as low-Parkin). In high-Parkin HeLa cells, phos-TBK1 levels reached a maximum after 30 min with valinomycin, while in low-Parkin HeLa cells, phos-TBK1 levels were comparable after 30 min and 1 hr. High-Parkin HeLa was used for Fig 1a, b, c, and d as well as supplementary Fig 1, 2, 3 and 4. For all other Figs, *PARKIN* genes were introduced by retrovirus infection. This is one of the reasons why val was used for 30 min in Fig1, but 1-3 hrs for the other Figs. Because 3 hrs valinomycin treatment may be unsuitable for evaluating OPTN recruitment to mitochondria/isolation membrane contact sites, we deleted the original Fig 2c and replaced it with the val 1 hr data (Please see Fig 2e in the revised manuscript).

2. The authors propose that OPTN needs to interact both with ubiquitin on mitochondria and with isolation membrane proteins such as ATG9A to allow TBK1 phosphorylation. However, their fluoppi experiments in Fig. 4 seem to contradict this. In the fluoppi experiments, the authors generate multimeric OPTN-Ub foci and this is apparently sufficient to induced TBK1 phosphorylation at S172 (shown in 4d,f). In this experiment, there is no induction of autophagy or formation of isolation membranes, and TBK1 nevertheless gets activated.

Figure 2 demonstrates that both ubiquitin on mitochondria and formation of the isolation membranes are needed to provide a platform for OPTN to assemble in close proximity to each other and subsequently induce TBK1 autophosphorylation. To determine if OPTN proximity is sufficient for TBK1 autophosphorylation (i.e., if engineered OPTN multimerization can bypass the autophagy machinery requirement for TBK1 autophosphorylation), we used the fluoppi assay. The results clearly showed that engineered OPTN multimerization induced TBK1 autophosphorylation without the need for the autophagy machinery. Although this is not a mitophagy experiment, the fluoppi assay provided crucial insights into the molecular mechanism underlying OPTN-mediated TBK1 autophosphorylation. The main text was rewritten to provide more clarity regarding the purpose of the fluoppi experiments.

3. Can the authors be more concrete/specific in the discussion about the molecular mechanisms that explain why this 'platform' that is created by OPTN-autophagy machinery interactions is so crucial for TBK1 activation? If I understand the model in Fig. 7D correctly, the OPTN-autophagy machinery interactions are mainly important because they reduce the distance between OPTN-bound TBK1 molecules so that they can trans-phosphorylate each other. But if TBK1 autophosphorylation was just a matter of proximity between OPTN-bound TBK1 molecules, interaction of OPTN with densely ubiquitinated mitochondria should already be sufficient for TBK1 phosphorylation. When multiple OPTN molecule bind to one ubiquitin chain or to closely adjacent ubiquitin chains (similar to the fluoppi experiments), TBK1 molecules binding to OPTN would be expected to be already closely enough to one another for trans-autophosphorylation.

The amount of phos-TBK1 during Parkin-mediated mitophagy was reduced in cells with the OPTN 4LA/F178A mutant, which cannot interact with the autophagy machinery (e.g. FIP200, ATG9A, and LC3) but can be targeted to mitochondria (see Fig 2c, d). ATG9A KO cells also had reduced amounts of phos-TBK1 relative to WT cells (See Fig 2j, k). Therefore, rather than OPTN-ubiquitin freely diffusing laterally on the outer membrane, we suggest that the contact site OPTN forms with ubiquitin and the autophagy machinery provides a more suitable platform for TBK1 autophosphorylation because it maintains TBK1 in a proximal position for a longer period of time.

The OPTN UBAN domain binds a ubiquitin-chain composed of two ubiquitin molecules (Oikawa et al. 2016 Nat Comm), and during Parkin-mediated mitophagy only shorter length poly-ubiquitin chains are generated on the mitochondrial surface (Swatek et al. 2019 Nature). Based on those findings, it is unlikely that multiple OPTN bind to one ubiquitin chain. Of course, we cannot rule out the possibility that TBK1 autophosphorylation does not occur on mitochondria in the absence of autophagy components. While full activation of TBK1 requires OPTN to associate with the isolation membrane, initial TBK1 autophosphorylation at the onset of mitophagy may occur based only on the OPTN-ubiquitin interaction. These explanations have been added to the Discussion in the revised manuscript.

Furthermore, based on comments from Reviewer 2, we performed time-lapse microscopy to observe OPTN dynamics during Parkin-mediated mitophagy (please see Fig 2l). HeLa cells stably expressing GFP-OPTN and pSu9-mCherry (a mitochondrial marker) were treated with valinomycin. GFP-OPTN was initially a peri- mitochondrial dot-like structure that elongated over time to a cup-shaped structure and which eventually ended up forming a spherical structure. The time-laps imaging showed that, at least in WT cells, OPTN is directly recruited to the contact sites and elongates along with the isolation membranes. We thus concluded that TBK1 is activated (autophosphorylated) at the contact site rather than on the outer membrane where OPTN-TBK can move freely.

4. Fig. 5c,d and P. 16: the mitophagy experiments in TBK1-/- cells expressing the different mutant forms of TBK1 are hard to interpret because it is not clear which mitophagy differences are statistically significant. The main text about this part (p. 16) is also confusing.

We regret the confusion. Reviewer 2 also noted that the main text for Fig 5 was difficult to interpret. One of the reasons that complicated interpretation of the data is the number of TBK1 mutants used. The L693Q and V700Q mutations used by Li et al. (2016 Nat Commun) were expected to inhibit mitophagy since those authors reported that the mutations prevented interactions with OPTN. However, our in-cell assay showed that the two mutants only moderately affected Parkin-mediated mitophagy. Furthermore, both L693Q and V700Q were engineered based on the X-ray structure and are not ALS pathogenic mutations. To simplify the data and to make data interpretation easier, we decided to delete the L693Q and V700A data. We also determined statistical significance and rewrote this section.

5. Many graphs lack statistics: Fig. 2b (pTBK1), Fig. 2f, Fig. 5b, Fig. 5d, Fig. 6c.

We apologize for the lack of statistical analyses. We repeated experiments (if the experiments had not been independently performed more than three times) with statistical significance and error bars incorporated into the relevant figures.

Other comments:

1. Fig. 1a: how do they know that the upper OPTN band is ubiquitinated OPTN?

Reviewer 2 raised the same question. To demonstrate that the upper OPTN band is ubiquitinated, cell lysates after mitophagy induction were incubated *in vitro* with a recombinant USP2 core domain, and the samples immunoblotted. As shown in supplementary Fig 1 in the revised manuscript, the upper OPTN band disappeared in a USP2-dependent manner. The upper NDP52 and TOMM20 bands similarly disappeared. Therefore, the upper OPTN, NDP52 and TOMM20 bands observed after mitophagy induction are ubiquitinated.

2. Fig. 1a,b: the bafilomycin stabilization of pTBK1, OPTN and pOPTN indicates that these proteins are substantially degraded by autophagy within 30-60 minutes. This seems extremely fast for mitophagy completion. Please discuss.

According to Kulak et al. (2014 Nat Methods), autophagy adaptor abundance (OPTN: 2.32E+4 and NDP52: 3.34E+4 in HeLa cell line) is low compared to that of mitochondria (TOMM20: 1.45E+6 in HeLa cell line). This is one of the reasons why autophagic degradation of adaptors is easier to see. Degradation of phos-TBK1 was likewise easy to detect, whereas total TBK1 was not. This discrepancy is likely based on differences in the abundance of phos-TBK1 and total TBK1. In addition, because autophagy adaptors are localized outside of the mitochondrial membrane they may be easier targets for lysosomal degradation than matrix proteins, which are localized inside the outer and inner membranes.

3. Fig. 1a and rest of the manuscript: is there a reason why the authors only looked at S177 phosphorylation of OPTN and not also at OPTN S473, which is also phosphorylated by TBK1?

Both mass spectrometry and mutational analyses indicated that OPTN S473 is phosphorylated during Parkin-mediated mitophagy and that OPTN phosphorylated at S473 strongly binds ubiquitin chains (Richter et al. 2016 PNAS and Heo et al. 2015 Mol Cell). However, because a phos-S473 OPTN antibody is, to the best of our knowledge, currently not commercially available, we did not focus on S473 phosphorylation.

4. Fig. 1e-f: the main text states that "NDP52 KO effects on the pS172 signal were comparable to controls", but the blot in 1e and the graph in 1f indicate a difference between NDP52KO and WT (significant difference shown in 1f). This is confusing.

We regret the over-interpretation. As the reviewer indicated, the amount of phos-TBK generated in response to mitophagy was reduced in *NDP52* KO cells relative to that in WT cells. This has been corrected. We would like to emphasize that, unlike *OPTN* deletion, *NDP52* deletion has relatively minor effects on TBK1 phosphorylation.

5. P. 9: "TBK1 phosphorylation however was not apparent in the OPTN mutant lines, even after 3 hrs with valinomycin, indicating that autophagy adaptors are essential for TBK1 activation (Fig. 2a)". However, the pTBK1 blot in Fig. 1a does show pTBK1 formation in the OPTN mutant (4LA etc.) lines. This is confusing.

We apologize for this error. We intended to state "TBK1 phosphorylation was not apparent in the Penta KO cells without OPTN expression even after 3 hrs with valinomycin, indicating that autophagy adaptors are essential for TBK1 activation". This sentence has been corrected in the revised manuscript.

6. P. 10: "we subtracted the basal phosphorylation signal from that generated post-valinomycin (1 hr) and bafilomycin (3 hr)". Do they mean "from that generated post-valinomycin (3 hr) and bafilomycin (3 hr)?"

The reviewer is correct, we have corrected the error.

7. P. 10, same paragraph: "the phosphorylation signal was ~90 but was less than 30 in ATG9A KO cells." Unclear what they mean by 90 and 30. 90% and 30%? 90-fold and 30-fold?

The newly generated pTBK1 levels following Parkin-mediated mitophagy were calculated as pTBK1 [val & baf 3 hrs] minus pTBK1 [DMSO]. Since pTBK1 [val & baf 3 hrs] in WT cells is set to 100%, the newly generated pTBK1 in WT cells was $100\% - 5\% = 95\%$. The calculated values for pTBK1 [DMSO] and pTBK1 [val & baf 3 hrs] in ATG9A KO cells were ~55% and ~85%, respectively. Consequently, newly generated pTBK1 in the ATG9A KO cells is $\sim 85\% - \sim 55\% = 30\%$. For clarity, we modified the figure to make the meaning of the numbers more apparent.

8. Fig. 3a: Do they have an idea what kind of ubiquitinated substrates are contained in the ubiquitin-positive condensates that accumulate in FIP200 KO and ATG9A KO cells (i.e. without valinomycin treatment)?

According to Kishi-Itakura et al. (2014 J Cell Sci), ferritin accumulates in the p62 condensates in FIP200 KO and ATG9A KO cells. However, it is unknown if the ferritin in the condensates is ubiquitinated. In the original manuscript, we confirmed by immunostaining that the p62-NBR1 condensates contain ferritin (Fig 3a in the original manuscript and supplementary Fig 7b in the revised manuscript).

9. P. 12 and Fig. 3a: please explain why they look at ferritin, to improve readability.

We thank the reviewer for the suggestion. As mentioned, ferritin is a known substrate that accumulates in p62 condensates, we thus sought to confirm its presence. We have included this explanation in the revised manuscript.

10. Fig. 3a: please also include Ub stain for NBR1.

We thank the reviewer for the suggestion. We obtained a rabbit anti-NBR1 antibody that allowed us to co-immunostain with the mouse anti-ubiquitin antibody (please see supplementary Fig 7b in the revised manuscript).

11. Fig. 3d: the OPTN blot shows 2 OPTN bands. What does the upper OPTN band represent here?

To determine if the two bands are genuine OPTN, total cell lysates prepared from HeLa cells treated with control siRNA or OPTN siRNA were subjected to phos-tag PAGE followed by immunoblotting with an anti-OPTN antibody. As shown below (Figure 2 for reviewers), the two bands (indicated as blue arrowheads) were absent in the OPTN knock down cells, indicating that both are derived from OPTN. Since phosphorylated species migrate slower in phos-tag PAGE, the upper band might be a phosphorylated form. The specific Ser/Thr phosphorylated in OPTN, however, remains to be determined. Heo et al. (2015 Mol Cell) also reported the two OPTN bands on phos-tag PAGE and that both were unchanged in *TBK1* KO cells, suggesting that at least the upper band is not affected by TBK1.

12. P. 14 and Fig. 4b: "Here, we found that phosphorylation of ... *TBK1* (S172) was induced by the *OPTN-ub fluoppi* (Fig. 4b)." However, Fig 4b does not show a p*TBK1* blot.

We immunoblotted phos-TBK1. Please see Fig 4b in the revised manuscript.

Reviewer #3 (Significance (Required)):

The novel insights are relevant.

*According to the prevailing model (prior to this manuscript), *TBK1* activation via autophosphorylation leads to *TBK1*-mediated phosphorylation of *OPTN* S177 and subsequent p*OPTN*-mediated recruitment of autophagic isolation membranes to the mitochondria. However, based on the model provided in this manuscript, *OPTN* needs to interact first with both autophagic membranes and ubiquitin before *TBK1* can become activated.*

Based on our time-lapse microscopy observations (Fig 2I), *OPTN* recruited to the vicinity of mitochondria was visible as a small dot-like structures that likely correspond to contact sites between mitochondria and the isolation membrane since *OPTN* colocalizes with *WIPI1* (please see supplementary Fig 2). These results support our proposed model that *OPTN* interacts with both isolation membranes and ubiquitin at the onset of mitophagy. Without *TBK1* activation, *OPTN* can interact with *ATG9A* vesicles, a seed for isolation membrane formation (Yamano et al 2020 JCB), and *TBK1* can interact with the *PI3K* complex (Nguyen et al 2023 Mol Cell). Therefore, *OPTN-TBK1* can be recruited to the contact site from the very beginning of mitophagy induction prior to *TBK1* being fully activated. Furthermore, the proposed model also includes an *OPTN-TBK1* positive feedback loop; however, the earliest reactions in the positive feedback loop are too difficult to observe. For example, it's widely known that *PINK1* and *Parkin* form a positive feedback loop to generate ubiquitin-chains on damaged mitochondria, but the initial reaction has yet to be observed. It remains unclear if *PINK1* is the first to phosphorylate mitochondrial ubiquitin (if this is the case, it remains unknown how ubiquitin comes to mitochondria) or if cytosolic *Parkin* first adds ubiquitin to the outer membrane albeit with very weak activity. Similarly, in our proposed model, we

Full Revision

cannot determine the earliest OPTN-TBK1 reaction. As described in the Discussion in the revised manuscript, it remains possible that in the absence of autophagy machinery OPTN distributed freely on the outer membrane can induce trans-autophosphorylation, albeit weakly, as the earliest reaction.

We would like to thank Reviewer 3 for the critical comments and suggestions. We have performed several of the suggested experiments, added new data, and rewritten the text. We hope that these changes have sufficiently addressed the reviewer's concerns.

Dear Professor Yamano,

Thank you for submitting the revised version of your manuscript via Review Commons to The EMBO Journal. The manuscript has now been sent back to the three referees who originally appraised the work; I have now heard back from all of them. As you will see from their comments, which I attach to the bottom of this e-mail, you have addressed all of their concerns satisfactorily. This said, I would like you to consider the minor modifications recommended by Reviewer #2.

Before I can formally accept your manuscript for publication, however, there are some remaining editorial points which need to be addressed. In this regard would you please:

- include up to five key words
- alphabetize the reference list, using et al. after the tenth author name,
- rename the Conflict of Interest statement the 'Disclosure and Competing Interests' statement,
- remove the author credit section from the manuscript,
- check the callout for Supplementary Table 1, no such table is included in the manuscript files,
- complete and return the author checklist,
- upload main figures as individual, high-res files and moved the legends so they appear after the References in the main manuscript file,
- include a table of contents with page numbers in Appendix Figure 1; use the nomenclature Appendix Figure S1-S8 in this table of contents, as well as in figure legends and callouts, and
- remove legends for Supplemental Figures from main manuscript file and insert them below the corresponding figures in the Appendix PDF.

My colleague Hannah Sonntag will contact you separately about supplying us with Source Data for all figures and supplementary figures.

Additionally, could you please supply the synopsis image in jpeg, TIFF or png format, and remove the bullet points for the summary from the main file. We include a two sentence synopsis of the paper (see <http://emboj.embopress.org/>). Please provide me with this summary along with 3-5 bullet points that capture the key findings of the paper.

EMBO Press is an editorially independent publishing platform for the development of EMBO scientific publications.

Best wishes,

William

William Teale, PhD
Editor
The EMBO Journal
w.teale@embojournal.org

We realize that it is difficult to revise to a specific deadline. In the interest of protecting the conceptual advance provided by the work, we recommend a revision within 3 months (18th Mar 2024). Please discuss the revision progress ahead of this time with the editor if you require more time to complete the revisions. Use the link below to submit your revision:

Referee #1:

In this manuscript, Yamano and colleagues show that Optn provides a platform for TBK1 activation by autophosphorylation and that TBK1 is activated after the interaction of Optn with the autophagy machinery and ubiquitin and not before. They show that TBK1 phosphorylation is induced during autophagy. Both the interaction of Optn with autophagosomes and the autophagy core are shown to be required for TBK1 activation. Under basal conditions, depletion of the autophagosome machinery leads to an increase in autophagy receptors (except Optn) and TBK1 phosphorylation which colocalize with ubiquitin in insoluble moieties.

Compare to the previous version, many changes were made to the manuscript and most criticisms were addressed. The manuscript is now suitable for publication in EMBO J

Referee #2:

Revision by Yamano et al. corroborates their study on the mechanism of TBK1 activation during Parkin-mediated mitophagy. All my previous concerns have been convincingly addressed. The new data is of high quality and improves the manuscript.

Minor points related to the new data that should be resolved are:

- More detailed description of image analysis, i.e. how was it decided/quantified whether OPTN localizes to mitochondria or not (Fig. 1l)? Methods section should contain information on whether a script was used or judgment was performed manually? Moreover, it would be helpful to show examples of the cytosol/homo/heterogenous OPTN distribution in Fig. 1k/l as done (or instead of) in Fig. 2g.
- Scale bars in the microscopy images Fig. 2g are missing.
- Fig. 2i lacks annotation of what samples 1/2/3 are.

Referee #3:

The authors have adequately addressed my concerns.

TOKYO MEDICAL AND DENTAL UNIVERSITY

Koji Yamano Ph.D.
Medical Research Institute
Tokyo Medical and Dental University
1-5-45 Yushima, Bunkyo-ku,
Tokyo, 113-8510, JAPAN
Tel: +81-3-5803-5313
E-mail: kojibiom@tmd.ac.jp

January 10, 2024

Dear Editor,

We would like to submit our revised manuscript "Optineurin provides a mitophagy contact site for TBK1 activation" Manuscript ID: EMBO-J-2023-116123.

We would like to thank the editor and reviewers for investing tremendous efforts and times for our manuscript. Our point-by-point responses are indicated below.

Reviewer #2

Revision by Yamano et al. corroborates their study on the mechanism of TBK1 activation during Parkin-mediated mitophagy. All my previous concerns have been convincingly addressed. The new data is of high quality and improves the manuscript.

Minor points related to the new data that should be resolved are:

- More detailed description of image analysis, i.e. how was it decided/quantified whether OPTN localizes to mitochondria or not (Fig. 1l)? Methods section should contain information on whether a script was used or judgment was performed manually?

We manually quantified OPTN recruitment to damaged mitochondria (Figs 1L and 2G) and the number of WIPI2 foci near mitochondria (Fig 2F). We explained them in the figure legends.

Moreover, it would be helpful to show examples of the cytosol/homo/heterogenous OPTN distribution in Fig. 1k/l as done (or instead of) in Fig. 2g.

We added example images for OPTN distribution to Fig 1L.

- Scale bars in the microscopy images Fig. 2g are missing.

We apologize for this point. The scale bars were now added to Figs 2G and 2L.

- Fig. 2i lacks annotation of what samples 1/2/3 are.

We apologize for this point. The annotation of samples 1/2/3 were now added to Fig 2I.

Editorial points that need to be addressed

- include up to five key words

We included five key words in the revised manuscript.

- alphabetize the reference list, using et al. after the tenth author name

We corrected it accordingly.

- rename the Conflict of Interest statement the 'Disclosure and Competing Interests' statement

We corrected it accordingly.

- remove the author credit section from the manuscript

We removed the author contributions from the manuscript.

- check the callout for Supplementary Table 1, no such table is included in the manuscript files

We are sorry for have not uploaded Supplementary Table 1. We uploaded it when submitting a revised version.

- complete and return the author checklist

We completed and uploaded the author checklist.

- upload main figures as individual, high-res files and moved the legends so they appear after the References in the main manuscript file

We moved the figure legends after the references in the main manuscript.

- include a table of contents with page numbers in Appendix Figure 1; use the nomenclature Appendix Figure S1-S8 in this table of contents, as well as in figure legends and callouts

We made an Appendix file that includes Table S1 (reagents and cell lines), Table S2 (antibodies), Table S3 (plasmids and siRNAs), Figs S1-S3 with a table of contents.

- remove legends for Supplemental Figures from main manuscript file and insert them below the corresponding figures in the Appendix PDF.

We removed legends for Supplemental Figures from main manuscript. We moved some of the original Supplemental Figures to Appendix, and others to Figure EV.

*Please define the annotated p values *****/**/* in the legend of figure 1d, f, j, h, l; 2b, d, f-g, i, k; 3c; 4e-f; 5b, d; 6c, i; 7c; supplementary figures 2c; 3c; 5c, f; 6b; as appropriate.*

We added them to the figure legends accordingly.

Please indicate the statistical test used for data analysis in the legends of figures 1d, f, j, h, l; 2b, d, f-g, i, k; 3c; 4e-f; 5b, d; 6c-d, i; 7c; supplementary figures 2c; 3c; 5c, f; 6b."

We indicate the statistical test in the figure legends accordingly.

Please note that the box plots need to be defined in terms of minima, maxima, centre, bounds of box and whiskers, and percentile in the legends of figures 2f; 3a; 4e-f.

We added information for the box plots to the figure legends accordingly.

Please note that information related to n is missing in the legend of figure 2f."

We corrected it accordingly.

Please note that scale bar and its definition are missing for figure 2l.

We corrected it accordingly.

Please note that the blue arrowheads are not defined in the legends of figures 1a, d-e, i; 2a, c, h, j; 3d; 5a, e; supplementary figures 1; 4a; 5a-b, d-e; This needs to be rectified.

We corrected it accordingly.

Please note that the orange and blue arrowheads are not defined in the legends of figures 1c; 4b; 6h; 7b; This needs to be rectified.

We corrected it accordingly.

Please note that the blue arrowheads and dots are not defined in the legend of figure 1g; This needs to be rectified.

We corrected it accordingly.

Please note that the blue dots are not defined in the legend of figure 3e. This needs to be rectified."

We corrected it accordingly.

All corrections in the revised manuscript are highlighted in yellow. With all those alterations we made, we hope that our manuscript is now suitable for publication in the EMBO Journal.

Sincerely,

Koji Yamano, Ph.D.
Associate Professor, Tokyo Medical and Dental University

Dear Koji,

I am pleased to inform you that your manuscript has been accepted for publication in the EMBO Journal.

Congratulations on a really insightful study!

Best wishes,

William

William Teale, PhD
Editor
The EMBO Journal
w.teale@embojournal.org
